# LONG-CONTEXT GENERALIZATION WITH SPARSE ATTENTION

**Pavlo Vasylenko[1,2], Hugo Pitorro[2], André F. T. Martins[1,2,3,4], Marcos Treviso[1,2,4]**
[1]Instituto Superior Técnico, Universidade de Lisboa, [2]Instituto de Telecomunicações,
[3]TransPerfect, [4]ELLIS Unit Lisbon

## ABSTRACT

Transformer-based architectures traditionally employ softmax to compute attention weights, which produces dense distributions over all tokens in a sequence. While effective in many settings, this density has been shown to be detrimental for tasks that demand precise focus on fixed-size patterns: as sequence length increases, non-informative tokens accumulate attention probability mass, leading to dispersion and representational collapse. We show in this paper that dynamically sparse attention mechanisms using $\alpha$-entmax can avoid these issues, due to their ability to assign exact zeros to irrelevant tokens. Furthermore, we introduce Adaptive-Scalable Entmax (ASEntmax), which endows $\alpha$-entmax with a learnable temperature parameter, allowing the attention distribution to interpolate between sparse (pattern-focused) and dense (softmax-like) regimes. Our empirical evaluation on synthetic tasks and language modeling demonstrates that ASEntmax substantially outperforms softmax, scalable softmax, and fixed-temperature $\alpha$-entmax baselines, achieving up to $1000\times$ length extrapolation on synthetic benchmarks and superior long-context generalization on language modeling while preserving short-context performance, including better perplexity trends and higher retrieval accuracies at $8\times$ training length. Source code: https://github.com/deep-spin/asentmax

## 1 INTRODUCTION

The transformer architecture (Vaswani et al., 2017) has become the foundation of modern large language models (LLMs), establishing new benchmarks across diverse domains. However, as researchers push these models toward increasingly longer contexts—from thousands to millions of tokens—several fundamental limitations emerge that can be traced to the **softmax** transformation used in attention. Three critical limitations stand out: **representational collapse** occurs due to softmax's inability to maintain distinct attention patterns as sequence length grows, erasing meaningful distinctions between tokens (Barbero et al., 2024); **over-squashing** is exacerbated by softmax's dense probability distribution, leading to exponential dilution of gradients (Alon & Yahav, 2021; Barbero et al., 2024); and **attention dispersion** arises from softmax's fundamental property that forces probability mass to be distributed across all tokens, with attention weights necessarily approaching an uniform distribution as context grows (Veličković et al., 2025; Nakanishi, 2025).

Previous approaches to address these challenges include positional encoding innovations such as ALiBi (Press et al., 2022) and RoPE (Su et al., 2024), which help to mitigate position bias issues. Recent works directly target the root cause—the softmax function itself. Nakanishi (2025) proposes Scalable-Softmax to scale logits based on context length, while Veličković et al. (2025) identify fundamental limitations of softmax for sharp out-of-distribution generalization and propose to learn adaptive temperatures to control the sharpness of softmax. While effective, these solutions often require careful tuning or address only a subset of the challenges.

In this paper, we address the root cause of these problems by replacing softmax with $\alpha$-**entmax** (Peters et al., 2019), a *differentiable* sparse transformation that induces probability distributions where irrelevant tokens receive *exactly zero attention*. While $\alpha$-entmax has been used successfully in transformers (Correia et al., 2019; Gonçalves et al., 2025), its length generalization properties, to the

Figure 1: Long-context generalization on Multi-query Multi-token Associative Recall (left) and Max Retrieval (right). SSMax represents the Scalable Softmax approach by Nakanishi (2025), and Adaptive Temperature (Adapt. Temp.) represents the approach by Veličković et al. (2025). While all methods benefit from using NAPE (NoPE + ALiBi), our adaptive-scaling version of $\alpha$-entmax exhibits the best extrapolation results, effectively handling extremely long sequences.

best of our knowledge, have never been studied. We show theoretically and empirically that $\alpha$-entmax consistently helps to address challenges in long context modeling. Our key contributions include:[1]

- **Non-dispersion**: We establish that $\alpha$-entmax attention distributions maintain consistent focus regardless of sequence length, with entropy bounded by $\mathcal{O}(\log s)$ rather than approaching maximum entropy $\mathcal{O}(\log n)$ as with softmax, where $s \ll n$ is the number of tokens with nonzero probability.

- **Representational preservation**: $\alpha$-entmax attention with sparse support can avoid *representational collapse* and reduces the number of gradient paths from $\mathcal{O}(n^L)$ to $\mathcal{O}(s^L)$, alleviating *over-squashing* by strengthening the gradient flow for long-range dependencies.

- **Adaptive-scalable $\alpha$-entmax**: We introduce **ASEntmax**, which adaptively adjust sparsity based on sequence length to maintain optimal token selection even in extremely long contexts.

- **Empirical results**: We demonstrate that ASEntmax achieves superior performance across synthetic and real-world tasks. For example, as shown in Figure 1, ASEntmax achieves 95.3% accuracy on associative recall at 65K tokens after training on just 64 tokens—a $1000\times$ length extrapolation.

## 2 BACKGROUND

### 2.1 TRANSFORMERS

In this work, we study (causal) transformers with sparse attention distributions created by replacing softmax with $\alpha$-entmax. We present the precise mathematical formulation below, following closely the notation from (Barbero et al., 2024). Concretely, given a sequence of token embeddings $\boldsymbol{X} \in \mathbb{R}^{n \times d}$, where $n$ is the sequence length and $d$ is the hidden dimension, transformers compute query, key, and value projections $\boldsymbol{Q} = \boldsymbol{X}\boldsymbol{W}_Q$, $\boldsymbol{K} = \boldsymbol{X}\boldsymbol{W}_K$, and $\boldsymbol{V} = \boldsymbol{X}\boldsymbol{W}_V$. We denote with $\boldsymbol{q}_i, \boldsymbol{k}_i, \boldsymbol{v}_i \in \mathbb{R}^d$ the $d$-dimensional query, key, and value vectors of the $i$-th token. For each query position $i$, the representation at layer $\ell$ for the $i$-th token is computed as:

$$\boldsymbol{u}_i^{(\ell)} = \sum_{j \leq i} p_{ij}^{(\ell)} \text{norm}_1^{(\ell)}\left(\boldsymbol{v}_j^{(\ell-1)}\right) + \boldsymbol{v}_i^{(\ell-1)}, \quad \boldsymbol{v}_i^{(\ell)} = \text{FFN}^{(\ell)}\left(\text{norm}_2^{(\ell)}\left(\boldsymbol{u}_i^{(\ell)}\right)\right) + \boldsymbol{u}_i^{(\ell)}, \quad (1)$$

where $p_{ij}^{(\ell)}$ are attention weights, $\text{FFN}^{(\ell)}$ is the feed-forward network, $\text{norm}(\cdot)$ represent LayerNorm modules (Xiong et al., 2020). The output is computed as $\boldsymbol{y}_i = \text{norm}_3\left(\boldsymbol{v}_i^{(L)}\right)$. The attention weights $p_{ij}^{(\ell)} = \pi(\boldsymbol{z}_i^{(\ell)})_j$ are computed by applying a transformation $\pi : \mathbb{R}^n \to \triangle_n$ to the attention logits $z_{ij}^{(\ell)} = \langle \boldsymbol{q}_i^{(\ell)}, \boldsymbol{k}_j^{(\ell)} \rangle / \sqrt{d}$, where $\triangle_n := \{\boldsymbol{p} \in \mathbb{R}^n : \boldsymbol{p} \geq \boldsymbol{0}, \boldsymbol{1}^\top \boldsymbol{p} = 1\}$ represents the probability simplex. Standard transformers employ the softmax function as $\pi$. In this work, we study transformers by casting $\pi$ as the $\alpha$-entmax transformation.

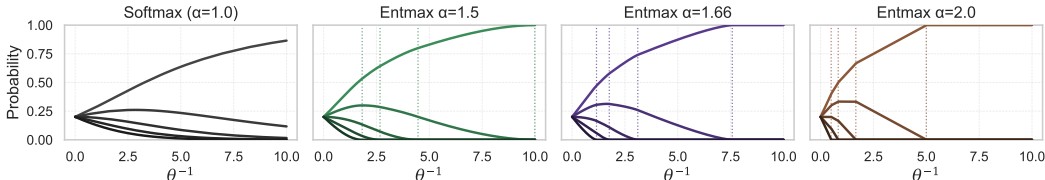

Figure 2: Visualization of $\alpha$-entmax$(\boldsymbol{z}/\theta)$ for different values of $\alpha$. Each panel shows how probability mass is distributed among five elements of $\boldsymbol{z} = [2.0, 1.8, 1.6, 1.4, 1.2]$ as the temperature parameter decreases ($\theta^{-1}$ increases). The vertical lines show the temperature that leads to zero probability.

## 2.2 $\alpha$-ENTMAX

$\alpha$-entmax (Peters et al., 2019) is a **differentiable** transformation that generalizes softmax by allowing for **sparse** probability distributions. For an input vector $\boldsymbol{z} \in \mathbb{R}^n$ and $\alpha > 1$, $\alpha$-entmax is defined as:

$$\alpha\text{-entmax}(\boldsymbol{z})_i = [(\alpha - 1)z_i - \tau(\boldsymbol{z})]_+^{\frac{1}{\alpha-1}} , \tag{2}$$

where $[\cdot]_+ := \max(0, \cdot)$ and $\tau : \mathbb{R}^n \to \mathbb{R}$ yields a threshold that ensures the resulting distribution sums to 1. A key property of $\alpha$-entmax is that tokens with scores below the threshold receive **exactly zero probability**, creating sparse attention patterns. When $\alpha \to 1^+$, this reduces to the standard softmax function. The sparsity level increases with $\alpha$, with $\alpha = 2$ corresponding to the sparsemax function (Martins & Astudillo, 2016). Figure 2 illustrates $\alpha$-entmax for different values of $\alpha$. We provide more information on $\alpha$-entmax in §A. While $\alpha$-entmax is a suitable choice for sparse attention, its theoretical and empirical impact on long inputs is still unclear. In the next section, we demonstrate how it fundamentally changes the way attention behaves for long contexts.

## 3 THEORETICAL PROPERTIES OF $\alpha$-ENTMAX FOR LONG CONTEXTS

We analyze the theoretical properties of $\alpha$-entmax that make it especially suitable for long-context modeling, focusing on how it addresses the fundamental limitations of softmax.

### 3.1 NON-VANISHING ATTENTION PROBABILITIES

A critical limitation of softmax in transformers is that attention weights inevitably decrease as the sequence length increases. Our first result demonstrates how $\alpha$-entmax avoids this issue.

**Lemma 1** (Non-Vanishing Attention Property). *Consider scalars $a_1, ..., a_{n-1}, c \in \mathbb{R}$. Let $\boldsymbol{x} = [a_1, ..., a_{n-1}, c]^\top \in \mathbb{R}^n$ and $\boldsymbol{x}^* = [a_1, ..., a_{n-1}, b, c]^\top \in \mathbb{R}^{n+1}$, with all entries bounded. The following properties hold:*

- *For all $\alpha \geq 1$, we have $\alpha$-entmax$(\boldsymbol{x})_n \geq \alpha$-entmax$(\boldsymbol{x}^*)_{n+1}$. In the softmax case ($\alpha = 1$), Barbero et al. (2024, Lemma B.1) have shown that the inequality is always strict: softmax$(\boldsymbol{x})_n >$ softmax$(\boldsymbol{x}^*)_{n+1}$.*

- *For all $\alpha > 1$, there is some $b_{\max} \in \mathbb{R}$ such that, for any $b \leq b_{\max}$, we have $\alpha$-entmax$(\boldsymbol{x})_n = \alpha$-entmax$(\boldsymbol{x}^*)_{n+1}$.*

*Furthermore, for $\alpha > 1$, the difference $\alpha$-entmax$(\boldsymbol{x})_n - \alpha$-entmax$(\boldsymbol{x}^*)_{n+1}$ can take any value in $[0, \alpha$-entmax$(\boldsymbol{x})_n]$ by appropriate choice of $b$.*

The proof can be found in §D.1. This result demonstrates a fundamental difference between softmax and $\alpha$-entmax. Unlike softmax, where adding a new token always reduces the attention probability of existing tokens strictly, $\alpha$-entmax allows a distinct behavior: the attention probability can remain unchanged. This occurs because the $\alpha$-entmax's thresholding effect allows tokens with logits below a certain threshold to receive exactly zero attention, letting the model focus only on the relevant tokens.

---

[1]Our code will be available upon acceptance.

Having established that $\alpha$-entmax prevents the vanishing of individual attention weights, we now formalize the broader concept of attention dispersion to better understand how attention distributions as a whole behave as the sequence length increases.

## 3.2 ATTENTION DISPERSION AND CONCENTRATION

Recent work by Nakanishi (2025) and Veličković et al. (2025) has highlighted attention dispersion as a fundamental limitation of softmax for long context generalization. Building upon these insights, we provide a formal definition to characterize attention dispersion and show how $\alpha$-entmax naturally exhibits concentration properties that address these limitations.

**Definition 1** (Attention Dispersion). *Let $f : \mathbb{R}^n \to \triangle_n$ denote a transformation (such as softmax) mapping logits to the probability simplex $\triangle_n := \{\boldsymbol{p} \in \mathbb{R}^n \; : \; \boldsymbol{p} \geq \boldsymbol{0}, \boldsymbol{1}^\top \boldsymbol{p} = 1\}$.*

*1. $f$ exhibits **complete dispersion** if for any bounded sequence of logits $(z_n)_{n \in \mathbb{N}}$, the normalized entropy approaches 1 as the sequence length increases:*

$$\lim_{n \to \infty} \frac{H(f(\boldsymbol{z}_{1:n}))}{\log n} = 1. \tag{3}$$

*2. $f$ exhibits **concentration resilience** if there are bounded sequences of logits where the normalized entropy remains bounded away from 1:*

$$\lim_{n \to \infty} \frac{H(f(\boldsymbol{z}_{1:n}))}{\log n} < 1. \tag{4}$$

These definitions allow us to examine how softmax and $\alpha$-entmax behave as sequence length grows:

**Proposition 1** (Dispersion Properties of Attention Mechanisms). *Comparing softmax and $\alpha$-entmax ($\alpha > 1$) attention mechanisms:*

*1. **$\alpha$-entmax can retain probability, while softmax always leaks:** For any $\alpha > 1$ and any logits $\boldsymbol{z} \in \mathbb{R}^n$, there are logits $\boldsymbol{z}^* \in \mathbb{R}^N$ with $N > n$ such that:*

$$\alpha\text{-entmax}(\boldsymbol{z})_i = \alpha\text{-entmax}(\boldsymbol{z}^*)_i \quad \forall i \leq n. \tag{5}$$

*This is impossible for $\alpha = 1$ (softmax), for which we always have $\text{softmax}(\boldsymbol{z})_i > \text{softmax}(\boldsymbol{z}^*)_i$.*

*2. **Softmax exhibits complete dispersion:** For any fixed temperature $\theta > 0$ and any bounded sequence of logits $(z_n)_{n \in \mathbb{N}}$:*

$$\lim_{n \to \infty} \frac{H(\text{softmax}(\boldsymbol{z}_{1:n}/\theta))}{\log n} = 1. \tag{6}$$

*3. **$\alpha$-entmax can exhibit strong concentration resilience:** When the support size grows sublinearly as $|\mathcal{S}| = \mathcal{O}(n^\beta)$ with $\beta < 1$, $\alpha$-entmax maintains bounded normalized entropy:*

$$\lim_{n \to \infty} \frac{H(\alpha\text{-entmax}(\boldsymbol{z}_{1:n}))}{\log n} \leq \beta < 1. \tag{7}$$

The full proof can be consulted in §C. The key takeaway of this result is that the entropy of attention distributions reveals how concentrated or dispersed they are across tokens. While softmax distributions approach maximum entropy $\Theta(\log n)$ as the sequence length increases (indicating complete dispersion), $\alpha$-entmax distributions maintain bounded entropy $\mathcal{O}(\log s)$ where $s$ is the support size. This allows models with $\alpha$-entmax to maintain focused, low-entropy attention patterns even when processing extremely long sequences, as long as the support size is smaller than the full sequence length. This non-dispersion property means transformers with $\alpha$-entmax **can scale to very long contexts without the attention becoming dispersed**, maintaining their ability to focus on relevant information regardless of how much additional context is present. However, attention dispersion is not the only obstacle to effective long-sequence modeling.

## 3.3 Representational Preservation and Over-squashing Alleviation

Two other critical challenges in long-context transformers are *representational collapse* and *over-squashing*, both exacerbated by the diffuseness of softmax attention (Barbero et al., 2024). More concretely, *collapse* means $\|\boldsymbol{v}_i^{(L)} - \boldsymbol{v}_j^{(L)}\|_1 \to 0$ as $n \to \infty$, and *over-squashing* means gradients from distant inputs vanish across $\mathcal{O}(n^L)$ paths. Here, we show that $\alpha$-entmax can mitigate both:

**Proposition 2** (Representational Preservation and Reduced Gradient Paths). *Let $\alpha > 1$. Consider a depth-$L$ transformer with residual connections and attention weights given by $\alpha$-entmax. Suppose each attention distribution has support size at most $s$ with $s \ll n$. Then:*

1. *(**Preserved representations**) There exist input families $\boldsymbol{v}^{(0)} \in \mathbb{R}^{n \times d}$ and $\boldsymbol{v}^{*(0)} \in \mathbb{R}^{(n+1) \times d}$ and a constant $c > 0$ such that $\|\boldsymbol{v}_n^{(L)} - \boldsymbol{v}_{n+1}^{*(L)}\|_1 \geq c$ for all $n$; i.e., $\alpha$-entmax can maintain distinct token representations as $n \to \infty$. In contrast, Barbero et al. (2024) shows that for softmax attention, $\|\boldsymbol{v}_n^{(L)} - \boldsymbol{v}_{n+1}^{*(L)}\|_1 \to 0$ as $n \to \infty$.*

2. *(**Alleviated over-squashing**) The number of effective gradient paths scales as $\mathcal{O}(s^L)$ (rather than $\mathcal{O}(n^L)$ under softmax), alleviating over-squashing by concentrating gradient flow on fewer active paths.*

*Full statements and proofs are in App. §D.2–§D.3.*

**Empirical evidence.** We corroborate Proposition 2 with controlled experiments. For representational *collapse*, following the Barbero et al. (2024)-style one-token extension probe, we implement a counterexample with $\alpha \in \{1.0, 1.5, 1.75, 2.0\}$, and report the $L_1$ gap between $\boldsymbol{v}_n^{(L)}$ and $\boldsymbol{v}_{n+1}^{*(L)}$. With softmax attention, the gap rapidly decays toward $0$ with length, while $\alpha$-entmax preserves a non-vanishing margin up to 128K tokens, with larger $\alpha$ yielding stronger preservation (App. §D.2). For over-squashing, we analyzed gradient flow through an 8-layer network on a copying task, where the model must copy information across long distances. We find that $\alpha$-entmax attention sustains substantially larger norms across depths and lengths, consistent with $\mathcal{O}(s^L)$ path growth, whereas softmax degrades sharply with sequence length (App. §D.3).

## 4 Adaptive-Scalable $\alpha$-entmax (ASEntmax)

In the previous section we saw that $\alpha$-entmax, for any choice of $\alpha > 1$, can avoid some of the pitfalls of softmax thanks to its ability to assign zero weight to many tokens, ignoring irrelevant information. But what if it ignores *too many* tokens? Can it handle situations where many tokens are *relevant* and should be attended? We show in this section that indeed the model might not be able to cope with this for a fixed $\alpha$ and a fixed temperature, and we propose a practical solution.

### 4.1 Controlling Sparsity in Long Contexts via ASEntmax

As sequence length grows, the spread of attention logits increases—for IID Gaussian logits, the expected range satisfies $\mathbb{E}[\Delta] \sim 2\sigma\sqrt{2\log n}$ (Kamath, 2015). With a fixed temperature, this makes attention overly peaky at long $n$. We address this with *Adaptive-Scalable $\alpha$-entmax* (ASEntmax), which rescales logits *per head* as a function of context length and content:

$$\text{ASEntmax}(\boldsymbol{z}) = \alpha\text{-entmax}((\delta + \beta(\log n)^\gamma)\boldsymbol{z}), \tag{8}$$

where $\beta, \gamma, \delta \in \mathbb{R}$ are head-specific scalars (inverse temperature). Concretely, for each head, we obtain vectors $\boldsymbol{\beta}$ and $\boldsymbol{\gamma}$ whose entries contain these coefficients for each query:

$$\boldsymbol{\beta} = \text{softplus}(\boldsymbol{X}\boldsymbol{w}_\beta) \in \mathbb{R}_+^n, \quad \boldsymbol{\gamma} = s\tanh(\boldsymbol{X}\boldsymbol{w}_\gamma) \in (-s, s)^n, \tag{9}$$

where $\boldsymbol{w}_\beta, \boldsymbol{w}_\gamma \in \mathbb{R}^d$ are learnable, head-specific projection vectors. This characterization allows the model to learn a slowly rising ($\gamma > 0$) or dampening ($\gamma < 0$) temperature schedule without interfering in the positional encodings (which would happen with negative values of $\beta$). Specifically, for IID Gaussian logits $\mathcal{N}(0, \sigma)$, when $\delta = 0$ and $\gamma = -0.5$ the scaling counteracts the growth of logit ranges ($\Delta_n$) as context increases:

$$\beta(\log n)^{-0.5} \cdot \Delta_n = \beta(\log n)^{-0.5} \cdot 2\sigma\sqrt{2\log n} = 2\sigma\beta\sqrt{2}, \tag{10}$$

Figure 3: Learned positions per head. Besides a simple linear fit baseline ($\beta n$), we also show the fit given by $\delta + \beta \log n$ and $\delta + \beta(\log n)^\gamma$, which are used by SSMax and ASEntmax, respectively. These plots further reinforce the idea that having a $\gamma$ parameter is beneficial for length extrapolation.

which remains constant as $n$ increases, preventing excessive sparsification. Furthermore, with this parameterization, ASEntmax can recover standard $\alpha$-entmax when $\beta = 0$, hence allowing a smooth transition between scaled and unscaled regimes. By making $\boldsymbol{w}_\beta$ and $\boldsymbol{w}_\gamma$ head-specific and learnable, ASEntmax can adapt to the optimal scaling behavior for each head, balancing the natural concentration benefits of $\alpha$-entmax with precise control over how sparsity patterns evolve with sequence length. Finally, we note that simply scaling the query-key products is appealing from a practical perspective since it allows the direct use of fast optimized kernels for $\alpha$-entmax, such as AdaSplash (Gonçalves et al., 2025), without any modifications.

**Empirical Analysis.** To empirically validate the importance of the parameter $\gamma$ in our proposed scaling formulation, we conducted experiments on a language modeling task using a 120M-parameter transformer trained on 5B tokens from the FineWeb dataset (Penedo et al., 2024). Following the methodology from SSMax (Nakanishi, 2025), we implemented learnable scaling parameters for the attention logits, but with a key difference: while Nakanishi (2025) uses a global scaling parameter, we learn separate scaling parameters for each attention head, motivated by Correia et al. (2019)'s finding that attention heads develop distinct sparsity patterns. Figure 3 presents the learned scaling behaviors for representative attention heads, along with fitted curves from different scaling models. First, we note that different heads learn *significantly distinct patterns*, highlighting the need of head-specific scaling. Second, the results demonstrate that a simple log-scaling model $\delta + \beta \log n$ provides poor fits for many heads. In contrast, the inclusion of a $\gamma$ power provides consistently better fits across different attention heads. The complete distribution of fitted $\beta$ and $\gamma$ values across all heads is provided in §F, and additional training details can be found in §F.1.

## 4.2 INTERACTION WITH POSITIONAL ENCODING

Sparse transformations do not merely reweight attention, they effectively *change* which edges exist in the attention graph. This makes the choice of positional encoding especially important when replacing softmax with $\alpha$-entmax. Recall that a head computes logits of the form $z_{ij} = \boldsymbol{q}_i^\top \boldsymbol{k}_j + b_{ij}$, where $b_{ij}$ is the relative positional contribution (possibly zero). Under $\alpha$-entmax, token $j$ is attended by query $i$ iff it survives the query-dependent thresholding $(\alpha - 1) z_{ij} > \tau(\boldsymbol{z}_i)$. Thus, positional encodings acting on $z_{ij}$ through $b_{ij}$ directly affect the set of active tokens.

With **NoPE** (Kazemnejad et al., 2023) we have $b_{ij} = 0$, so (apart from causal masking) attention scores are purely content-based. This can be beneficial for semantic retrieval, but may be less robust under large length shifts without an explicit positional prior. With **ALiBi** (Press et al., 2022) we add a head-specific linear distance bias $b_{ij} = m_h(j - i)$ with $m_h > 0$, inducing a smooth recency preference under softmax. Under $\alpha$-entmax, thresholding yields exact zeros: any position whose biased score falls below the entmax threshold is pruned. Since the ALiBi penalty decreases with distance, for finite content logits there is an *effective* head- and query-dependent cutoff beyond which sufficiently distant tokens receive exactly zero mass, yielding a localized support. With **RoPE** (Su et al., 2024), relative position enters through rotations in query-key interactions, producing oscillatory distance-dependent score contributions; combined with $\alpha$-entmax, this can lead to *content-dependent sparse supports* whose active distances vary with the representations and RoPE frequencies, and the resulting sparsity can be more fragmented. We visualize this interaction in Fig. 4, and provide additional theoretical statements and empirical evaluations in §E.

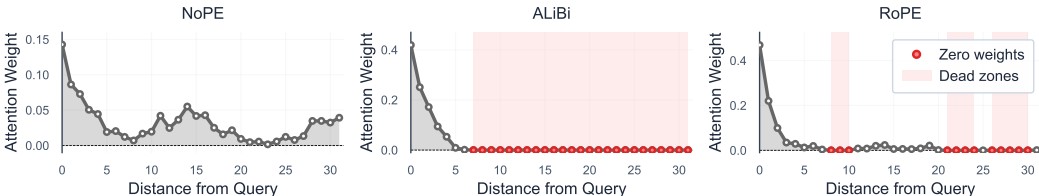

Figure 4: Example of attention weight profiles for different positional encodings with $\alpha$-entmax with $\alpha = 1.5$. NoPE induces content-driven sparsity. ALiBi induces attention windows with a clear cutoff. RoPE promotes frequency-dependent patterns with potential periodic dead zones.

**Robust locality + sparse long-range retrieval.** With that in mind, we propose **NAPE**, where half of the heads induce recency bias via ALiBi, while the other half use NoPE and thus are more content-driven. This division is particularly effective under length extrapolation: ALiBi heads maintain consistent local receptive fields, while NoPE heads allows the model to retrieve distant key-value evidence when needed. Finally, ASEntmax complements NAPE by adapting logit scaling as context grows, preventing the sparsity induced by the $\alpha$-entmax transformation from becoming overly aggressive or overly diffuse at extreme lengths.

## 5 EXPERIMENTS

### 5.1 SYNTHETIC TASKS

A number of works have turned to synthetic tasks as a probing ground for transformers' length-generalization capabilities (Anil et al., 2022; Dziri et al., 2023; Zhou et al., 2024). Such tasks, like copying a sequence and sorting numbers, allow precise control over training and test lengths, revealing whether a model has truly learned an algorithm that scales or merely memorized patterns within a limited length. Vanilla transformers struggle in this setting: they often achieve perfect accuracy on sequences up to the training length, yet fail catastrophically on even slightly longer sequences (Press et al., 2022). To quantitatively evaluate our proposed improvements, we embrace this paradigm of synthetic tasks for long-sequence testing. Concretely, we evaluate our models on a diverse set of synthetic tasks designed to test different aspects of long-context modeling, covering both position-agnostic reasoning (where token positions are not critical) and position-sensitive operations (where relative or absolute positions matter):

- **Retrieval-focused tasks:** These include *Max Retrieval* (Barbero et al., 2024), which requires identifying maximum values in sequences, and *Multi-query Multi-token Associative Recall* (MQMTAR)—a variant of that proposed by Arora et al. (2024), but with multi-token keys and values—which involves matching queries to their corresponding key-value pairs. Both tasks test the model's ability to maintain focus on relevant tokens regardless of their positions in long contexts.

- **Memory-dependent tasks:** We evaluate models on *Copy* (reproducing input sequences). It assesses how well the model preserves token representations and accesses specific positional information throughout the network. On this line, we also evaluate on 2Back, described in §G.

- **Ordering tasks:** This category contains tasks such as *Sort* (arranging tokens in ascending order) and *Reverse* (outputting tokens in reverse order). These evaluate compositional generalization and positional reasoning, becoming increasingly challenging as sequence length grows.

**Experimental Setup.** We use small decoder-only transformers and keep the number of layers as low as possible so that the results reflect the attention method's capabilities rather than model scale. The only exception is *Reverse*, where we raise $L$ until a plain softmax baseline reaches at least $1.5\times$ the in-distribution performance. For positional information we default to NAPE (NoPE+ALiBi), where half the heads have no positional encoding (NoPE) and the other half use ALiBi with linear slopes.[2] We treat NAPE as a practical default, not a contribution, and report RoPE and standalone

---

[2]Hard-ALiBi (Jelassi et al., 2024) allows zero slopes on some heads, which is equivalent to NoPE; our NAPE default mirrors this practical configuration.

Table 1: Exact match accuracy (%) on representative tasks. For each task, we report in-distribution sequence length $n$ in the first column ($n = 64$ for all tasks), followed by OOD results at increasing sequence lengths. $L$ indicates the number of layers. Best results are in **bold**.

| Method | MQMTAR ($L = 4$) | | | | | | | Reverse ($L = 6$) | | | | |
|---|---|---|---|---|---|---|---|---|---|---|---|---|
| | ID | 2× | 4× | 16× | 64× | 256× | 1024× | ID | 1.5× | 2× | 4× | 8× |
| Softmax | 100.0 | **100.0** | **100.0** | 99.5 | 97.8 | 80.2 | 3.0 | 100.0 | 36.0 | 0.0 | 0.0 | 0.0 |
| Top-K (K=32) | 100.0 | 99.9 | 6.2 | 0.0 | 0.0 | 0.0 | 0.0 | 100.0 | **100.0** | 98.7 | 57.0 | 0.0 |
| SSMax | 99.9 | **100.0** | 99.9 | 99.6 | 98.3 | 90.6 | 26.7 | 100.0 | 54.6 | 0.0 | 0.0 | 0.0 |
| Entmax | 100.0 | **100.0** | **100.0** | 99.2 | 92.7 | 66.8 | 9.3 | 100.0 | 99.0 | 86.0 | 28.5 | 0.2 |
| ASEntmax | 100.0 | **100.0** | 100.0 | **99.7** | **99.6** | **99.0** | **95.3** | 100.0 | **100.0** | **99.8** | **96.4** | **56.7** |

| Method | Copy ($L = 2$) | | | | | | | Sort ($L = 2$) | | | |
|---|---|---|---|---|---|---|---|---|---|---|---|
| | ID | 2× | 4× | 8× | 16× | 32× | 64× | ID | 2× | 4× | 8× | - |
| Softmax | 100.0 | **100.0** | 99.9 | 99.9 | 99.4 | 96.1 | 85.5 | 100.0 | 0.0 | 0.0 | 0.0 | - |
| Top-K (K=32) | 100.0 | 99.7 | 96.8 | 26.7 | 0.0 | 0.0 | 0.0 | 100.0 | 92.5 | 0.0 | 0.0 | - |
| SSMax | 100.0 | **100.0** | **100.0** | 99.9 | 99.6 | 99.3 | 95.8 | 100.0 | 0.0 | 0.0 | 0.0 | - |
| Entmax | 100.0 | 99.0 | 86.0 | 28.5 | 0.2 | 0.0 | 0.0 | 100.0 | 99.3 | 57.8 | 0.0 | - |
| ASEntmax | 100.0 | **100.0** | 99.9 | 99.7 | 99.4 | 96.3 | 86.6 | 100.0 | **100.0** | **79.7** | 0.0 | - |

ALiBi in App. §H for completeness.[3] We employ $\alpha = 1.5$ for both Entmax and ASEntmax models, and $\delta = 1$ for SSMax and ASEntmax. Further hyperparameters and ablations appear in App. §G.

**Discussion.** The results, shown in Table 1, reveal a critical factor for length generalization: attention sparsity. Specifically, ASEntmax dramatically outperforms others at extreme lengths—maintaining 96.4% accuracy at 256× test length on MQMTAR (vs. 80.2% for softmax) and 96.4% at 4× on Reverse (vs. 0% for softmax). Moreover, the consistent superiority of ASEntmax over basic $\alpha$-entmax confirms the benefits of adaptive scaling, particularly at extreme lengths where fixed-$\alpha$ may become too sparse or too diffuse. SSMax performs well on the Copy task, even outperforming other methods, but struggles on more complex tasks like MQMTAR and Reverse at extreme lengths. This indicates that while scaling logits helps maintain peak attention magnitude, the explicit sparsity of $\alpha$-entmax provides additional benefits by completely removing irrelevant connections. These findings are further supported by results on the Max Retrieval task (Figure 1 right), where sparse attention mechanisms demonstrate superior length extrapolation compared to dense approaches, with ASEntmax maintaining over 60% accuracy even at 4096-length sequences—a dramatic improvement over standard Softmax and Adaptive Temperature (Veličković et al., 2025). Finally, Copy and Reverse show moderate generalization (up to 64× and 8× respectively), while Sort fails beyond 4× length for all methods. This pattern suggests that tasks requiring precise global ordering (Sort, Reverse) are inherently more challenging for length generalization than tasks dependent on local or independent token properties. We provide per-task results, including results for other positional encoding methods such as RoPE, ALiBi, and NoPE, in §H.

## 5.2 LANGUAGE MODELING

To validate our approach on real-world tasks, we train 420M-parameter decoder-only models following the LLaMA 3 architecture (details in App. G.3) on the high-quality DCLM-Edu dataset (Allal et al., 2025), for 7B tokens with a context length of $n = 2048$. As in the synthetic experiments, we report results in the main paper using NAPE (NoPE+ALiBi). Results with RoPE are reported in App. H.9. We evaluate short-context performance on Lambada, HellaSwag, PIQA, ARC-C, Winogrande, and OpenBookQA, and assess long-context generalization via perplexity on ArXiv/PubMed subsets of The Pile (Gao et al., 2020) as well as RULER's Needle-in-a-Haystack tasks (Hsieh et al., 2024).

**Discussion.** In short-context evaluation (Table 2), all models perform comparably. We note that performance on ARC-C, Winogrande, and OpenBookQA is near-random. Among the other benchmarks,

---

[3]Across all attention methods tested—including softmax—NAPE consistently performs best. We study its interaction with $\alpha$-entmax and positional encodings in §E, with further empirical insights in §E.4 and §H.9.

Table 2: Downstream-task results on short-context datasets. Best results are in **bold**.

| Method | Lambada (PPL) | Lambada | Hellaswag | PIQA | Arc-C | WinoGrande | OpenbookQA |
|---|---|---|---|---|---|---|---|
| Softmax | 52.4 | 30.9 | 33.1 | **65.1** | 25.6 | 49.5 | 28.2 |
| SSMax | 48.9 | 31.6 | 32.9 | **65.1** | 25.0 | **51.5** | **30.4** |
| Entmax | 47.9 | 32.1 | 32.8 | 63.6 | 24.6 | 50.9 | 29.0 |
| ASEntmax | **41.6** | **34.3** | **33.4** | 63.8 | **26.0** | 50.0 | 28.6 |

Table 3: Long-context perplexity on ArXiv and PubMed from The Pile. Best results are in **bold**.

| Model | ArXiv (ID) | | ArXiv (OOD) | | | PubMed (ID) | | PubMed (OOD) | | |
|---|---|---|---|---|---|---|---|---|---|---|
| | 1K | 2K | 4K | 8K | 16K | 1K | 2K | 4K | 8K | 16K |
| Softmax | 21.63 | 17.10 | 13.87 | 12.46 | 12.71 | 19.54 | 18.05 | 15.59 | 15.31 | 18.23 |
| SSMax | 21.57 | 17.02 | 13.74 | 12.29 | 12.31 | 19.35 | 17.84 | 15.14 | 13.75 | 14.72 |
| Entmax | 21.36 | **16.85** | 13.36 | 11.04 | 10.07 | 19.10 | **17.64** | 14.79 | 12.86 | 13.02 |
| ASEntmax | **21.30** | 16.86 | **13.31** | **10.89** | **10.01** | **19.04** | 17.69 | **14.76** | **12.61** | **12.90** |

ASEntmax achieves the highest scores on Lambada (both in terms of perplexity and accuracy) and HellaSwag, while Softmax and SSMax perform best on PIQA. For long-context modeling (Table 3), ASEntmax outperforms all other models. On ArXiv, it shows strong extrapolation, maintaining a decreasing perplexity trend even when extended to $8\times$ the pre-trained sequence length. On PubMed, although all models struggle with $8\times$ extrapolation, ASEntmax still leads by a margin of about 1 perplexity point vs SSMax. In retrieval tasks from RULER (Table 4), entmax-based models perform better overall, with both Entmax and ASEntmax surpassing their softmax counterparts at the extrapolation $4\times$ and $8\times$. ASEntmax in particular shows strong extrapolation in passkey retrieval on the simple haystack task (S-NIAH-1), maintaining near-perfect performance even at $8\times$ context length. On the harder S-NIAH-2 variant, despite performance dropping substantially across all models, ASEntmax remains the best performer at $2\times$ and $4\times$ length generalization.

## 6 RELATED WORKS

**Attention Dispersion.** Recent work has identified attention dispersion as a fundamental limitation in softmax-based transformers (Dong et al., 2021; Zhai et al., 2023; Veličković et al., 2025). For example, Veličković et al. (2025) demonstrate that softmax attention inevitably disperses focus as sequence length increases, while Nakanishi (2025) propose SSMax to scale attention logits based on sequence length. Our approach employs $\alpha$-entmax (Peters et al., 2019), which naturally produces sparse distributions by assigning exactly zero probability to irrelevant tokens. We provide theoretical guarantees that $\alpha$-entmax maintains bounded normalized entropy as sequence length increases—a property softmax fundamentally lacks. Our ASEntmax further improves $\alpha$-entmax with learnable, context-dependent scaling, leading to consistent gains over SSMax across diverse tasks.

**Representational Collapse and Over-Squashing.** Studies analyzing attention patterns in neural networks have noted that increasing depth and context length can induce representational degeneration (Dong et al., 2021; Noci et al., 2022; Arroyo et al., 2025). In particular, Barbero et al. (2024) prove that with softmax attention, token representations become indistinguishable as sequence length increases and gradient paths grow as $\mathcal{O}(n^L)$, causing exponential signal dilution. Our analysis shows, theoretically and empirically, that $\alpha$-entmax can address both limitations by maintaining distinct token representations and by reducing gradient paths to $\mathcal{O}(s^L)$ for increasing sequence lengths $n$.

**Positional Encodings.** The design of positional encodings plays a central role in enabling transformers to generalize to long contexts. ALiBi (Press et al., 2022) introduced linear attention biases with fixed slopes that encourage recency, and has since inspired several extensions that either mitigate softmax-related issues (e.g., Hard-ALiBi (Jelassi et al., 2024)) or induce various learned decay patterns (KERPLE (Chi et al., 2022), FIRE (Li et al., 2024)). More recently, Stick-Breaking Attention (Tan et al., 2025) can be interpreted as a dynamic variant of ALiBi, in which the slopes are input-dependent and calculated adaptively from spans of tokens. Our work incorporates NAPE (NoPE +

Table 4: Retrieval performance on RULER benchmark. All models use NAPE positional encoding and were trained on 2048-token contexts. Best results are in **bold**.

| Model | S-NIAH-1 (ID) | | S-NIAH-1 (OOD) | | | S-NIAH-2 (ID) | | S-NIAH-2 (OOD) | |
|---|---|---|---|---|---|---|---|---|---|
| | 1K | 2K | 4K | 8K | 16K | 1K | 2K | 4K | 8K |
| Softmax | **100.0** | 99.4 | 94.2 | 11.4 | 0.8 | **100.0** | **100.0** | 4.8 | 0.0 |
| SSMax | **100.0** | 99.8 | 99.2 | 92.0 | 75.2 | 99.4 | 99.2 | 64.4 | 14.8 |
| Entmax | 99.8 | 99.8 | 89.0 | 21.6 | 1.2 | 99.6 | 99.4 | 64.8 | 7.2 |
| ASEntmax | 99.6 | **100.0** | **100.0** | **99.8** | **97.4** | 99.4 | 99.4 | **83.2** | **25.4** |

ALiBi). In this formulation, ALiBi heads combined with $\alpha$-entmax create hard attention windows similar to Hard-ALiBi (Jelassi et al., 2024). NoPE, in turn, can learn positional bias (see App. H.1), and when combined with ASEntmax, it induces a learnable, input-dependent recency bias.

**Attention Scaling.** Recent work has shown that scaling attention logits is key for maintaining sharp attention in long contexts. Methods like YaRN (Peng et al., 2024) and the entropy-aware approach of Zhang et al. (2024) use dynamic logit scaling—often with modified RoPE—to stabilize attention during training on extended contexts. Scalable-Softmax (Nakanishi, 2025) and SWAN (Puvvada et al., 2025) apply a $\log n$-based logit scaling to control dispersion without requiring post-training adaptation. InfoScale (Li et al., 2025a) derive scaling rules from the principle of entropy invariance, while Scale-invariant Attention (Anson et al., 2025) introduces position-dependent transformations to balance attention across both local and global contexts. Across these methods, adaptive scaling consistently improves extrapolation to longer sequences. Our ASEntmax builds on this line of work by introducing context-dependent, learnable scaling within the $\alpha$-entmax framework, enabling sparse, focused attention as context length increases.

**Sparse Attention.** Previous sparse attention approaches include structured patterns like Longformer (Beltagy et al., 2020) and BigBird (Zaheer et al., 2020), as well as adaptive methods like $\alpha$-entmax (Peters et al., 2019), top-$k$ related methods (Gupta et al., 2021; Zeng et al., 2025), and chunk-based approaches (Mohtashami & Jaggi, 2023; Hu et al., 2025). A complementary line of work targets efficient long-context *inference* via structured or dynamic sparsification, e.g., attention sinks (Xiao et al., 2024), dynamic sparse prefill accelerators (Jiang et al., 2024; Lai et al., 2025; Li et al., 2025b), and block/permutation sparse schemes (Chen et al., 2025; Xu et al., 2025; Zhang et al., 2025). In contrast to many inference-time sparsification strategies, we study a more general modification—the choice of the *attention transformation*—and analyze how it affects length generalization through dispersion, representational collapse, and over-squashing. Finally, we note that ASEntmax employs AdaSplash (Gonçalves et al., 2025) for computing $\alpha$-entmax attention efficiently and thus inherits its runtime properties. Hence, these inference-oriented sparsification methods are largely orthogonal to ASEntmax and can be combined for speeding up inference while retaining theoretical benefits.

## 7 CONCLUSIONS

In this paper, we present a principled approach to long-context modeling by replacing softmax with $\alpha$-entmax in transformer attention. Our theoretical analysis demonstrates how this simple change addresses three fundamental limitations: it avoids attention dispersion through naturally sparse distributions, prevents representational collapse by maintaining distinct token representations, and alleviates over-squashing by reducing gradient paths from $\mathcal{O}(n^L)$ to $\mathcal{O}(s^L)$, where $s \ll n$ is the number of tokens with nonzero probability. We further introduce Adaptive-Scalable $\alpha$-entmax (ASEntmax), which adaptively adjusts sparsity based on sequence length for each attention head and query input. Our empirical results confirm these theoretical predictions across both synthetic and real-world tasks. On synthetic benchmarks, ASEntmax achieves 95.3% accuracy on associative recall at $1000\times$ the training length, substantially outperforming softmax and existing alternatives. On language modeling with 420M-parameter models, ASEntmax maintains decreasing perplexity at $8\times$ the training context length and achieves 97.4% retrieval accuracy at 16K tokens after training on only 2K tokens. These findings suggest that addressing the fundamental mathematical limitations of transformer attention mechanisms provides a direct path to robust long-context generalization.

## REPRODUCIBILITY STATEMENT

To facilitate the reproducibility of our results, we make our code publicly available at `https://github.com/deep-spin/asentmax`. For efficient softmax attention, we have used FlashAttention-2 (Dao, 2024), and for $\alpha$-entmax we relied on AdaSplash (Gonçalves et al., 2025). All theoretical results presented in §3 are accompanied by complete proofs in Appendix §C and §D. For our synthetic task experiments, we provide detailed task descriptions, data generation procedures, model architectures, and hyperparameters in Appendix §G.1-G.2. Our language modeling experiments use publicly available datasets: DCLM-Edu (Allal et al., 2025) for training and standard benchmarks (LAMBADA, HellaSwag, PIQA, ARC-C, Winogrande, OpenBookQA), The Pile (Gao et al., 2020) (ArXiv, PubMed), and RULER (Hsieh et al., 2024) for evaluation, with complete model specifications and training details provided in Appendix §G.3.

## ACKNOWLEDGMENTS

We thank the SARDINE Lab members for reviewing this paper and providing helpful feedback. This work was supported by the Portuguese Recovery and Resilience Plan through project C645008882-00000055 (Center for ResponsibleAI), by the project DECOLLAGE (ERC-2022-CoG 101088763), and by FCT/MECI through national funds and when applicable co-funded EU funds under UID/50008: Instituto de Telecomunicações.

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

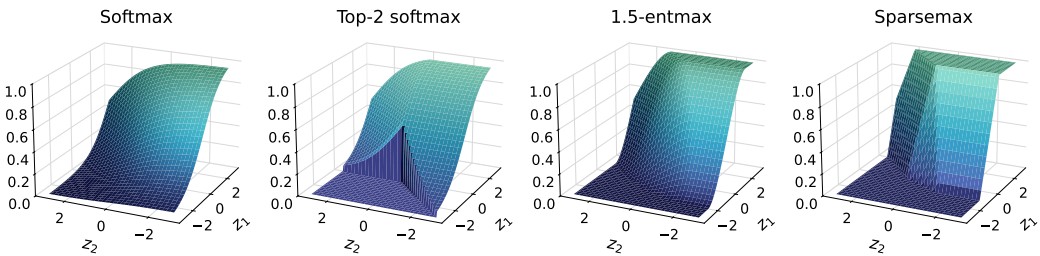

Figure 5: Visualization of $\alpha$-entmax for different values of $\alpha$. We also include top-$k$ softmax with $k = 2$ for completeness. Each panel shows how $p_0$ varies for the input $\boldsymbol{z} = [0, z_1, z_2]$.

## A  $\alpha$-ENTMAX TRANSFORMATION

The $\alpha$-entmax transformation of a score vector $\boldsymbol{z} \in \mathbb{R}^n$ is defined as follows (Peters et al., 2019):

$$\alpha\text{-entmax}(\boldsymbol{z}) := \arg \max_{\boldsymbol{p} \in \triangle_n} \boldsymbol{p}^\top \boldsymbol{z} + H_\alpha(\boldsymbol{p}), \quad \triangle_n := \{\boldsymbol{p} \in \mathbb{R}^n : \boldsymbol{p} \geq \boldsymbol{0}, \boldsymbol{1}^\top \boldsymbol{p} = 1\}, \quad (11)$$

where $H_\alpha(\boldsymbol{p})$ is the Tsallis($\alpha$) entropy (Tsallis, 1988), defined as

$$H_\alpha(\boldsymbol{p}) := \begin{cases} \frac{1}{\alpha(\alpha-1)} \sum_j (p_j - p_j^\alpha), & \alpha \neq 1 \\ -\sum_j p_j \log p_j, & \alpha = 1. \end{cases} \quad (12)$$

Solving the optimization problem above corresponds to finding a threshold $\tau$ so that $\boldsymbol{p}^\star$ sums to 1. Given $\tau$, we can easily evaluate $\alpha$-entmax as follows:

$$\alpha\text{-entmax}(\boldsymbol{z}) = [(\alpha - 1)\boldsymbol{z} - \tau\boldsymbol{1}]_+^{\frac{1}{\alpha-1}}, \quad (13)$$

where $[\cdot]_+$ is the ReLU function. Figure 5 illustrates how $\alpha$-entmax behaves for different choices of $\alpha$. From the equation above, it is clear that coordinates with $(\alpha - 1)z_i \leq \tau$ become exactly zero. In other words, $\alpha$-entmax yields dynamic sparsity, where the pattern of zeros depends on the input, with $\alpha$ controlling the propensity of sparsity: larger values leads to increasingly sparser outputs.

**Relation to softmax and sparsemax.**  The $\alpha$-entmax family continuously interpolates between softmax and sparsemax. When $\alpha \to 1^+$, the Tsallis entropy $H_\alpha$ converges to Shannon entropy and $\alpha$-entmax converges to the usual softmax mapping. At $\alpha = 2$, we have $H_2(\boldsymbol{p}) \propto \|\boldsymbol{p}\|_2^2$ (Gini entropy), and the resulting optimizer coincides with *sparsemax* (Martins & Astudillo, 2016).

**$q$-logarithm and $q$-exponential view.**  A useful way to interpret $\alpha$-entmax is through $q$-deformed exponentials (Tsallis, 1988). For $q \neq 1$, define the $q$-exponential

$$\exp_q(x) := \left[1 + (1 - q)x\right]_+^{\frac{1}{1-q}}, \quad (14)$$

with $\exp_1(x) = \exp(x)$ obtained by continuity. Let $q := 2 - \alpha$ so that $1 - q = \alpha - 1$ and $\frac{1}{1-q} = \frac{1}{\alpha-1}$. Then the closed form of $\alpha$-entmax can be written as a normalized $q$-exponential of a shifted score:

$$\alpha\text{-entmax}(\boldsymbol{z})_i \propto \exp_{2-\alpha}(z_i - c) \quad \text{for a scalar } c \text{ chosen so that } \sum_i \alpha\text{-entmax}(\boldsymbol{z})_i = 1. \quad (15)$$

This makes the connection with softmax explicit as softmax corresponds to the $q \to 1$ limit (no truncation), while for $\alpha > 1$ (i.e., $q < 1$) the ReLU operation yields exact zeros (sparsity).

**$\alpha$-entmax computation.**  Order-based algorithms have only been proposed for $\alpha = 2$ (Martins & Astudillo, 2016) and $\alpha = 1.5$ (Peters et al., 2019). In contrast, root-finding algorithms apply for all values of $\alpha$. Letting $\max(\boldsymbol{z}) = 1$ and following Peters et al. (2019), we have

$$0 \leq \tau^\star \leq 1 - n^{1-\alpha}. \quad (16)$$

Blondel et al. (2020) propose a bisection or binary search approach to finding $\tau^\star$. Similar in spirit, Gonçalves et al. (2025) introduces a GPU-oriented solver of $\alpha$-entmax that uses a hybrid *Halley-bisection* method, which combines the fast local convergence of higher-order root-finding with the convergence guarantees of bisection.

## B    MODEL DEFINITION AND NOTATION

In this work, we study (causal) transformers with sparse attention distributions created by replacing softmax with $\alpha$-entmax. We present the precise mathematical formulation of our model below, following closely the notation from (Barbero et al., 2024).

Let $\boldsymbol{Q} = \boldsymbol{X}\boldsymbol{W}_Q, \boldsymbol{K} = \boldsymbol{X}\boldsymbol{W}_K, \boldsymbol{V} = \boldsymbol{X}\boldsymbol{W}_V \in \mathbb{R}^{n \times d}$ be the query, key, and value projections of the input embeddings respectively, where $n$ is sequence length and $d$ the hidden size. We denote with $\boldsymbol{q}_i, \boldsymbol{k}_i, \boldsymbol{v}_i \in \mathbb{R}^d$ the $d$-dimensional query, key, and value vectors of the $i$-th token. For a single attention head, transformers compute the representation of the $i$-th token through the following layer-wise transformations:[4]

$$\boldsymbol{u}_i^{(\ell)} = \sum_{j \leq i} p_{ij}^{(\ell)} \text{norm}_1^{(\ell)}\left(\boldsymbol{v}_j^{(\ell-1)}\right) + \boldsymbol{v}_i^{(\ell-1)}, \tag{17}$$

$$\boldsymbol{v}_i^{(\ell)} = \text{FFN}^{(\ell)}\left(\text{norm}_2^{(\ell)}\left(\boldsymbol{u}_i^{(\ell)}\right)\right) + \boldsymbol{u}_i^{(\ell)}, \tag{18}$$

$$\boldsymbol{y}_i = \text{norm}_3\left(\boldsymbol{v}_i^{(L)}\right), \tag{19}$$

where $\ell$ is the later index, $p_{ij}^{(\ell)}$ represents the attention weights, $\text{FFN}^{(\ell)} : \mathbb{R}^d \rightarrow \mathbb{R}^d$ represents the feed-forward network, and $\text{norm}_1^{(\ell)}, \text{norm}_2^{(\ell)}$, and $\text{norm}_3$ are normalization functions. The final representation $\boldsymbol{y}_i$ is computed after applying $L$ transformer layers. For next-token prediction tasks, the model output typically depends solely on $\boldsymbol{y}_n$, the final representation of the last token. The attention weights $p_{ij}^{(\ell)}$ are computed by applying a transformation $\pi : \mathbb{R}^n \rightarrow \triangle_n$ as follows:

$$p_{ij}^{(\ell)} = \pi\left(\boldsymbol{z}_i^{(\ell)}\right)_j, \tag{20}$$

where $\boldsymbol{z}_i^{(\ell)} \in \mathbb{R}^n$ is the vector of logits for token $i$ at layer $\ell$, with elements $z_{ij}^{(\ell)} = \langle \boldsymbol{q}_i^{(\ell)}, \boldsymbol{k}_j^{(\ell)} \rangle / \sqrt{d}$. The function $\pi$ maps these logits to a probability distribution over the $n$ tokens, with $\triangle_n$ denoting the probability simplex. In standard transformers, $\pi$ is the softmax function:

$$\text{softmax}(\boldsymbol{z})_j = \frac{\exp(z_j)}{\sum_{k \leq i} \exp(z_k)}. \tag{21}$$

In our approach, we replace softmax with $\alpha$-entmax (§A). We group the attention weights into an attention matrix at the $\ell$-th layer, defined element-wise as $[\boldsymbol{P}^{(\ell)}]_{ij} := p_{ij}^{(\ell)}$. This is a row-stochastic lower triangular matrix that can also be interpreted as a probabilistic directed graph. Finally, when incorporating positional information, we modify the attention logits computation according to the chosen positional encoding strategy:

- **NoPE**: $z_{ij}^{(\ell)} = \langle \boldsymbol{q}_i^{(\ell)}, \boldsymbol{k}_j^{(\ell)} \rangle / \sqrt{d}$.

- **ALiBi**: $z_{ij}^{(\ell)} = \langle \boldsymbol{q}_i^{(\ell)}, \boldsymbol{k}_j^{(\ell)} \rangle / \sqrt{d} + m \cdot (j - i)$, where $m \in \mathbb{R}$ is a slope hyperparameter.

- **RoPE**: $z_{ij}^{(\ell)} = (\boldsymbol{q}_i^{(\ell)})^\top \boldsymbol{R}^{j-i} \boldsymbol{k}_j^{(\ell)}$, where $\boldsymbol{R} \in \mathbb{R}^{d \times d}$ is a block-diagonal rotation matrix.

## C    NON-DISPERSION OF $\alpha$-ENTMAX

A critical problem in long-context modeling is the dispersion of attention, where relevant signals get diluted across increasingly long sequences. For clarity, we begin by examining how $\alpha$-entmax behaves with two-level logits, and then proceed to define dispersion more rigorously and how $\alpha$-entmax naturally counteracts this issue.

**Lemma 2** (Threshold Behavior for Two-level Logits). *Consider logits $\boldsymbol{z} \in \mathbb{R}^n$ where $k$ tokens have value $M$ and $(n - k)$ tokens have value $m$ with $M > m$.*

---

[4]Following Barbero et al. (2024), we omit the linear projections used to compute the vectors from the output of previous layers for clarity; however, this does not impact our derivations and conclusions.

1. *For $\alpha > 1$, when $\Delta := M - m \geq \frac{k^{-(\alpha-1)}}{\alpha-1}$, only the $k$ tokens with value $M$ receive non-zero attention. The threshold converges to $\tau(\boldsymbol{z}) = (\alpha-1)M - k^{-(\alpha-1)}$, and each high-value token receives attention $\frac{1}{k}$ while others receive zero attention. As a consequence, $\alpha$-entmax maintains a constant attention weight of $\Theta(\frac{1}{k})$ on high-value tokens regardless of the total sequence length $n$.*

2. *In contrast, softmax (with fixed temperature $\theta > 0$) necessarily disperses with attention weights of $\Theta(\frac{1}{n})$ as $n$ increases. For softmax to maintain concentration of at least $c \in (0, 1)$ on the $k$ high-value tokens, the required logit difference must grow logarithmically with $n$:*

$$\frac{\Delta}{\theta} \geq \ln\left(\frac{n-k}{k} \cdot \frac{c}{1-c}\right) \tag{22}$$

*Proof.* We prove the two parts below.

**Part (i):** Let $\mathcal{S}$ be the support set. A token $i$ is in $\mathcal{S}$ if and only if $z_i > \frac{\tau(\boldsymbol{z})}{\alpha-1}$ where $\tau(\boldsymbol{z})$ satisfies:

$$\sum_{i=1}^{n} \left[(\alpha-1)z_i - \tau(\boldsymbol{z})\right]_+^{\frac{1}{\alpha-1}} = 1. \tag{23}$$

For our two-level distribution, this becomes:

$$\sum_{i:z_i=M} \left[(\alpha-1)M - \tau(\boldsymbol{z})\right]_+^{\frac{1}{\alpha-1}} + \sum_{i:z_i=m} \left[(\alpha-1)m - \tau(\boldsymbol{z})\right]_+^{\frac{1}{\alpha-1}} = 1. \tag{24}$$

For only tokens with value $M$ to receive non-zero attention, we need:

$$(\alpha-1)M - \tau(\boldsymbol{z}) > 0 \quad \text{and} \quad (\alpha-1)m - \tau(\boldsymbol{z}) \leq 0. \tag{25}$$

Rearranging: $(\alpha-1)m \leq \tau(\boldsymbol{z}) < (\alpha-1)M$. In this regime:

$$k \cdot \left[(\alpha-1)M - \tau(\boldsymbol{z})\right]^{\frac{1}{\alpha-1}} = 1. \tag{26}$$

Solving for $\tau(\boldsymbol{z})$:

$$\left[(\alpha-1)M - \tau(\boldsymbol{z})\right]^{\frac{1}{\alpha-1}} = \frac{1}{k} \tag{27}$$

$$(\alpha-1)M - \tau(\boldsymbol{z}) = k^{-(\alpha-1)} \tag{28}$$

$$\tau(\boldsymbol{z}) = (\alpha-1)M - k^{-(\alpha-1)}. \tag{29}$$

For this threshold to satisfy $\tau(\boldsymbol{z}) \geq (\alpha-1)m$, we need:

$$(\alpha-1)M - k^{-(\alpha-1)} \geq (\alpha-1)m \tag{30}$$

$$M - m \geq \frac{k^{-(\alpha-1)}}{\alpha-1} \tag{31}$$

$$\Delta \geq \frac{k^{-(\alpha-1)}}{\alpha-1}. \tag{32}$$

Thus, when $\Delta \geq \frac{k^{-(\alpha-1)}}{\alpha-1}$, only the $k$ tokens with value $M$ receive non-zero attention, each receiving an attention of $\frac{1}{k}$. Therefore, when $\Delta \geq \frac{k^{-(\alpha-1)}}{\alpha-1}$, the attention weights for $\alpha$-entmax are:

$$\alpha\text{-entmax}(\boldsymbol{z})_i = \begin{cases} \frac{1}{k} & \text{if } z_i = M \\ 0 & \text{if } z_i = m. \end{cases} \tag{33}$$

These weights remain $\Theta(\frac{1}{k})$ for high-value tokens regardless of $n$, demonstrating that $\alpha$-entmax can maintain constant attention on important tokens even as sequence length grows, as long as $k$ is fixed as $n \to \infty$.

**Part (ii):** For softmax with temperature $\theta > 0$, the attention weight for tokens with logit $M$ is:

$$\text{softmax}(\boldsymbol{z}/\theta)_i = \frac{\exp(M/\theta)}{k \exp(M/\theta) + (n-k) \exp(m/\theta)}. \tag{34}$$

For softmax to maintain concentration of at least $c$ on the $k$ high-value tokens combined:

$$\frac{k \exp(M/\theta)}{k \exp(M/\theta) + (n-k) \exp(m/\theta)} \geq c. \tag{35}$$

Through algebraic manipulation:

$$k \exp(M/\theta) \geq c \left[ k \exp(M/\theta) + (n-k) \exp(m/\theta) \right] \tag{36}$$

$$(1-c)k \exp(M/\theta) \geq c(n-k) \exp(m/\theta) \tag{37}$$

$$\frac{k \exp(M/\theta)}{(n-k) \exp(m/\theta)} \geq \frac{c}{1-c} \tag{38}$$

$$\frac{k}{n-k} \exp(\Delta/\theta) \geq \frac{c}{1-c} \tag{39}$$

$$\exp(\Delta/\theta) \geq \frac{n-k}{k} \cdot \frac{c}{1-c}. \tag{40}$$

Taking the natural logarithm:

$$\frac{\Delta}{\theta} \geq \ln\left(\frac{n-k}{k} \cdot \frac{c}{1-c}\right). \tag{41}$$

This shows that as $n$ grows, the required $\Delta$ for maintaining concentration with softmax grows logarithmically with $n$. In contrast, for $\alpha$-entmax, assuming we have a $k$ that is fixed as $n$ grows, the condition $\Delta \geq \frac{k^{-(\alpha-1)}}{\alpha-1}$ is independent of $n$, enabling constant focus regardless of sequence length. $\qquad\square$

We now prove Proposition 1, which concerns the concept of dispersion presented in Definition 1.

*Proof.* We address each claim in turn. For bounded sequences $(z_n)_{n\in\mathbb{N}}$, we assume $m, M \in \mathbb{R}$ with $m \leq M$ such that $m \leq z_i \leq M$ for every $i \in \mathbb{N}$.

**Part (i) - $\alpha$-entmax can retain probability, while softmax always leaks:** For $\alpha > 1$, consider logits $\boldsymbol{z} \in \mathbb{R}^n$ and an extended sequence $\boldsymbol{z}^* \in \mathbb{R}^N$ with $N > n$, where all additional elements have values below the threshold, $z_i^* \leq \tau(\boldsymbol{z})/(\alpha-1)$ for $i > n$. By the non-vanishing attention property of $\alpha$-entmax (Lemma 1), these additional elements receive exactly zero probability, resulting in:

$$\alpha\text{-entmax}(\boldsymbol{z})_i = \alpha\text{-entmax}(\boldsymbol{z}^*)_i \quad \forall i \leq n. \tag{42}$$

This demonstrates that $\alpha$-entmax can produce identical distributions despite arbitrarily different sequence lengths, maintaining the same concentration regardless of whether we have a distinct number of tokens. In contrast, for softmax ($\alpha = 1$), Barbero et al. (2024) proved that adding any element to the sequence strictly decreases the probability assigned to existing elements, making such invariance impossible.

**Part (ii) - Complete dispersion of softmax:** For softmax with constant temperature $\theta > 0$, the attention weights for bounded logits can be bounded as:

$$\frac{\exp(m/\theta)}{\sum_{j=1}^n \exp(z_j/\theta)} \leq \text{softmax}(\boldsymbol{z}_{1:n}/\theta)_i \leq \frac{\exp(M/\theta)}{\sum_{j=1}^n \exp(z_j/\theta)}. \tag{43}$$

Since $\sum_{j=1}^n \exp(z_j/\theta) \geq n \cdot \exp(m/\theta)$ and $\sum_{j=1}^n \exp(z_j/\theta) \leq n \cdot \exp(M/\theta)$, we have:

$$\frac{\exp(m/\theta)}{n \cdot \exp(M/\theta)} \leq \text{softmax}(\boldsymbol{z}_{1:n}/\theta)_i \leq \frac{\exp(M/\theta)}{n \cdot \exp(m/\theta)}. \tag{44}$$

This simplifies to:

$$\frac{1}{n} \exp\left(-\frac{\Delta}{\theta}\right) \leq \text{softmax}(\boldsymbol{z}_{1:n}/\theta)_i \leq \frac{1}{n} \exp\left(\frac{\Delta}{\theta}\right), \tag{45}$$

where $\Delta = M - m$ is bounded.

These bounds show that as $n \to \infty$, all softmax weights are $\Theta(1/n)$. For the entropy:

$$H(\text{softmax}(\boldsymbol{z}_{1:n}/\theta)) = -\sum_{i=1}^{n} \text{softmax}(\boldsymbol{z}_{1:n}/\theta)_i \log \text{softmax}(\boldsymbol{z}_{1:n}/\theta)_i \to \log n. \tag{46}$$

Thus, $\lim_{n\to\infty} \frac{H(\text{softmax}(\boldsymbol{z}/\theta))}{\log n} = 1$, showing complete dispersion.

**Part (iii) - Strong concentration resilience of $\alpha$-entmax:** First, we focus on the two-level case from Lemma 2, where $k$ tokens have logit value $M$ and $(n-k)$ tokens have value $m$. When $\Delta = M - m \geq \frac{k^{-(\alpha-1)}}{\alpha-1}$, only the $k$ tokens with value $M$ receive non-zero attention:

$$\alpha\text{-entmax}(\boldsymbol{z}_{1:n})_i = \begin{cases} \frac{1}{k} & \text{if } z_i = M \\ 0 & \text{if } z_i = m. \end{cases} \tag{47}$$

The Shannon entropy of this distribution is:

$$H(\alpha\text{-entmax}(\boldsymbol{z}_{1:n})) = -\sum_{i=1}^{k} \frac{1}{k} \log \frac{1}{k} = \log k. \tag{48}$$

The normalized entropy is:

$$\frac{H(\alpha\text{-entmax}(\boldsymbol{z}_{1:n}))}{\log n} = \frac{\log k}{\log n}. \tag{49}$$

For fixed $k$ as $n \to \infty$, this ratio approaches 0, confirming concentration resilience.

For cases where the support grows sublinearly as $k := |\mathcal{S}| = \mathcal{O}(n^\beta)$ for some $\beta < 1$, the Shannon entropy is bounded by:

$$H(\alpha\text{-entmax}(\boldsymbol{z}_{1:n})) \leq \log k = \mathcal{O}(\log n^\beta) = \mathcal{O}(\beta \log n). \tag{50}$$

The normalized entropy is therefore:

$$\lim_{n\to\infty} \frac{H(\alpha\text{-entmax}(\boldsymbol{z}_{1:n}))}{\log n} \leq \beta < 1. \tag{51}$$

This confirms that the normalized entropy remains strictly bounded away from 1, even with growing support, as long as the growth is sublinear.

$\square$

This proposition shows that the entropy of attention distributions reveals how concentrated or dispersed they are across tokens. While softmax distributions with bounded logits must approach maximum entropy $\mathcal{O}(\log n)$ as sequence length increases (indicating complete dispersion), $\alpha$-entmax distributions can maintain bounded entropy $\mathcal{O}(\log k)$ where $k$ is the support size. This allows models with $\alpha$-entmax to maintain focused, low-entropy attention patterns even when processing extremely long sequences.

Moreover, this proposition demonstrates that $\alpha$-entmax attention distributions have a remarkable property: they do not necessarily disperse as sequence length increases. They can maintain identical attention patterns regardless of context length. This non-dispersion property means transformers with $\alpha$-entmax can scale to very long contexts without the attention becoming diluted, maintaining their ability to focus on relevant information regardless of how much additional context is present.

# D  REPRESENTATIONAL COLLAPSE AND OVER-SQUASHING

## D.1  PROOF OF LEMMA 1

Adding a new element to the sequence of logits can only redistribute probability mass, so $\alpha\text{-entmax}(\boldsymbol{x})_n \geq \alpha\text{-entmax}(\boldsymbol{x}^*)_{n+1}$ must always hold, with equality iff $\alpha\text{-entmax}(\boldsymbol{x}^*)_n = 0$. Since softmax ($\alpha = 1$) cannot return zeros, we must have a strict inequality for $\alpha = 1$.

For $\alpha > 1$, we need to find the value $b_{\max}$ such that, for any $b \leq b_{\max}$, $\alpha\text{-entmax}(\boldsymbol{x}^*)_n = 0$ holds. From the definition of $\alpha$-entmax equation 2, a token $i$ receives non-zero probability iff $(\alpha - 1)z_i > \tau(\boldsymbol{z})$, where $\tau(\boldsymbol{z})$ is the threshold ensuring the sum of probabilities equals 1. Therefore, for the token $b$ in the extended sequence $\boldsymbol{x}^*$ to receive zero probability (thus not affecting other probabilities), we need:

$$(\alpha - 1)b \leq \tau(\boldsymbol{x}^*). \tag{52}$$

We know that $\tau(\boldsymbol{x}^*) \geq \tau(\boldsymbol{x})$ in general for $\alpha$-entmax, as shown by Peters et al. (2019); Martins et al. (2022, Lemma 3; Proposition 4). Therefore, a sufficient condition is:

$$(\alpha - 1)b \leq \tau(\boldsymbol{x}). \tag{53}$$

Solving for $b$, we get:

$$b \leq \frac{\tau(\boldsymbol{x})}{\alpha - 1}. \tag{54}$$

Thus, we can define $b_{\max} = \frac{\tau(\boldsymbol{x})}{\alpha-1}$. For any $b \leq b_{\max}$, the token at position $n$ in $\boldsymbol{x}^*$ receives zero attention, meaning it doesn't affect the normalization. Therefore, $\tau(\boldsymbol{x}^*) = \tau(\boldsymbol{x})$, which means that the condition equation 54 is both necessary and sufficient, and:

$$\alpha\text{-entmax}(\boldsymbol{x})_n = \left[(\alpha - 1)c - \tau(\boldsymbol{x})\right]_+^{\frac{1}{\alpha-1}} = \left[(\alpha - 1)c - \tau(\boldsymbol{x}^*)\right]_+^{\frac{1}{\alpha-1}} = \alpha\text{-entmax}(\boldsymbol{x}^*)_{n+1}. \tag{55}$$

By choosing different values of $b$ such that $b \leq b_{\max}$, we can control the change in threshold $\tau(\boldsymbol{x}^*)$ and consequently the difference $\alpha\text{-entmax}(\boldsymbol{x})_n - \alpha\text{-entmax}(\boldsymbol{x}^*)_{n+1}$ can be as large as $\alpha\text{-entmax}(\boldsymbol{x})_n$.

## D.2  PROOF OF PROPOSITION 2 FOR REPRESENTATIONAL PRESERVATION

We prove the first part of Proposition 2 by exhibiting the counterexample below, following the synthetic construction introduced by Barbero et al. (2024).

**Proposition 3** (Counterexample to Representational Collapse with $\alpha$-entmax). *Let $\boldsymbol{v} \in \mathbb{R}^{(n-1)\times d}$ be a sequence of embedding vectors, and define:*

$$\boldsymbol{v}^{(0)} = [\boldsymbol{v}, \boldsymbol{v}_a]^\top \in \mathbb{R}^{n\times d}, \quad \boldsymbol{v}^{*(0)} = [\boldsymbol{v}, \boldsymbol{v}_a, \boldsymbol{v}_a]^\top \in \mathbb{R}^{(n+1)\times d}, \tag{56}$$

*where the final token $\boldsymbol{v}_a \in \mathbb{R}^d$ is repeated.*

*For appropriate choice of embeddings and $\alpha > 1$, there exists a constant $c > 0$ independent of $n$ such that:*

$$\|\boldsymbol{v}_n^{(L)} - \boldsymbol{v}_{n+1}^{*(L)}\|_1 \geq c > 0 \tag{57}$$

*after $L$ transformer layers with $\alpha$-entmax attention, demonstrating that representational collapse is not inevitable.*

*In contrast, Barbero et al. (2024) proved that for softmax attention, $\|\boldsymbol{v}_n^{(L)} - \boldsymbol{v}_{n+1}^{*(L)}\|_1 \to 0$ as $n \to \infty$ for any such construction.*

*Proof.* We prove this through explicit construction.

Since $\boldsymbol{v}_{1:n-1}^{(0)} = \boldsymbol{v}_{1:n-1}^{*(0)}$ and both sequences end with $\boldsymbol{v}_a$, the attention logits computed by the final tokens are:

$$\boldsymbol{z}_n^{(1)} = [\boldsymbol{v}_a^\top \boldsymbol{v}_1, \ldots, \boldsymbol{v}_a^\top \boldsymbol{v}_{n-1}, \boldsymbol{v}_a^\top \boldsymbol{v}_a], \tag{58}$$

$$\boldsymbol{z}_{n+1}^{*(1)} = [\boldsymbol{v}_a^\top \boldsymbol{v}_1, \ldots, \boldsymbol{v}_a^\top \boldsymbol{v}_{n-1}, \boldsymbol{v}_a^\top \boldsymbol{v}_a, \boldsymbol{v}_a^\top \boldsymbol{v}_a]. \tag{59}$$

Consider the specific embedding choice where:

- $v_a$ is chosen such that $v_a^\top v_a = \phi$ for some $\phi \in \mathbb{R}$.

- $v_i$ for $i = 1, \ldots, n-1$ are chosen such that $v_a^\top v_i = b$ for some value $b < \phi$.

This construction yields the following attention logits:

$$z_n^{(1)} = [b, b, \ldots, b, \phi], \tag{60}$$

$$z_{n+1}^{*(1)} = [b, b, \ldots, b, \phi, \phi]. \tag{61}$$

**Specific counterexample.** Consider $d = 1$, $\alpha = 2.0$, $\phi = 0.5$, and $b = 0$. We can construct inputs as $v_a = \sqrt{0.5}$ and $v_i = 0$ for $i = 1, \ldots, n-1$. The attention logits are:

$$z_n^{(1)} = [0, \ldots, 0, 0.5], \tag{62}$$

$$z_{n+1}^{*(1)} = [0, 0, \ldots, 0, 0.5, 0.5]. \tag{63}$$

For $\alpha = 2.0$ (sparsemax), the attention distributions are:

$$\text{sparsemax}(z_n^{(1)}) = [p_b, p_b, \ldots, p_b, p_n] \quad \text{(dense)} \tag{64}$$

$$\text{sparsemax}(z_{n+1}^{*(1)}) = [0, 0, \ldots, 0, \frac{1}{2}, \frac{1}{2}] \quad \text{(sparse)} \tag{65}$$

where $0 < p_b < p_n < 1$. As $n \to \infty$, we have $p_n \to p^*$ where:

$$p^* = [(\alpha - 1)(\phi - b)]^{\frac{1}{\alpha - 1}} = [(2 - 1)(0.5 - 0)]^{\frac{1}{2-1}} = (0.5)^1 = 0.5. \tag{66}$$

The average representation is $\bar{v} = \frac{1}{n-1} \sum_{i=1}^{n-1} 0 = 0$, and therefore:

$$\lim_{n \to \infty} \|v_n^{(1)} - v_{n+1}^{*(1)}\|_1 = (1 - p^*)|\bar{v} - v_a| = (1 - 0.5)|0 - \sqrt{0.5}| = 0.5 \times \sqrt{0.5} \approx 0.354. \tag{67}$$

This demonstrates that the $L_1$ difference remains bounded at approximately $c = 0.354$, independent of sequence length $n$. **This specific example already establishes the existence of a counterexample to representational collapse with $\alpha$-entmax**. We now extend this result to prove Proposition 3 for general constructions of $v$ and $v_a$.

**General Construction.** With values of $\phi$ such that $b + 2^{-(\alpha-1)}/(\alpha - 1) \le \phi < b + 1/(\alpha - 1)$, $\alpha$-entmax produces the following distributions:[5]

$$\alpha\text{-entmax}(z_n^{(1)}) = [p_1^{(1)}, p_2^{(1)}, \ldots, p_{n-1}^{(1)}, p_n^{(1)}], \tag{68}$$

$$\alpha\text{-entmax}(z_{n+1}^{*(1)}) = [0, 0, \ldots, 0, \frac{1}{2}, \frac{1}{2}]. \tag{69}$$

In particular, since the first $n-1$ positions share the same representation, we have $p_1^{(1)} = p_2^{(1)} = \ldots = p_{n-1}^{(1)} = (1 - p_n^{(1)})/(n - 1) = p_b^{(1)} > 0$, with $0 < p_n^{(1)} < 1$. This leads to representations:

$$v_n^{(1)} = p_b^{(1)} v_1 + \cdots + p_b^{(1)} v_{n-1} + p_n^{(1)} v_a, \tag{70}$$

$$v_{n+1}^{*(1)} = \frac{1}{2} v_a + \frac{1}{2} v_a = v_a. \tag{71}$$

Let, $\bar{v} := \frac{1}{n-1} \sum_{i=1}^{n-1} v_i$ denote the average of the first block of vectors. Taking the $L_1$-norm of the representations difference:

$$\|v_n^{(1)} - v_{n+1}^{*(1)}\|_1 = \left\| (1 - p_n^{(1)})\bar{v} + p_n^{(1)} v_a - v_a \right\|_1 \tag{72}$$

$$= (1 - p_n^{(1)}) \|\bar{v} - v_a\|_1. \tag{73}$$

We need to show that the above expression does not tend to 0 as $n \to \infty$. To that end, we need (i) $\lim_{n \to \infty} p_n^{(1)} = p^* < 1$, and (ii) $\lim_{n \to \infty} \|\bar{v} - v_a\|_1 = c > 0$.

---

[5]The upper bound ensures a dense output for $z_n^{(1)}$, following Lemma 1 with $\tau = (\alpha - 1)\phi - 1$. The lower bounds ensure a sparse output for $z_{n+1}^{*(1)}$, following Lemma 2 with $k = 2$.

**First condition.** We need to choose parameters so that as $n \to \infty$, the original sequences remains **dense** and the extended sequence is in the **sparse** regime. From our analysis with Lemma 1 and 2, this requires:

$$\frac{2^{-(\alpha-1)}}{\alpha - 1} \leq \phi - b < \frac{1}{\alpha - 1}. \tag{74}$$

From the $\alpha$-entmax definition, we have:

$$(p_b^{(1)})^{\alpha-1} = (\alpha - 1)b - \tau, \tag{75}$$

$$(p_n^{(1)})^{\alpha-1} = (\alpha - 1)\phi - \tau, \tag{76}$$

where $p_b^{(1)}$ is the probability for each $b$ token and $p_n^{(1)}$ is the probability for the $\phi$ token. Subtracting the first equation from the second:

$$(p_n^{(1)})^{\alpha-1} - (p_b^{(1)})^{\alpha-1} = (\alpha - 1)(\phi - b). \tag{77}$$

From the normalization constraint $(n-1)p_b^{(1)} + p_n^{(1)} = 1$:

$$p_b^{(1)} = \frac{1 - p_n^{(1)}}{n - 1}. \tag{78}$$

Substituting:

$$(p_n^{(1)})^{\alpha-1} - \left(\frac{1 - p_n^{(1)}}{n - 1}\right)^{\alpha-1} = (\alpha - 1)(\phi - b). \tag{79}$$

We know from the dense regime that $p_n^{(1)} \to p^*$, with $0 < p^* < 1$. Thus, as $n \to \infty$, the probability $p_n^{(1)}$ satisfies:

$$p_n^{(1)} \to p^* \text{ where } p^* \text{ solves } (p^*)^{\alpha-1} - \lim_{n\to\infty}\left(\frac{1 - p^*}{n - 1}\right)^{\alpha-1} = (\alpha - 1)(\phi - b). \tag{80}$$

Since $\left(\frac{1-p^*}{n-1}\right)^{\alpha-1} \to 0$ as $n \to \infty$, we get:

$$(p^*)^{\alpha-1} = (\alpha - 1)(\phi - b) \tag{81}$$

Therefore:

$$p^* = [(\alpha - 1)(\phi - b)]^{\frac{1}{\alpha-1}} < 1. \tag{82}$$

The inequality holds because $\phi - b < \frac{1}{\alpha-1}$, so $(\alpha - 1)(\phi - b) < 1$. The key insight is that as $n \to \infty$, the small probabilities on the $b$ tokens become negligible, but their sum $(n-1)p_b^{(1)} = 1 - p^*$ remains finite, so each individual $p_b^{(1)} \to 0$.

**Second condition.** Choose the input sequence such that $\bar{v} = \frac{1}{n-1}\sum_{i=1}^{n-1} v_i \neq v_a$, and use the construction:

- $v_a = \sqrt{\phi}e_1$ where $e_1$ is the first standard basis vector.

- $v_i = \frac{b}{\sqrt{\phi}}e_1 + e_2$ for $i = 1, \dots, n - 1$.

Note that this construction satisfies the logit constraints:

$$v_a^T v_i = \sqrt{\phi} \cdot \frac{b}{\sqrt{\phi}} + 0 \cdot 1 = b, \tag{83}$$

$$v_a^T v_a = (\sqrt{\phi})^2 = \phi. \tag{84}$$

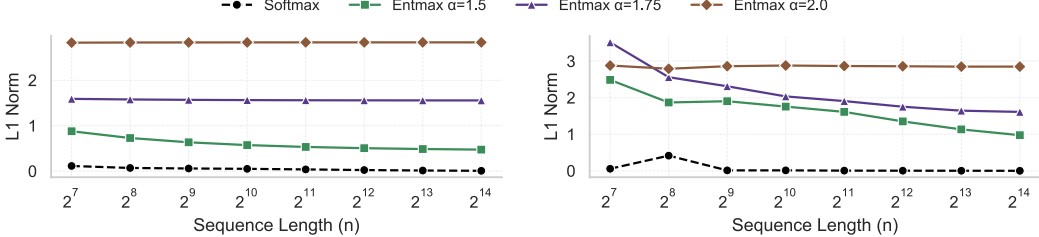

Figure 6: $L_1$ norm of representation difference between original sequence and extended sequence after 6 transformer layers, with constant prefix (left) and random prefix (right). With softmax ($\alpha = 1.0$), representation difference rapidly approaches zero, demonstrating inevitable collapse. $\alpha$-entmax ($\alpha > 1.0$) maintains bounded differences even at extreme sequence lengths.

The average representation is:

$$\bar{\boldsymbol{v}} = \frac{1}{n-1}\sum_{i=1}^{n-1}\boldsymbol{v}_i = \frac{b}{\sqrt{\phi}}\boldsymbol{e}_1 + \boldsymbol{e}_2. \tag{85}$$

Since $\boldsymbol{v}_a = \sqrt{\phi}\boldsymbol{e}_1$, we have:

$$\|\bar{\boldsymbol{v}} - \boldsymbol{v}_a\|_1 = \left\|\frac{b}{\sqrt{\phi}}\boldsymbol{e}_1 + \boldsymbol{e}_2 - \sqrt{\phi}\boldsymbol{e}_1\right\|_1 = \left|\frac{b-\phi}{\sqrt{\phi}}\right| + 1 > 0 \tag{86}$$

The bound is strictly positive because $b \neq \phi$ by the logit difference requirement, and because of the constant $+1$ term from the $\boldsymbol{e}_2$ component. Therefore, $\lim_{n\to\infty}\|\bar{\boldsymbol{v}} - \boldsymbol{v}_a\|_1 \not\to 0$. For the case $d = 1$, we can use the simpler construction $v_i = \frac{b}{\sqrt{\phi}}$ and $v_a = \sqrt{\phi}$, where the non-collapse condition becomes verifying that $\frac{b}{\sqrt{\phi}} \neq \sqrt{\phi}$, which follows from $b \neq \phi$. In contrast, as shown by Barbero et al. (2024), the resulting representations become increasingly similar as $n \to \infty$ with softmax ($\alpha = 1.0$), regardless of the input content, leading to representational collapse. $\square$

**Empirical Verification of Representational Preservation.** To empirically validate our theoretical analysis, we conducted the following experiment: we implemented the counterexample construction using identity projection matrices for queries/keys/values and tested two scenarios with $d = 1$:

1. **Constant prefixes:** $b = 1$ and $\phi = 1.2$.
2. **Random prefixes:** $b \sim \mathcal{U}(0,1)$ and $\phi = 1.2$.

Using a 6-layered transformer with residual connections, we experiment with increasing sequence lengths $n \in \{128, 256, \ldots, 16384\}$ and $\alpha \in \{1.0, 1.5, 1.75, 2.0\}$, and compute $\|\boldsymbol{v}_n^{(L)} - \boldsymbol{v}_{n+1}^{*(L)}\|_1$. Figure 6 shows how representational differences evolve across increasing sequence lengths. As established in the previous counterexample, while softmax attention inevitably leads to representational collapse in long contexts, $\alpha$-entmax can maintain distinct representations even as sequence length grows.

### D.3 PROOF OF PROPOSITION 2 FOR OVER-SQUASHING ALLEVIATION

The following proposition demonstrates how $\alpha$-entmax helps alleviate the problem of over-squashing, the exponential dilution of gradient signals through deep networks. For clarity, we follow the same set of assumptions as Barbero et al. (2024)—independence of attention coefficients from the values and approximation of layer normalization with a constant factor.

**Proposition 4** (Over-squashing Alleviation with $\alpha$-entmax). *Consider an $L$-layer transformer-like model where the attention distribution for each head is computed by $\alpha$-entmax with $\alpha > 1$. For a token $n$ in the final layer, let $\boldsymbol{v}_n^{(L)} \in \mathbb{R}^d$ be its hidden representation and*

$$\boldsymbol{y}_n = \text{norm}_3\big(\boldsymbol{v}_n^{(L)}\big) \tag{87}$$

*be its final normalized output. The sensitivity of $\boldsymbol{y}_n$ to the initial embedding $\boldsymbol{v}_i^{(0)}$ of token $i$ experiences less over-squashing with $\alpha$-entmax than with softmax attention. Specifically, if the support size of the $\alpha$-entmax attention distributions is $|\mathcal{S}_j^{(\ell)}| = s \ll n$ for tokens $j$ across all layers $\ell$, then the number of gradient paths from token $i$ to token $n$ is reduced from $\mathcal{O}(n^L)$ to $\mathcal{O}(s^L)$, and consequently helping to alleviate over-squashing by providing stronger gradient signals.*

*Proof.* We begin by expanding $\frac{\partial \boldsymbol{y}_n}{\partial \boldsymbol{v}_i^{(0)}}$ through the chain rule. Since $\boldsymbol{y}_n = \mathrm{norm}(\boldsymbol{v}_n^{(L)})$, and $\boldsymbol{v}_n^{(L)}$ is the output of $L$ transformer layers, we have:

$$\frac{\partial \boldsymbol{y}_n}{\partial \boldsymbol{v}_i^{(0)}} = \frac{1}{\beta_3} \frac{\partial \boldsymbol{v}_n^{(L)}}{\partial \boldsymbol{v}_i^{(0)}}, \tag{88}$$

where $\frac{1}{\beta_3}$ accounts for the normalization assumption. Expanding the gradient through all $L$ layers:

$$\frac{\partial \boldsymbol{v}_n^{(L)}}{\partial \boldsymbol{v}_i^{(0)}} = \sum_{k_1, k_2, \dots, k_{L-1}} \frac{\partial \boldsymbol{v}_n^{(L)}}{\partial \boldsymbol{v}_{k_{L-1}}^{(L-1)}} \frac{\partial \boldsymbol{v}_{k_{L-1}}^{(L-1)}}{\partial \boldsymbol{v}_{k_{L-2}}^{(L-2)}} \cdots \frac{\partial \boldsymbol{v}_{k_1}^{(1)}}{\partial \boldsymbol{v}_i^{(0)}}. \tag{89}$$

Due to causal masking, the only non-zero terms occur when $i \leq k_1 \leq k_2 \leq \cdots \leq k_{L-1} \leq n$. For each pair of adjacent layers, we have:

$$\frac{\partial \boldsymbol{v}_j^{(\ell+1)}}{\partial \boldsymbol{v}_k^{(\ell)}} = \left( \frac{\sigma_\psi}{\beta_2^{(\ell)}} + 1 \right) \frac{\partial \boldsymbol{u}_j^{(\ell)}}{\partial \boldsymbol{v}_k^{(\ell)}}, \tag{90}$$

where $\boldsymbol{u}_j^{(\ell)} = \sum_{k \leq j} p_{j,k}^{(\ell)} \frac{\boldsymbol{v}_k^{(\ell)}}{\beta_1^{(\ell)}} + \boldsymbol{v}_j^{(\ell)}$, and $p_{j,k}^{(\ell)}$ are the attention probabilities computed using $\alpha$-entmax. Thus, for $k \leq j$, we have:

$$\frac{\partial \boldsymbol{u}_j^{(\ell)}}{\partial \boldsymbol{v}_k^{(\ell)}} = \frac{p_{j,k}^{(\ell)}}{\beta_1^{(\ell)}} + \delta_{j,k} \boldsymbol{I}, \tag{91}$$

where $\delta_{j,k}$ is the Kronecker delta and reflects the contribution from the residual connection, which happens when $k = j$. For simplicity, let $\bar{p}_{j,k}^{(\ell)} = \frac{p_{j,k}^{(\ell)}}{\beta_1^{(\ell)}} + \delta_{j,k}$. Taking the norm and combining all layers, we obtain:

$$\left\| \frac{\partial \boldsymbol{y}_n}{\partial \boldsymbol{v}_i^{(0)}} \right\| \leq C \sum_{k_1 \geq i} \sum_{k_2 \geq k_1} \cdots \sum_{k_{L-1} \geq k_{L-2}} \bar{p}_{n,k_{L-1}}^{(L-1)} \bar{p}_{k_{L-1}, k_{L-2}}^{(L-2)} \cdots \bar{p}_{k_1, i}^{(0)}, \tag{92}$$

where $C = \frac{1}{\beta_3} \prod_{\ell=1}^{L} \left( \frac{\sigma_\psi}{\beta_2^{(\ell)}} + 1 \right)$ is a constant independent of the sequence length.

The crucial distinction between softmax and $\alpha$-entmax lies in the attention probabilities $p_{j,k}^{(\ell)}$. For $\alpha$-entmax with $\alpha > 1$, many tokens receive exactly zero attention. Specifically, if we define the support set for the $j$-th token as $\mathcal{S}_j^{(\ell)} = \{k \mid p_{j,k}^{(\ell)} > 0 \text{ and } k \leq j\}$, then $p_{j,k}^{(\ell)} = 0$ for all $k \notin \mathcal{S}_j^{(\ell)}$. Consequently, $\bar{p}_{j,k}^{(\ell)} = 0$ when $k \notin \mathcal{S}_j^{(\ell)}$ and $j \neq k$ (i.e., when there is no contribution from either attention or the residual connection). This means we can rewrite our bound as:

$$\left\| \frac{\partial \boldsymbol{y}_n}{\partial \boldsymbol{v}_i^{(0)}} \right\| \leq C \sum_{k_1 \in \mathcal{T}_1} \sum_{k_2 \in \mathcal{T}_2(k_1)} \cdots \sum_{k_{L-1} \in \mathcal{T}_{L-1}(k_{L-2})} \bar{p}_{n,k_{L-1}}^{(L-1)} \bar{p}_{k_{L-1}, k_{L-2}}^{(L-2)} \cdots \bar{p}_{k_1, i}^{(0)}. \tag{93}$$

where we precisely characterize the gradient flow paths via the sets $\mathcal{T}_1$ and $\mathcal{T}_\ell(k_{\ell-1})$, which identify tokens that receive non-zero gradient contributions:

$$\mathcal{T}_1 = \{k \in \{i, i+1, \dots, n\} : k \in \mathcal{S}_i^{(0)} \text{ or } k = i\}, \tag{94}$$

$$\mathcal{T}_\ell(k_{\ell-1}) = \{k \in \{k_{\ell-1}, k_{\ell-1}+1, \dots, n\} : k \in \mathcal{S}_{k_{\ell-1}}^{(\ell-1)} \text{ or } k = k_{\ell-1}\} \quad \text{for } \ell > 1. \tag{95}$$

These sets have the following meaning:

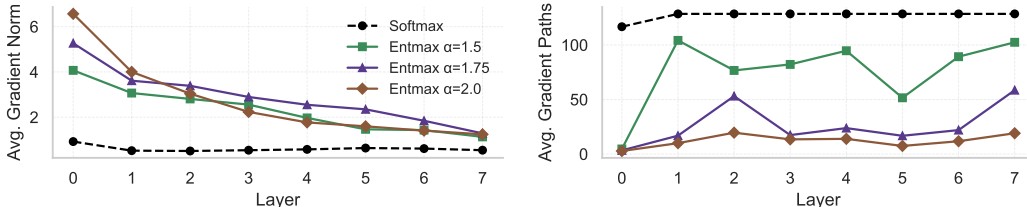

Figure 7: **Left:** Layer-wise gradient norms in an 8-layer transformer with sequence length $n = 256$. $\alpha$-entmax with $\alpha > 1.0$ maintain substantially stronger gradient signals, especially in earlier layers, compared to softmax. This demonstrates how $\alpha$-entmax alleviates over-squashing by enabling more effective gradient flow through the network, with gradient norms up to 6x higher than softmax. **Right:** Visualization of the average number of non-zero gradient paths in a 8-layer transformer per layer, showing how $\alpha$-entmax creates fewer paths, which helps to alleviate over-squashing compared to softmax, which always selects all possible paths (within machine precision).

- $\mathcal{T}_1$ represents the tokens in layer $\ell = 1$ that can receive non-zero gradients from token $i$ in layer $\ell = 0$, either through attention ($k \in \mathcal{S}_i^{(0)}$) or via the residual connection ($k = i$).

- $\mathcal{T}_\ell(k_{\ell-1})$ represents the tokens in layer $\ell$ that can receive non-zero gradients from token $k_{\ell-1}$ in layer $\ell - 1$, either through attention ($k \in \mathcal{S}_{k_{\ell-1}}^{(\ell-1)}$) or via the residual connection ($k = k_{\ell-1}$).

The causal constraint ($k \geq i$ for $\mathcal{T}_1$ and $k \geq k_{\ell-1}$ for $\mathcal{T}_\ell$) is explicitly incorporated in these definitions to account for the causal attention mask. Importantly, the cardinality of these sets directly corresponds to the potential gradient paths through the network. While softmax attention would yield $|\mathcal{T}_\ell(k_{\ell-1})| = n - k_{\ell-1} + 1$ paths from each token, $\alpha$-entmax's sparsity ensures $|\mathcal{T}_\ell(k_{\ell-1})| = |\mathcal{S}_{k_{\ell-1}}^{(\ell-1)}| + 1$.

Hence, if the support size $|\mathcal{S}_j^{(\ell)}| = s \ll n$ and assuming that $i = j$ tokens are always in the support due to the residual connections, this reduces the number of terms in the sum from $\mathcal{O}(n^L)$ to $\mathcal{O}(s^L)$, drastically reducing the total number of gradient paths. Furthermore, as a direct consequence of Lemma 1, since $\alpha$-entmax may concentrate probability mass on fewer tokens, the non-zero $p_{j,k}^{(\ell)}$ values can be larger than with softmax. In such cases, the gradients along the remaining paths will be stronger, helping to further alleviate over-squashing by concentrating gradient flow on important tokens. $\square$

**Empirical Verification of Over-squashing Alleviation.** To empirically validate our theoretical prediction that $\alpha$-entmax reduces gradient paths from $\mathcal{O}(n^L)$ to $\mathcal{O}(s^L)$, we conducted a controlled experiment using a delayed copying task. In this task, the model is presented with a sequence consisting of a prefix of random tokens, followed by a separator token, after which it must reproduce the prefix tokens—creating a natural long-range dependency. We trained 8-layer transformers with different attention mechanisms on sequences of length 256, where the model must copy information from the beginning of the sequence to predict tokens after the separator.

Figure 7 shows the average gradient norm per layer when backpropagating from the loss to each layer input. Softmax exhibits consistently low gradient norms across all layers, indicating severe gradient dilution. In contrast, $\alpha$-entmax variants maintain substantially stronger gradient signals, particularly in the earlier layers of the network. This confirms that gradient information propagates more effectively through the network with $\alpha$-entmax, preserving signal strength even when flowing through multiple layers.

The right part of Figure 7 quantifies the average number of non-zero gradient paths per layer. With softmax, nearly all possible connections remain active within numerical precision, creating $\mathcal{O}(n^2)$ paths per layer. This compounds across layers, resulting in $\mathcal{O}(n^L)$ total paths. $\alpha$-entmax dramatically reduces the number of active paths, with stronger sparsity (higher $\alpha$ values) creating even sharper reductions. Notably, in the first layer less than 5 tokens are kept active on average. This empirically confirms our theoretical claim that $\alpha$-entmax prunes the computational graph to $\mathcal{O}(s^L)$ paths.

# E  INTERACTION WITH POSITIONAL ENCODING

Here, we study—theoretically and empirically—how $\alpha$-entmax with $\alpha > 1$ (sparse) interacts with different positional encoding methods. We follow the same model definition and notation as in §B.

## E.1  NO POSITIONAL ENCODING (NoPE)

We adopt the same set of assumptions as Wu et al. (2025). Namely,

**A1** There exists $C \in \mathbb{R}$ such that $\max_{t \in \mathbb{N}} \{ \|\boldsymbol{W}_Q^{(t)}\|_2, \|\boldsymbol{W}_K^{(t)}\|_2 \} \leq C$.

**A2** The sequence $\{ \| \prod_{t=0}^{k} \boldsymbol{W}_V^{(t)} \|_2 \}_{k=0}^{\infty}$ is bounded.

The first assumption tells that key and query weight matrices are bounded. The second assumption ensures that the node representations' trajectories across $t$ layers stays within a fixed interval $[-C^2, C^2]$.

**Proposition 5** (No Positional Encoding with $\alpha$-entmax). *Let $\mathcal{G}$ be the causal mask graph and $\boldsymbol{P}^{(\ell)} \in \mathbb{R}^{n \times n}$ represent the causal attention matrix at layer $\ell$ computed row-wise using $\alpha$-entmax ($\alpha$ subscript) with $\alpha \in (1, 2]$ or softmax (soft subscript). In particular, let $p_{ij}^{(\ell)}$ denote the $(i, j)$ probability entry in $\boldsymbol{P}^{(\ell)}$. Further, let $\tilde{\boldsymbol{P}}^{(\ell)} = \boldsymbol{P}^{(\ell)} \cdots \boldsymbol{P}^{(0)}$ represent the product of attention matrices through layer $\ell$, which we call the cumulative attention matrix, which captures how information from tokens in the input layer flows to tokens in layer $\ell$ through the composition of attention operations.*

*For softmax, Wu et al. (2025) have shown that*

$$\lim_{\ell \to \infty} \tilde{p}_{soft,i1}^{(\ell)} = 1$$

*for all $1 < i \leq n$.*

*Under assumptions **A1-A2**, for any indices $1 < j \leq i \leq n$, with $\alpha$-entmax we have:*

1. ***Edge deletion:*** *For every layer $\ell$, an edge $(j, i) \in \mathcal{G}$ is present in the dynamic graph $\mathcal{G}_\alpha^{(\ell)}$ if and only if $(\alpha - 1)z_{ij}^{(\ell)} > \tau(\boldsymbol{z}_i^{(\ell)})$, where $z_{ij}^{(\ell)} = \langle \boldsymbol{q}_i^{(\ell)}, \boldsymbol{k}_j^{(\ell)} \rangle$ and $\tau(\boldsymbol{z}_i^{(\ell)})$ is the entmax threshold. Otherwise, $p_{\alpha,ij}^{(\ell)} = 0$ and the edge is removed for that layer.*

2. ***Modified attention patterns:*** *Unlike softmax, $\alpha$-entmax with $\alpha > 1$ creates sparse attention patterns by completely removing some connections. For tokens that survive the threshold, the behavior of their attention weights depends on:*

   - *How far the token's logit sits above the threshold*
   - *The relative differences between logits*

   *For tokens that remain connected through the dynamic attention graph $\mathcal{G}_\alpha^{(\ell)}$ at all layers, the cumulative attention still exhibits a decay pattern:*

$$\tilde{p}_{\alpha,ij}^{(\ell)} \leq C(1 - \delta_{ij})^\ell \tag{96}$$

   *where $\delta_{ij}$ depends on the connectivity pattern of the dynamic attention graph. This decay rate differs from softmax due to edge pruning and the redistribution of probability mass.*

3. ***Disrupted limit behavior:*** *Unlike softmax, $\alpha$-entmax does not necessarily converge to the first token:*

   (a) *If for every layer $\ell$, there exists at least one directed path from token 1 to token $i$ in the dynamic graph $\mathcal{G}_\alpha^{(\ell)}$, then:*

$$\lim_{\ell \to \infty} \tilde{p}_{\alpha,i1}^{(\ell)} = 1. \tag{97}$$

> *(b) If at some layer $\ell_0$, directed paths from token 1 to token $i$ are deleted in $\mathcal{G}_\alpha^{(\ell_0)}$, then:*
>
> $$0 \le \lim_{\ell \to \infty} \tilde{p}_{\alpha,i1}^{(\ell)} < 1, \tag{98}$$
>
> *with the exact limit determined by the structure of the strongly connected components formed in the dynamic graph.*

*Proof.* **(i) Edge deletion:** For $\alpha$-entmax, a coefficient is non-zero if and only if its pre-activation exceeds the layer-specific threshold. The stated condition follows directly from the definition of $\alpha$-entmax:

$$p_{\alpha,ij}^{(\ell)} = [(\alpha - 1)z_{ij}^{(\ell)} - \tau(\boldsymbol{z}_i^{(\ell)})]_+^{\frac{1}{\alpha-1}}, \tag{99}$$

where $[x]_+ = \max(0, x)$.

**(ii) Modified attention patterns:** For $\alpha$-entmax, the attention weights are determined by thresholding:

$$p_i^\alpha = [(\alpha - 1)z_i - \tau]_+^{\frac{1}{\alpha-1}}. \tag{100}$$

Let $\tau' = \frac{\tau}{\alpha-1}$ for simplicity. Consider two tokens with logits $z_i \ge z_j$ both in the support. The ratio of their probabilities is:

$$\frac{p_j^\alpha}{p_i^\alpha} = \left(\frac{z_j - \tau'}{z_i - \tau'}\right)^{\frac{1}{\alpha-1}}. \tag{101}$$

For softmax, the ratio is:

$$\frac{p_j^{\text{soft}}}{p_i^{\text{soft}}} = e^{-(z_i - z_j)}. \tag{102}$$

The comparison between these ratios depends on how far $z_j$ sits above $\tau'$. Let $\Delta = z_i - z_j$ and $b = z_j - \tau'$. Through algebraic manipulation, we can show:

$$\frac{p_j^\alpha}{p_i^\alpha} \gtrless \frac{p_j^{\text{soft}}}{p_i^{\text{soft}}} \Leftrightarrow b \gtrless \frac{\Delta}{e^{(\alpha-1)\Delta} - 1}. \tag{103}$$

This means the relative behavior of attention weights in $\alpha$-entmax compared to softmax depends on the specific configuration of logits and thresholds, rather than following a simple universal relationship. Since tokens that remain connected must distribute probability mass among fewer options (due to pruning), there exists some $0 < \delta_{ij} < 1$ such that:

$$\tilde{p}_{\alpha,ij}^{(\ell)} \le C(1 - \delta_{ij})^\ell. \tag{104}$$

The specific value of $\delta_{ij}$ depends on the connectivity pattern of the dynamic attention graph and may differ significantly from the softmax case due to edge pruning and probability redistribution.

**(iii) Disrupted limit behavior.**

**(a) Case where paths to token the first token persist:**

Suppose that for every layer $\ell$, there exists at least one directed path from token 1 to token $i$ in the dynamic graph $\mathcal{G}_\alpha^{(\ell)}$— the unique "center node" as defined by Wu et al. (2025). For any token $j > 1$, the geometric decay established in part (ii) applies:

$$\tilde{p}_{\alpha,ij}^{(\ell)} \le C(1 - \delta_{ij})^\ell \to 0 \text{ as } \ell \to \infty. \tag{105}$$

Since the row sums of $\tilde{\boldsymbol{P}}^{(\ell)}$ must equal 1 (as it is a product of row-stochastic matrices), and all entries $\tilde{p}_{\alpha,ij}^{(\ell)}$ with $j > 1$ approach 0, we have:

$$\lim_{\ell \to \infty} \tilde{p}_{\alpha,i1}^{(\ell)} = 1 - \lim_{\ell \to \infty} \sum_{j=2}^{i} \tilde{p}_{\alpha,ij}^{(\ell)} = 1. \tag{106}$$

**(b) Case where paths to token 1 are cut:** The key difference from softmax arises when edge deletion creates a configuration where token 1 cannot reach token $i$. Let $\ell_0$ be the first layer where paths from token 1 to token $i$ are removed in $\mathcal{G}_\alpha^{(\ell_0)}$. Let $\mathcal{C}_i^{(\ell_0)} \subset \{1, 2, ..., n\}$ be the set of tokens in the same strongly connected component as token $i$ in $\mathcal{G}_\alpha^{(\ell_0)}$. By our assumption, $1 \notin \mathcal{C}_i^{(\ell_0)}$. At layer $\ell_0$, the attention probability is distributed only among tokens in $\mathcal{C}_i^{(\ell_0)}$:

$$\sum_{j \in \mathcal{C}_i^{(\ell_0)}} p_{\alpha,ij}^{(\ell_0)} = 1 \text{ and } p_{\alpha,i1}^{(\ell_0)} = 0. \tag{107}$$

For all layers $\ell > \ell_0$, the multiplication by zero ensures $\tilde{p}_{\alpha,i1}^{(\ell)} = 0$, and therefore:

$$0 \le \lim_{\ell \to \infty} \tilde{p}_{\alpha,i1}^{(\ell)} < 1. \tag{108}$$

The exact limit depends on the structure of the strongly connected components formed in the dynamic graph through subsequent layers. In fact, the limit can be exactly zero whenever all paths from token 1 to token $i$ are removed from $\mathcal{G}_\alpha$ since layer $\ell_0$. $\qquad\square$

This proposition demonstrates a fundamental difference between softmax and $\alpha$-entmax transformers: while softmax inevitably leads to concentration of attention on the first token, $\alpha$-entmax can potentially disrupt this position bias through its ability to dynamically prune edges in the attention graph. This provides a theoretical foundation for using sparse attention mechanisms to mitigate position bias in transformer architectures.

### E.2  ALIBI

We start by recalling the definition of ALiBi from (Press et al., 2022).

**Definition 2** (ALiBi Positional Encoding). *Let $H$ be the number of attention heads. A general form of ALiBi bias for head $h \in \{1, 2, \ldots, H\}$ can be defined as:*

$$b_{ij}^{(h)} = \begin{cases} m_h(j - i) & \text{if } j \le i \\ 0 & \text{otherwise,} \end{cases} \tag{109}$$

*where $m_h \in \mathbb{R}_+$ is the slope parameter for head $h$, defined as $m_h = 2^{-\frac{h}{H}}$.*

Now, we consider how it interacts with $\alpha$-entmax.

**Proposition 6** (ALiBi with $\alpha$-entmax). *Consider $\alpha$-entmax attention with the ALiBi bias from the above definition. Assume the raw attention logits $z_{ij}^{(h)} \in [z_{\min}^{(h)}, z_{\max}^{(h)}]$ for all $i, j$. Let*

$$d_{\max}^{(h)} = \left\lfloor \frac{z_{\max}^{(h)} - z_{\min}^{(h)} + \frac{1}{\alpha - 1}}{m_h} + 1 \right\rfloor. \tag{110}$$

*Then, any token $j$ with $(i - j) > d_{\max}^{(h)}$ receives zero attention from token $i$ at head $h$.*

*Proof.* For a token $j$ to receive non-zero attention from token $i$ with $\alpha$-entmax, we require:

$$(\alpha - 1)(z_{ij}^{(h)} + b_{ij}^{(i)}) > \tau, \tag{111}$$

where $\tau$ is the threshold ensuring normalization. Since slopes are positive ($m_i > 0$), we have:

$$(\alpha - 1)(z_{ij}^{(h)} - (i - j)m_h) > \tau. \tag{112}$$

In the extreme case where only the closest token receives attention (single-support case), we can solve for $\tau$ exactly:

$$1 = \left[(\alpha - 1)(z_{\max}^{(h)} - \tau)\right]^{\frac{1}{\alpha - 1}} \Rightarrow (\alpha - 1)(z_{\max}^{(h)} - \tau) = 1 \Rightarrow \tau = z_{\max}^{(h)} - \frac{1}{\alpha - 1}. \tag{113}$$

Since the logits drop by at least $m_h$ per position due to the ALiBi bias, we have $z_{\max}^{(h)} \geq z_{\min}^{(h)} + m_h$, which gives us:

$$\tau \geq z_{\min}^{(h)} + m_h - \frac{1}{\alpha - 1}. \tag{114}$$

For a token $j$ with distance $(i - j)$, even with the maximum logit $z_{\max}^{(h)}$:

$$(\alpha - 1)(z_{\max}^{(h)} - (i - j)m_h) \leq (\alpha - 1)(z_{\min}^{(h)} - m_h) - 1. \tag{115}$$

Solving for $(i - j)$:

$$(i - j) \geq \frac{z_{\max}^{(h)} - z_{\min}^{(h)} + m_h + \frac{1}{\alpha - 1}}{m_h}. \tag{116}$$

Hence,

$$d_{\max}^{(h)} = \left\lfloor \frac{z_{\max}^{(h)} - z_{\min}^{(h)} + \frac{1}{\alpha - 1}}{m_h} + 1 \right\rfloor. \tag{117}$$

$\square$

This proposition establishes that with $\alpha$-entmax and ALiBi positional bias, there exists a head-dependent hard cutoff distance $d_{\max}^{(h)}$ beyond which tokens receive exactly zero attention. This creates an adaptive but bounded attention window that depends on both content relevance ($z_{\max}^{(h)} - z_{\min}^{(h)}$) and the sparsity parameter $\alpha$, naturally limiting the effective context without requiring explicit truncation. This property allows the model to focus computational resources on a relevant window of tokens, which can be particularly valuable for efficiently processing long documents.

### E.3 RoPE

To analyze the interaction between RoPE (Su et al., 2024) and $\alpha$-entmax, we first establish our notation. Following Barbero et al. (2025), we consider queries $\boldsymbol{q}_i \in \mathbb{R}^d$ and keys $\boldsymbol{k}_j \in \mathbb{R}^d$, where $i$ and $j$ are token positions in the sequence. RoPE decomposes these vectors into $d/2$ two-dimensional chunks, denoted as $\boldsymbol{q}_i^{(k)} \in \mathbb{R}^2$ and $\boldsymbol{k}_j^{(k)} \in \mathbb{R}^2$ for $k \in \{1, \dots, d/2\}$. Each chunk rotates at a different frequency $g_k = \theta^{-2(k-1)/d}$, where $\theta$ (typically 10,000) is the base wavelength parameter.

RoPE applies position-dependent rotations through matrices $\rho(g_k)^i$ to transform the original queries and keys. The resultant raw attention logit between query position $i$ and key position $j$ is: In a standard transformer with RoPE,

$$z_{ij} = \sum_{k=1}^{d/2} \langle \boldsymbol{q}_i^{(k)}, \rho(g_k)^{j-i} \boldsymbol{k}_j^{(k)} \rangle, \tag{118}$$

where $\rho(g_k)$ is the 2D rotation matrix with frequency $g_k$.

**Query-Key Interaction with RoPE.** Let $\phi_{ijk}$ be the angle between the original (unrotated) vectors $\boldsymbol{q}_i^{(k)}$ and $\boldsymbol{k}_j^{(k)}$. That is:

$$\cos(\phi_{ijk}) = \frac{\langle \boldsymbol{q}_i^{(k)}, \boldsymbol{k}_j^{(k)} \rangle}{\|\boldsymbol{q}_i^{(k)}\|_2 \|\boldsymbol{k}_j^{(k)}\|_2}. \tag{119}$$

For a single 2D chunk, since rotation preserves vector magnitudes, the contribution to the raw score is:

$$\langle \boldsymbol{q}_i^{(k)}, \rho(g_k)^{j-i} \boldsymbol{k}_j^{(k)} \rangle = \|\boldsymbol{q}_i^{(k)}\|_2 \|\rho(g_k)^{j-i} \boldsymbol{k}_j^{(k)}\|_2 \cos(\phi_{ijk} + g_k(j - i)) \tag{120}$$

$$= \|\boldsymbol{q}_i^{(k)}\|_2 \|\boldsymbol{k}_j^{(k)}\|_2 \cos(\phi_{ijk} + g_k(j - i)), \tag{121}$$

where $\phi_{ijk}$ is the original angle between $\boldsymbol{q}_i^{(k)}$ and $\boldsymbol{k}_j^{(k)}$. As shown in Proposition 3.1 of Barbero et al. (2025), RoPE allows for maximal attention at any arbitrary distance. However, RoPE combined with $\alpha$-entmax creates a hard boundary on attention distance due to the thresholding effect, which we analyze next.

**Approximation for Small Angles.** First, note that we can use the angle-sum expansion for cosine as follows:

$$\cos(\phi_{ijk} + g_k(j - i)) = \cos(\phi_{ijk})\cos(g_k(j - i)) - \sin(\phi_{ijk})\sin(g_k(j - i)). \quad (122)$$

Further, note that for small angles $\phi_{ijk}$ we can use a second-order Taylor expansion for $g_k(j - i)$:[6]

$$\cos(g_k(j - i)) \approx 1 - \frac{g_k^2(j - i)^2}{2}. \quad (123)$$

Finally, applying the dot-product:

$$z_{ij} \approx \sum_{k=1}^{d/2} \|\boldsymbol{q}_i^{(k)}\|_2 \|\boldsymbol{k}_j^{(k)}\|_2 \cos(\phi_{ijk}) - \sum_{k=1}^{d/2} \|\boldsymbol{q}_i^{(k)}\|_2 \|\boldsymbol{k}_j^{(k)}\|_2 \cos(\phi_{ijk}) \frac{g_k^2(j - i)^2}{2} + \sin\text{ terms} \quad (124)$$

$$\approx z_{\max} - \sum_{k=1}^{d/2} c_k g_k^2(i - j)^2. \quad (125)$$

Here, we simplified the last step by focusing on the quadratic decay from the cosine term while omitting the sine terms $-\sin(\phi_{ijk})g_k(j - i)$. For semantically aligned tokens where $\phi_{ijk} \approx 0$, the sine term's contribution is minimal since $\lim_{x \to 0} \sin(x) = 0$.

> **Proposition 7** (Maximum Attention Distance for RoPE (Small-Angle Regime)). *Let* $g_{\max} = \max_k g_k$ *be the maximum frequency in RoPE. Within the* small-angle domain *where*
>
> $$|i - j| \leq \frac{\pi}{2g_{\max}},$$
>
> *so that all rotational angles* $\theta = g_k(i - j)$ *satisfy* $|\theta| \leq \frac{\pi}{2}$*, and assuming* $z_{\max} > \frac{\tau(\boldsymbol{z}_i)}{\alpha - 1}$*, there exists a* critical distance $d_{\max}$ *beyond which tokens receive exactly zero attention:*
>
> $$d_{\max} = \left\lfloor \sqrt{\frac{z_{\max} - \frac{\tau(\boldsymbol{z}_i)}{\alpha - 1}}{\sum_{k=1}^{d/2} c_k g_k^2}} \right\rfloor. \quad (126)$$

*Proof.* For a token to receive non-zero attention under $\alpha$-entmax, we must have $z_{ij} > \frac{\tau(\boldsymbol{z}_i)}{\alpha - 1}$. Substituting the decay pattern from Equation 124, which is valid within the small-angle domain where cosine can be approximated using a Taylor expansion $\cos\theta \approx 1 - \frac{\theta^2}{2}$:

$$z_{\max} - \sum_{k=1}^{d/2} c_k g_k^2(i - j)^2 > \frac{\tau(\boldsymbol{z}_i)}{\alpha - 1}. \quad (127)$$

Rearranging for $(i - j)^2$:

$$(i - j)^2 < \frac{z_{\max} - \frac{\tau(\boldsymbol{z}_i)}{\alpha - 1}}{\sum_{k=1}^{d/2} c_k g_k^2}. \quad (128)$$

Taking the floor of the square root gives us $d_{\max}$. □

Note that this analysis applies to the first attention window. Due to the periodicity of rotation operations, at distances beyond $\frac{\pi}{g_k}$ for any frequency component $k$, the attention pattern may exhibit additional windows of non-zero attention, which we address next.

---

[6] $\cos(x) \approx 1 - x^2/2 + \text{higher order terms}$.

**Proposition 8** (Frequency-Specific Cutoff for RoPE). *For each frequency component $k$, let $\beta_k = \frac{\tau(\boldsymbol{z}_i)}{(\alpha-1)\|\boldsymbol{q}_i^{(k)}\|_2\|\boldsymbol{k}_j^{(k)}\|_2}$. Since at least one token must receive non-zero attention for $\alpha$-entmax to yield a valid probability distribution, $\beta_k \leq 1$ must hold for at least one component. Assuming $\beta_k \in [-1, 1]$ (covering all possible cosine values), there exists a sequence of distances $\{d_{k,n}\}_{n=0}^{\infty}$ at which its contribution to attention crosses the threshold.*

*The first such distance is:*

$$d_{k,0} = \left\lfloor \frac{1}{g_k} \arccos\left( \frac{\tau(\boldsymbol{z}_i)}{(\alpha-1)\|\boldsymbol{q}_i^{(k)}\|_2\|\boldsymbol{k}_j^{(k)}\|_2} \right) \right\rfloor. \tag{129}$$

*Due to the periodicity of cosine, subsequent threshold crossings occur at approximately:*

$$d_{k,n} \approx \frac{2\pi n \pm d_{k,0}}{g_k}, \quad n \in \mathbb{N}. \tag{130}$$

*Furthermore, $d_{k,0}$ is non-increasing in $\alpha$ and inversely proportional to $g_k$.*

*Proof.* For a single frequency component $k$, the contribution to the raw score from Equation 120 is:

$$\langle \boldsymbol{q}_i^{(k)}, \rho(g_k)^{j-i}\boldsymbol{k}_j^{(k)} \rangle = \|\boldsymbol{q}_i^{(k)}\|_2\|\boldsymbol{k}_j^{(k)}\|_2 \cos(g_k(j-i) + \phi_{ijk}). \tag{131}$$

For this to exceed the attention threshold under optimal alignment ($\phi_{ijk} = 0$, which maximizes the contribution):

$$(\alpha-1)\|\boldsymbol{q}_i^{(k)}\|_2\|\boldsymbol{k}_j^{(k)}\|_2 \cos(g_k(j-i)) > \tau(\boldsymbol{z}_i) \tag{132}$$

$$\cos(g_k(j-i)) > \frac{\tau(\boldsymbol{z}_i)}{(\alpha-1)\|\boldsymbol{q}_i^{(k)}\|_2\|\boldsymbol{k}_j^{(k)}\|_2}. \tag{133}$$

Taking the arccos of both sides and dividing by $g_k$ gives the first threshold crossing distance $d_{k,0}$:

$$|i - j| > \frac{1}{g_k} \arccos\left( \frac{\tau(\boldsymbol{z}_i)}{(\alpha-1)\|\boldsymbol{q}_i^{(k)}\|_2\|\boldsymbol{k}_j^{(k)}\|_2} \right). \tag{134}$$

Due to the $2\pi$-periodicity of cosine, subsequent threshold crossings occur at distances $\frac{2\pi n \pm d_{k,0}}{g_k}$ for integers $n > 0$. Moreover, $\tau(\boldsymbol{z}_i)/(\alpha-1)$ grows as $\alpha$ increases, making the arccos term smaller and consequently decreasing $d_{k,0}$. The inverse proportionality to $g_k$ is evident directly from the formula. $\square$

These theoretical analyses of RoPE with $\alpha$-entmax reveal two interesting takeaways. First, different frequency components in RoPE naturally create attention windows of different widths. High-frequency components (large $g_k$) produce very narrow windows focused on local context, while low-frequency components (small $g_k$) enable attention over longer distances. Second, the sparsity pattern induced by the combination of RoPE and $\alpha$-entmax is not uniform but varies across frequency components, creating a more complex attention structure than simple distance-based decay methods like ALiBi.

### E.4 COMPARISON BETWEEN POSITIONAL ENCODING METHODS WITH $\alpha$-ENTMAX

The positional encoding scheme used in a transformer has significant implications for how attention behaves over long contexts. Our theoretical approach from previous subsections reveals that $\alpha$-entmax interacts with positional encodings in ways that fundamentally alter attention behavior compared to softmax. We summarize our theoretical findings next, along with an empirical analysis within a controlled experimental setting.

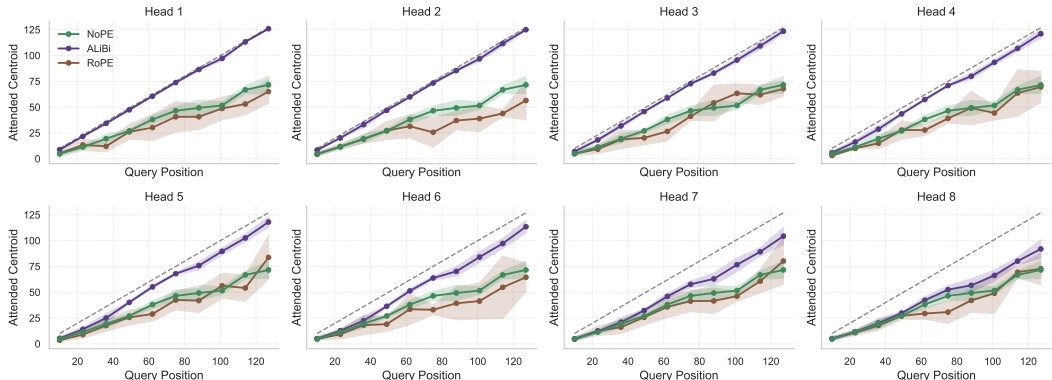

Figure 8: Per-head attention centroids across different positional encoding methods. Each panel represents one attention head's behavior. The dashed line shows the identity function (attending to self). NoPE heads consistently exhibit an early-token bias, ALiBi heads maintain proximity-based attention, and RoPE heads display more varied patterns with irregular fluctuations.

**Theoretical Analysis.** With NoPE (Kazemnejad et al., 2023), softmax transformers (without MLP layers) develop an implicit bias towards the first tokens as depth increases, as shown by Wu et al. (2025). $\alpha$-entmax disrupts this behavior through its ability to create disconnected attention graphs. By assigning exactly zero attention to some connections, it may remove the implicit bias encouraging attention to concentrate on early tokens.

When combined with ALiBi (Press et al., 2022), $\alpha$-entmax transforms the smooth linear decay into a hard attention window. For tokens separated by distance $d > d_{\max}^{(h)}$, where $d_{\max}^{(h)} = \left\lfloor \frac{1}{m_h}(z_{\max} - z_{\min} + \frac{1}{\alpha-1}) + 1 \right\rfloor$, attention weights become exactly zero. This creates an adaptive but bounded attention window that depends on the input ($z_{\max} - z_{\min}$) and the sparsity parameter $\alpha$.

With RoPE (Su et al., 2024), $\alpha$-entmax induces frequency-dependent sparsity. Each frequency component $k$ has a critical distance $d_k$ beyond which its contribution falls below the attention threshold. This creates a multi-scale attention pattern where nearby tokens interact through all frequency components, while distant tokens interact only through low-frequency components.

These interactions between positional encodings and $\alpha$-entmax have important practical implications, and further motivate the introduction of our hybrid approach, NAPE (NoPE + ALiBi). By creating natural, content-adaptive attention windows, NAPE combine the benefits of sparse, focused attention with awareness of token positions, allowing models to effectively balance local and global information processing. This provides a principled alternative to manually designed sparse attention patterns like sliding windows or dilated attention, with the advantage of adapting to content relevance rather than using fixed patterns.

**Empirical Analysis Setup.** We simulated a sequence of length $n = 128$ with attention heads using $\alpha$-entmax ($\alpha = 1.5$). For each query position $i$, we generated a vector of raw attention scores (logits) for all positions $j \leq i$ according to:

$$z_{ij} = z_{ij}^{\text{base}} + z_{ij}^{\text{prox}} + z_{ij}^{\text{noise}}, \tag{135}$$

where $z_{ij}^{\text{base}} \sim \mathcal{U}(Z_{\min}, Z_{\max})$ represents the base content-based affinity, $z_{ij}^{\text{prox}} = 0.2(Z_{\max} - Z_{\min})(1 - 0.5\frac{|j-i|}{i})$ introduces a proximity bias, and $z_{ij}^{\text{noise}} \sim \mathcal{N}(0, \sigma^2)$ adds Gaussian noise. The parameters were set to $Z_{\min} = -5.0$, $Z_{\max} = 5.0$, and $\sigma = 0.5$. This formulation models a realistic mixture of content-based attention (random component), a mild inherent bias toward nearby tokens (proximity component), and natural variation (noise component). We then applied different positional encoding methods to modify these base logits. Finally, we calculated the attention distribution using $\alpha$-entmax and determined the attention centroid for each query position $i$ as $\text{centroid}_i = \sum_{j=1}^{i} j \cdot p_{ij}$, where $p_{ij} = \alpha\text{-entmax}(\boldsymbol{z}_i)_j$.

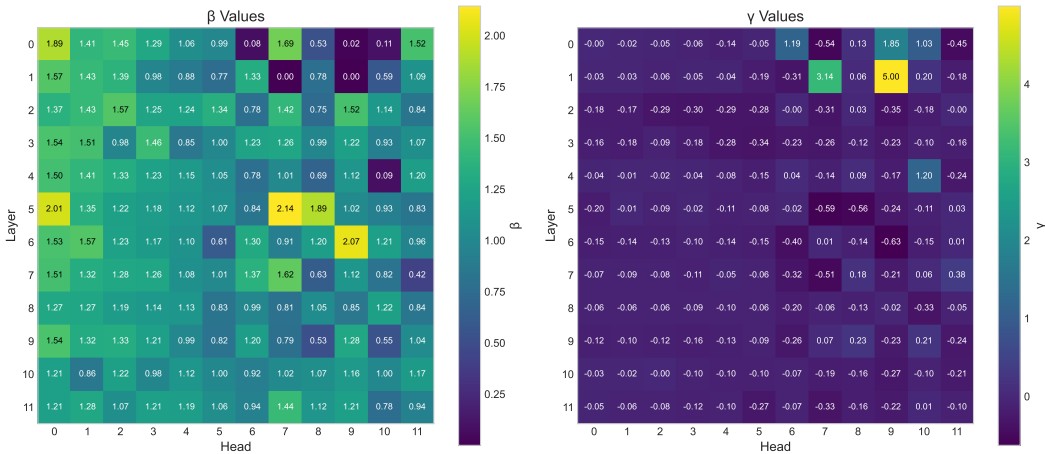

Figure 9: Heatmaps for $\beta$ and $\gamma$ per head for Scalable $\alpha$-entmax.

**Empirical Results.** The head-specific analysis in Figure 8 reveals distinct behaviors across positional encoding methods when combined with $\alpha$-entmax. While seems to NoPE exhibit a *weak* bias towards earlier positions, it also shows a modest variability, indicating more disperse attention. ALiBi clearly creates a consistent recency bias, with centroids following slightly below the identity line, maintaining low variability that indicates focused attention. RoPE demonstrates centroid patterns similar to NoPE but with lower entropy (higher variability), suggesting a focused attention in more distant positions. These observations may explain why NAPE—the hybrid NoPE+ALiBi—works well in practice, since ALiBi heads provide consistent positional structure focused on recent context, while NoPE heads can contribute complementary via early-token and semantic focus, creating a more balanced attention mechanism than either approach alone. In fact, as we show in §H.1, models equipped with NoPE are flexible enough and can acquire relative positional encoding, thus also supporting the original hypothesis of Kazemnejad et al. (2023). Therefore, in NAPE has the ability to encourage short-span focus with ALiBi alongside learning more longer-span focus that are guided via semantic information with NoPE.

# F    ADAPTIVE-SCALABLE $\alpha$-ENTMAX (ASENTMAX)

## F.1    LEARNING SCALERS FOR LANGUAGE MODELING

To verify our scaling approach, we follow the setup from (Nakanishi, 2025) and trained a language model with learnable scales $p_i$ for each $i$-th position. However, we do so independently for each head in the model. Specifically, we train a 12-layer transformer with approximately 120 million parameters. We set hidden size to 768, attention heads to 12, MLP intermediate size to 2048, learning rate to $6 \times 10^{-4}$, weight decay to 0.001, batch size to 1M tokens, and sequence length to 1024. Each head thus contains 1024 learnable parameters (1 per position). Finally, we train on the FineWeb dataset for a total of 5 billion tokens.

In Figure 3 we show how the $\delta_h + \beta_h(\log n)^{\gamma_h}$ scaling performs well for different attention heads. In contrast, removing $\gamma_h$—as done by Nakanishi (2025)—leads to a severe degradation fit for several heads. Specifically, the linear fit has an overall $R^2 = 0.12$, the log fit has $R^2 = -14$ (severe underfitting), and our log with a $\gamma$ exponent has $R^2 = 0.17$. The full set of $\beta_h$ and $\gamma_h$ learned by our approach are shown in Figure 9.

**Effect of Negative $\gamma_h$.** Experiments on the Copy task, shown in Table 10, suggest that, without scaling, $\alpha$-entmax can hurt performance, leading to a noticeable drop in accuracy in the OOD scenario. Introducing an adaptive temperature, however, substantially mitigates this effect. We hypothesize that Copy requires less sparse attention patterns, which can be accomplished by applying a negative power to the logarithm function. We confirm this hypothesis in Figure 13, which shows that ASEntmax learns negative values of $\gamma_h$ in all heads, resulting in more spread-out attention distributions.

# G    EXPERIMENTAL DETAILS

## G.1    SYNTHETIC DATA

Following the data diversity assumptions of Zhou et al. (2024), we generate a large number of samples—between 10 million and 50 million, depending on task complexity. See Table 5 for training/test details on for each task along with model hyperparameters. The 2Back and Local Count tasks are token-classification tasks, while the remaining tasks are generative. In Figure 10, we show examples of the tasks we introduce in this work.

**2Back.**    In this classification task, the model must predict the class of the token that appeared two positions earlier. Via this task, we examine the ability of models equipped with NoPE to learn relative positional bias and assess their behaviour in out-of-distribution scenarios (see H.1).

**Local Count.**    The Local Count task is a classification task in which the model must predict the number of times a word has occurred so far. We restrict the vocabulary size to 16, allowing multiple clusters of the same word to appear within a sequence multiple times. This increases the task's difficulty, as the model must distinguish between different clusters of identical words. We sample the number of repetitions for each cluster uniformly from $\mathcal{U}(1, 48)$ to test whether models equipped with NoPE can learn a longer focus span than observed in the 2Back task.

**Flip-Flop (Liu et al., 2023).**    Flip-Flop uses sequences made up of alternating instructions "write", "ignore", and "read" (*w, i ,r*). Each instruction is paired with a bit (0 or 1). The model must memorize the bit if it follows a *w*, and recall the last stored bit when it encounters an *r*. For example, in the string "i 0 w 0 i 1 i 1 w 1 i 0 i 1 r", the correct output is 1. We conducted experiments on two datasets with different "write" instruction probabilities: $10\%$ - sparse, $80\%$ - dense.

**Max Retrieval.**    We follow Veličković et al. (2025) in constructing the Max Retrieval dataset and the model architecture.

**Multi-Query Multi-Token Associative Recall (MQMTAR).**    MQMTAR is a retrieval task in which models must produce a sequence of multi-token values corresponding to the queries provided at the end of the input sequence (see Figure 10). MQMTAR employs three special tokens: (1) 0 for empty space; (2) 1 as the key–value delimiter; and (3) 3 as the query delimiter, which is also used to separate values in the target sequence. We set the lengths of both keys and values to be 2 tokens, resulting in 5 tokens per key-value pair in the input. The number of queries is 4, and the density of key-value pairs is $80\%$ of the total number of tokens. Finally, the size of the alphabet is 256, but we limited the size of the key/value vocabulary to 10K. This results in a total of 100K key–value pairs.

**Sort, Copy and Reverse.**    These are well-known tasks for testing models' length generalization (Kazemnejad et al., 2023). We use a small vocabulary size of 32 to generate more sequences with repeated tokens, since models must handle such repetitions increasingly as sequence length grows.

## G.2    MODELS FOR SYNTHETIC TASKS

All synthetic tasks are trained with a decoder-like transformer. We evaluate models in extreme settings by using as few layers as possible as our aim is to test the attention mechanism coupled with our positional-encoding strategy, rather than the scaling capabilities of transformer. However, for the Reverse task—which proved particularly challenging for softmax-based models—we increment the layer count until the softmax baseline generalizes to at least 1.5x the in-distribution length.

For experiments with RoPE, we use the Hugging Face implementation from LLaMA 3 (Grattafiori et al., 2024), which includes RoPE scaling. Since our sequences are relatively short, the base frequency is set to its default value of $10,000$. To improve length extrapolation in RoPE-based models, we apply a scaling factor of 16, which we found to be optimal for Flip-Flop under $4\times$ extrapolation (see Table 11); factors of 8 or 32 degrade performance. For NAPE, each ALiBi head uses a slope of $m = \frac{1}{h}$, where $h$ is the head index. We employ 8 attention heads for all tasks except Copy and MQMTAR, where 16 heads yields a performance boost across all models.

```
Multi-query Multi-token Associative Recall
           k1   v1        k2   v2      k3   v3       q1  q2
Input : 0 46 178 000 55 139 00 36 168 00 23 6246
Target: 2 68 278
```

```
2Back
Input : 0 0 2 5 2 6 9 7 1 2 3 3 8 2 2
Target: 0 0 0 0 2 5 2 6 9 7 1 2 3 3 8
```

```
Local Count
Input : 2 2 2 4 4 6 6 8 8 8 8 2 2
Target: 0 1 2 0 1 0 1 0 1 2 3 0 1
```

Figure 10: Examples of the introduced tasks. **MQMTAR**: Each digit is a token; the alphabet size is 10, and the number of queries is 2. **2Back**: A special token 0 is added at the beginning of the sequence to ensure the model has something to predict at the first two positions; the vocabulary size is 10. **Local Count**: The maximum number of repetitions is 4, and the vocabulary size is 8.

In case of ASEntmax, $\beta_h^i$ and $\gamma_h^i$ are computed per token $i$ via linear projections followed by activations softplus and tanh respectively, allowing to scale attention adaptively based on content. For our experiments with $\alpha$-entmax, we use $\alpha = 1.5$ as the default value for the $\alpha$-entmax models, unless mentioned otherwise. Furthermore, we use the Gemma2 implementation from Hugging Face, but disabling sliding-window attention in all layers. For experiments with $\alpha$-entmax, we replaced FlashAttention with AdaSplash (Gonçalves et al., 2025).

For optimization, we use the AdamW with default betas and a cosine learning-rate scheduler with warm-up, setting 10K warm-up steps. Given the large training corpus, we do not employ dropout or weight decay. In addition, we use bfloat16 in all experiments. All models are relatively small (2–10M parameters) and fit on a single GPU.

We observe that even when models are $100\%$ accurate in-distribution, they still require significantly more training; in some cases, the training loss reached as low as $10^{-8}$. Therefore, the best checkpoint is selected based on performance at $8\times$ the in-distribution sequence length. In some cases such as the Sort task and models with RoPE, where generalization up to $8\times$ was not possible, we use BLEU as an intermediate metric and $2\times$ or $4\times$ the in-distribution sequence length. We perform evaluation with $1K$ samples per sequence length. 2Back and Local Count are evaluated using accuracy, as they are classification tasks, whereas for the remaining tasks, we use exact match accuracy—assigning 1 only if the entire predicted sequence matches the reference and 0 otherwise. We report results for the single best-performing model selected from experiments conducted with multiple random seeds (3) and various learning rates.

For some tasks, we also report results for SEntmax—which learns $\beta$ and $\gamma$ directly, without linear projections or nonlinearities—and also for ASSMax—which applies our adaptive-scaling strategy to Softmax. Finally, for some tasks we also experiment with Stick-Breaking Attention (SB, Tan et al. 2025), which corresponds to a different positional encoding strategy.

### G.3 LANGUAGE MODELING

We train 420M-parameter decoder-only models following the LLaMA-3 architecture, using the following hyperparameters: (1) model dimension: 768; (2) number of layers: 24; (3) number of heads: 12. For RoPE and FFN layers, we use the default LLaMA-3 settings. The training is conducted using the *torchtitan* library (Liang et al., 2025). We train for a total of 13.5K steps, with 1.35K steps allocated for warm-up. The learning rate is set to $3 \cdot 10^{-4}$, and the cooldown phase corresponds to $10\%$ of the total steps with a minimum learning rate of $3 \cdot 10^{-5}$, where we apply cosine decay scheduling. We use the AdamW optimizer and train with bfloat16 precision. The global batch

Table 5: Task details and hyperparameters.

| Task | Samples | Length | Batch | Vocab. | Heads | Layers | Hid. dim. | Int. dim. |
|------|---------|--------|-------|--------|-------|--------|-----------|-----------|
| 2Back | 10M | 32-64 | 128 | 16 | 8 | 2 | 256 | 512 |
| Local Count | 10M | 64-128 | 128 | 16 | 8 | 3 | 128 | 512 |
| Flip-Flop | 10M | 32-64 | 128 | 4 | 8 | 4 | 256 | 512 |
| Copy | 20M | 32-64 | 128 | 32 | 16 | 2 | 256 | 1024 |
| Reverse | 30M | 32-64 | 128 | 32 | 8 | 6 | 256 | 512 |
| MQMTAR | 50M | 32-64 | 128 | 256 | 16 | 4 | 512 | 1024 |
| Sort | 40M | 32-64 | 128 | 32 | 8 | 2 | 256 | 1024 |

size is set to 524K tokens. For tokenization, we employ the LLaMA-3.2-1B tokenizer. For NAPE experiments, we choose original exponential ALiBi slopes.

We use the following benchmarks for short context evaluation: Lambada (Paperno et al., 2016), HellaSwag (Zellers et al., 2019), PIQA (Bisk et al., 2020), ARC-C (Clark et al., 2018), Winogrande (Sakaguchi et al., 2020), and OpenBookQA (Mihaylov et al., 2018). For length generalization, we use 500 documents from each dataset (ArXiv and PubMed), with all documents at least 16K tokens long, and 500 samples (default number of samples in RULER) for each length for S-NIAH-1 and S-NIAH-2. We evaluate models with RoPE in two settings: vanilla RoPE and RoPE with Adjusted Base Frequency (ABF) (Xiong et al., 2024), where the base frequency is set to 500,000, i.e., $50\times$ used in pre-training. Results for language modeling are in Appendix H.9.

# H    DETAILED RESULTS

In this section, we provide a more detailed analysis of each task.

## H.1    2BACK

Table 6: Accuracy (%) on 2Back.

| | ID | Out-of-Distribution | | | | | |
|-------|------|------|------|------|------|------|------|
| Model | 64 | 128 | 256 | 512 | 1024 | 2048 | 4096 |
| *RoPE* | | | | | | | |
| Softmax | 100.0 | 100.0 | 100.0 | 99.3 | 81.4 | 63.4 | 41.1 |
| SSMax | 100.0 | 100.0 | 100.0 | 99.8 | 98.5 | 90.4 | 69.0 |
| Entmax | 100.0 | 98.2 | 94.8 | 83.7 | 65.5 | 45.7 | 31.3 |
| ASEntmax | 100.0 | 100.0 | 100.0 | 95.0 | 61.2 | 36.2 | 22.1 |
| *NoPE* | | | | | | | |
| Softmax | 99.9 | 83.2 | 51.4 | 30.1 | 18.5 | 12.7 | 9.7 |
| SSMax | 100.0 | 80.5 | 47.2 | 27.2 | 16.9 | 11.8 | 9.2 |
| Entmax | 100.0 | 77.3 | 50.5 | 33.2 | 20.7 | 13.7 | 10.2 |
| ASEntmax | 100.0 | 90.3 | 55.1 | 36.5 | 24.3 | 16.2 | 11.5 |
| *NAPE* | | | | | | | |
| Softmax | 100.0 | 100.0 | 100.0 | 100.0 | 100.0 | 100.0 | 100.0 |
| SSMAx | 100.0 | 100.0 | 100.0 | 100.0 | 100.0 | 100.0 | 100.0 |
| Entmax | 100.0 | 100.0 | 100.0 | 100.0 | 100.0 | 100.0 | 100.0 |
| ASEntmax | 100.0 | 100.0 | 100.0 | 100.0 | 100.0 | 100.0 | 100.0 |

Following the hypothesis of Kazemnejad et al. (2023) that NoPE can learn a relative positional bias, we conducted experiments on the simple 2Back task, in which the model must predict the class of the token two positions earlier. Table 6 shows that models equipped with NoPE achieve perfect in-distribution performance. Moreover, the attention maps in Figure 11 (left) confirm that NoPE indeed acquires relative positional encodings, thus supporting the hypothesis. However, the

OOD attention maps (right part) reveal that, as sequence length increases, the recency bias diffuses unevenly across positions. Such behavior is detrimental in tasks requiring attention to a fixed-size local context (e.g., associative recall, previous instructions in code, n-grams in text). By contrast, ALiBi constrains attention to a local window irrespective of content. Moreover, we observe that ASEntmax partially mitigates diffusion in the attention maps (bottom right), which is also reflected in the accuracy gains shown in Table 11. In our design of positional encoding, NoPE + ALiBi (NAPE), half of the attention heads employ a faster-decaying variant of ALiBi to enforce a short-span focus, while the remaining heads use NoPE which can 1) learn a focus that spans longer and depends on position 2) guide attention semantically.

## H.2 LOCAL COUNT

As observed (Table 7), models using ALiBi perfectly solve the task, which is unsurprising given that ALiBi induces a recency bias. Furthermore, the results for ALiBi and NAPE suggest that models can rely exclusively on ALiBi heads in case of NAPE. With NoPE, however, the model is challenged because identical tokens are not distinguishable at the very first layer. Therefore, the model must develop a mechanism to locate the current cluster. Figure 12 indicates that by the third layer, the NoPE model exhibits a relative positional bias. Combined with the bias observed in 2Back, this indicates that NoPE models can acquire various content-based recency biases that differ from those induced by ALiBi or RoPE. Finally, for Local Count, we observe no improvement in NoPE models when using attention scaling.

Table 7: Accuracy (%) on Local Count.

| | ID | Out-of-distribution | | | | |
|---|---|---|---|---|---|---|
| **Model** | **128** | **256** | **512** | **1024** | **2048** | **4096** |
| *RoPE* | | | | | | |
| Softmax | 100.0 | 99.4 | 91.6 | 55.2 | 31.1 | 17.3 |
| SSMAx | 100.0 | 100.0 | 81.3 | 42.6 | 23.0 | 13.4 |
| Entmax | 100.0 | 99.9 | 89.3 | 47.1 | 24.9 | 14.1 |
| ASEntmax | 100.0 | 100.0 | 79.1 | 41.4 | 22.4 | 13.1 |
| *NoPE* | | | | | | |
| Softmax | 99.1 | 71.7 | 36.5 | 18.3 | 9.2 | 4.6 |
| SSMax | 99.1 | 71.4 | 36.8 | 18.6 | 9.3 | 4.7 |
| Entmax | 99.8 | 80.8 | 45.6 | 25.0 | 13.8 | 7.7 |
| ASEntmax | 99.6 | 78.1 | 42.6 | 22.5 | 11.9 | 6.4 |
| *ALiBi* | | | | | | |
| Softmax | 100.0 | 100.0 | 100.0 | 100.0 | 100.0 | 100.0 |
| Entmax | 100.0 | 100.0 | 100.0 | 100.0 | 100.0 | 100.0 |
| *NAPE* | | | | | | |
| Softmax | 100.0 | 100.0 | 100.0 | 100.0 | 100.0 | 100.0 |
| SSMax | 100.0 | 100.0 | 99.9 | 99.9 | 99.8 | 99.8 |
| Entmax | 100.0 | 100.0 | 100.0 | 100.0 | 100.0 | 100.0 |
| ASEntmax | 100.0 | 100.0 | 100.0 | 100.0 | 100.0 | 100.0 |

## H.3 MAX RETRIEVAL

Solving this task requires an extremely concentrated attention distribution. Such distribution can be achieved by either lowering the temperature $\theta$ for softmax or increasing the entmax parameter $\alpha$. As Table 8 shows, increasing $\alpha$ yields substantial performance gains. However, if $\alpha$ becomes too large, the distribution may collapse to a one-hot vector, causing entmax to lose all gradient signal and hindering learning (e.g. $\alpha > 16$). Instead, this issue can be alleviated by scaling entmax based on the sequence length. With this approach, ASEntmax with $\alpha = 1.5$, learned $\beta$, and elevated $\gamma$ achieves substantially improved performance on the task. We also note that Top-K with K=2 works roughly as

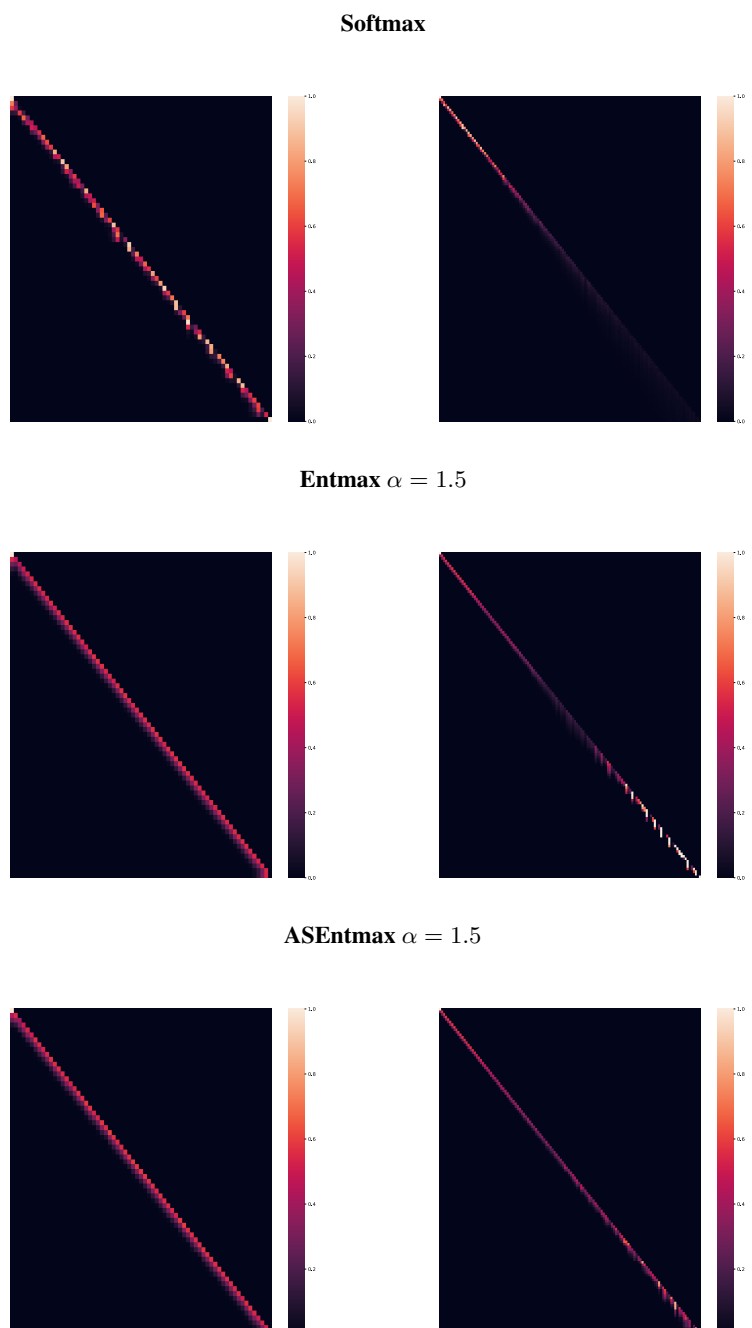

Figure 11: Comparison of attention maps for the 2Back task. **Left:** In-distribution, sequence length 64. **Right:** Out-of-distribution, sequence length 512 (for visualization clarity, we applied max pooling with a window size 4 and stride 4). The maps are shown for the second layer for all models. We can observe that diagonal patterns are less distorted with $\alpha$-entmax. Moreover, ASEntmax mitigate dispersal of the diagonal pattern up to $3\times$ the in-distribution sequence length and make it less distorted up to $8\times$.

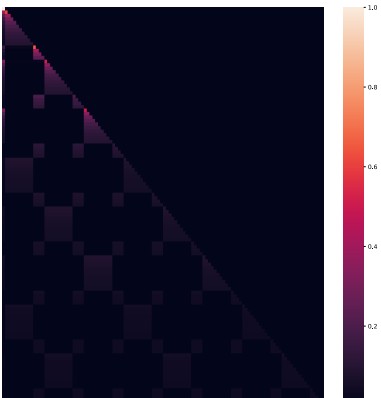 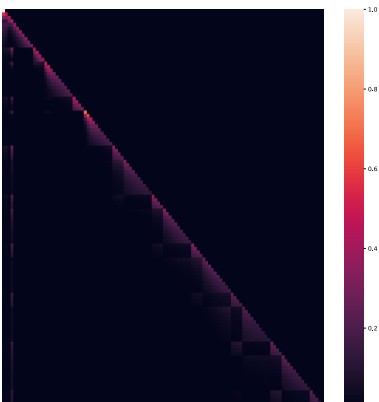

Figure 12: Attention maps of Entmax model on Local Count. **Left:** Layer 1. **Right:** Layer 3; We observe a local pattern: attention weights fade as relative distance increases. Input sequence: `(1..1 ×10, 2..2 ×4, 3..3 ×10, 2..2 ×4, 4..4 ×10, 2..2 ×4)..×4`

well as SSMax. However, lowering K to 1 is not possible since this would essentially reduce Top-K to an argmax operation, a non-differentiable function.

Table 8: Accuracy (%) on Max Retrieval

| | ID | Out-of-Distribution | | | | | | | |
|---|---|---|---|---|---|---|---|---|---|
| **Model** | **16** | **32** | **64** | **128** | **256** | **512** | **1024** | **2048** | **4096** |
| Softmax Veličković et al. (2025) | 98.6 | 97.1 | 94.3 | 89.7 | 81.3 | 70.1 | 53.8 | 35.7 | 22.6 |
| Adapt. temp. Veličković et al. (2025) | 98.6 | 97.1 | 94.5 | 89.9 | 82.1 | 72.5 | 57.7 | 39.4 | 24.9 |
| Softmax $\theta = \sqrt{d}$ | 99.2 | 98.5 | 96.7 | 93.2 | 86.7 | 73.5 | 54.4 | 36.4 | 24.1 |
| Softmax $\theta = 0.1$ | 99.5 | 99.0 | 97.8 | 95.1 | 89.6 | 77.9 | 60.2 | 41.2 | 28.5 |
| Softmax $\theta = 0.0004$ | 99.2 | 98.4 | 97.0 | 94.2 | 89.4 | 81.8 | 71.4 | 58.4 | 43.4 |
| SSMax | 99.4 | 98.9 | 97.8 | 95.9 | 92.3 | 85.0 | 74.7 | 59.9 | 44.7 |
| Top-K, $K = 2$ | 99.4 | 98.6 | 97.6 | 95.4 | 91.3 | 85.0 | 75.3 | 62.4 | 48.3 |
| Top-K, $K = 4$ | 98.8 | 97.7 | 95.0 | 89.5 | 79.1 | 64.6 | 48.9 | 38.9 | 32.4 |
| Entmax $\alpha = 1.5$ | 99.4 | 98.8 | 97.4 | 94.7 | 89.9 | 80.1 | 65.1 | 50.0 | 36.8 |
| Entmax $\alpha = 2$ | 99.5 | 99.1 | 98.0 | 96.0 | 92.1 | 84.5 | 72.0 | 58.4 | 44.6 |
| Entmax $\alpha = 4$ | 99.5 | 98.9 | 97.7 | 95.9 | 92.1 | 84.8 | 75.2 | 61.4 | 46.9 |
| Entmax $\alpha = 16$ | 99.6 | 99.4 | 98.7 | 97.5 | 95.2 | 91.0 | 82.8 | 70.3 | 53.4 |
| Entmax $\alpha = 32$ | 99.4 | 98.7 | 97.5 | 95.5 | 91.5 | 83.8 | 72.6 | 57.5 | 41.7 |
| Entmax $\alpha = 64$ | 99.1 | 98.4 | 96.8 | 93.9 | 88.7 | 78.6 | 64.6 | 45.5 | 28.1 |
| ASEntmax, $\alpha = 1.5$, $\beta_{learn}$, $\gamma = 1$ | 99.5 | 99.0 | 98.1 | 96.3 | 93.1 | 86.1 | 76.2 | 61.9 | 44.5 |
| ASEntmax, $\alpha = 1.5$, $\beta_{learn}$, $\gamma = 2$ | 99.6 | 99.2 | 98.4 | 96.9 | 94.4 | 89.0 | 81.4 | 69.5 | 55.1 |
| ASEntmax, $\alpha = 1.5$, $\beta_{learn}$, $\gamma = 3$ | 99.6 | 99.4 | **99.0** | **98.0** | **96.0** | **92.4** | **85.9** | **76.1** | **62.7** |
| ASEntmax, $\alpha = 1.5$, $\beta_{learn}$, $\gamma = 4$ | 99.3 | 98.7 | 97.6 | 95.4 | 91.3 | 84.6 | 73.6 | 59.7 | 45.9 |

### H.4 MQMTAR

We observe the same pattern across all tasks: despite theoretical extrapolation to $16\times$ via RoPE scaling, RoPE models poorly generalize beyond $4\times$. Moreover, although models with ALiBi can extrapolate up to $64\times$, ALiBi's limited span inevitably leads to performance degradation on very long sequences. However, NAPE provides a substantial boost in all models. As in the copy task,

Entmax alone underperforms Softmax, but adaptive scaling in ASEntmax makes the model superior, extending the generalization to an impressive $1024\times$. We also conducted experiments with ASSMax to demonstrate that, despite adaptive scaling benefits and improved performance in comparison with the model without scaling, softmax dispersion still causes a significant performance drop on very long sequences.

Table 9: Exact match accuracy (%) on Multi-query Multi-token Associative Recall.

| | ID | Out-of-Distribution | | | | | | | | | |
|---|---|---|---|---|---|---|---|---|---|---|---|
| **Model** | **64** | **128** | **256** | **512** | **1024** | **2048** | **4096** | **8192** | **16K** | **32K** | **65K** |
| *RoPE* | | | | | | | | | | | |
| Softmax | 100.0 | 3.1 | 0 | 0 | 0 | 0 | 0 | 0 | 0 | 0 | 0 |
| SSMAx | 99.8 | 6.2 | 0 | 0 | 0 | 0 | 0 | 0 | 0 | 0 | 0 |
| Entmax | 99.8 | 49.4 | 4.5 | 0 | 0 | 0 | 0 | 0 | 0 | 0 | 0 |
| ASEntmax | 100.0 | 66.9 | 0.8 | 0 | 0 | 0 | 0 | 0 | 0 | 0 | 0 |
| *NoPE* | | | | | | | | | | | |
| Softmax | 100.0 | 30.6 | 0 | 0 | 0 | 0 | 0 | 0 | 0 | 0 | 0 |
| Entmax | 100.0 | 26.1 | 0 | 0 | 0 | 0 | 0 | 0 | 0 | 0 | 0 |
| SSMax | 100.0 | 39.1 | 0 | 0 | 0 | 0 | 0 | 0 | 0 | 0 | 0 |
| ASEntmax | 100.0 | 58.3 | 0 | 0 | 0 | 0 | 0 | 0 | 0 | 0 | 0 |
| *ALiBi* | | | | | | | | | | | |
| Softmax | 100.0 | 100.0 | 99.5 | 99.0 | 98.0 | 93.9 | 77.8 | 38.8 | 6.8 | 0 | 0 |
| Entmax | 99.5 | 97.5 | 90.6 | 75.4 | 44.7 | 11.7 | 1.1 | 0 | 0 | 0 | 0 |
| *NAPE* | | | | | | | | | | | |
| Softmax | 100.0 | 100.0 | 100.0 | 99.4 | 99.5 | 98.2 | 97.8 | 90.2 | 80.2 | 34.2 | 3.0 |
| SSMax | 100.0 | 100.0 | 99.9 | 99.9 | 99.8 | 99.6 | 98.3 | 97.4 | 90.6 | 74.0 | 26.7 |
| ASSMax | 100.0 | 100.0 | 100.0 | 99.7 | 99.8 | 99.7 | 98.7 | 94.2 | 85.2 | 72.8 | 21.7 |
| Top-K, $K = 16$ | 100.0 | 100.0 | 99.9 | 41.8 | 5.8 | 0.3 | 0 | 0 | 0 | 0 | 0 |
| Top-K, $K = 32$ | 100.0 | 100.0 | 99.9 | 45.9 | 6.2 | 0.8 | 0 | 0 | 0 | 0 | 0 |
| Entmax | 100.0 | 100.0 | 100.0 | 99.9 | 99.2 | 97.8 | 92.7 | 86.2 | 66.8 | 35.8 | 9.3 |
| ASEntmax | 100.0 | 100.0 | 100.0 | 100.0 | 100.0 | 99.7 | 99.6 | 99.2 | 99.0 | 97.8 | 95.3 |
| SB Attention | 100.0 | 100.0 | 100.0 | 99.7 | 99.8 | 99.2 | 94.6 | 68.6 | 12.4 | 0.4 | 0 |

## H.5 COPY

As shown by Jelassi et al. (2023), transformers generalize to the Copy task especially with appropriate positional encodings. Table 10 shows that the Softmax transformer generalizes up to $64\times$ the ID length. Notably, SSMax outperforms all other models which might suggest that scaling is crucial for this task.

As we can also see, without scaling, $\alpha$-entmax can hurt performance, leading to a noticeable drop in accuracy in the OOD scenario. Introducing an adaptive temperature, however, substantially mitigates this effect: ASEntmax matches Softmax performance outperforming Entmax 4 times of the sequence length. We hypothesize that Copy requires less sparse attention patterns, which can be accomplished by applying a negative power to the logarithm function. We confirm this hypothesis in Figure 13, which shows that ASEntmax learns negative values of $\gamma$ in all heads, resulting in more spread-out attention distributions.

## H.6 FLIP-FLOP

We first conducted an ablation study to evaluate model performance with various RoPE scaling factors (Table 11). Although the random baseline accuracy for Flip-Flop is $50\%$, our generative training setup with a vocabulary of 7 tokens (4 main and 3 special) can yield accuracies below $50\%$. Therefore, we treat accuracies at or below $50\%$ as poor and select a scaling factor of 16 as optimal. The RoPE scaling factor defines the expansion multiple to which the model must generalize. Throughout all

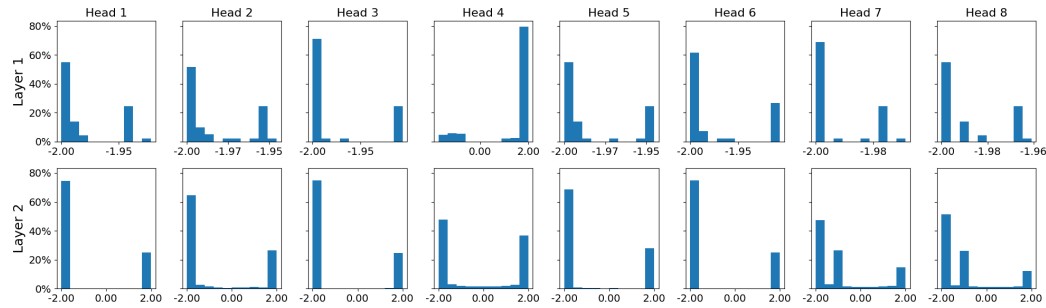

Figure 13: Distributions of $\gamma$ per head and layer for ASEntmax trained on Copy.

Table 10: Exact match accuracy (%) on Copy task.

| | ID | Out-of-Distribution | | | | | |
|---|---|---|---|---|---|---|---|
| **Model** | **64** | **128** | **256** | **512** | **1024** | **2048** | **4096** |
| *RoPE* | | | | | | | |
| Softmax | 100.0 | 2.8 | 0 | 0 | 0 | 0 | 0 |
| SSMax | 100.0 | 0 | 0 | 0 | 0 | 0 | 0 |
| ASSMax | 99.9 | 19.9 | 0 | 0 | 0 | 0 | 0 |
| Entmax | 100.0 | 34.3 | 0 | 0 | 0 | 0 | |
| ASEntmax | 100.0 | 5.3 | 0 | 0 | 0 | 0 | 0 |
| *NoPE* | | | | | | | |
| Softmax | 56.3 | 0 | 0 | 0 | 0 | 0 | 0 |
| SSMax | 56.1 | 0 | 0 | 0 | 0 | 0 | 0 |
| Entmax | 34.6 | 0 | 0 | 0 | 0 | 0 | 0 |
| ASEntmax | 45.8 | 0 | 0 | 0 | 0 | 0 | 0 |
| *ALiBi* | | | | | | | |
| Softmax | 100.0 | 99.8 | 99.8 | 98.8 | 98.3 | 93.9 | 26.8 |
| Entmax | 100.0 | 100.0 | 96.6 | 14.6 | 0.1 | 0 | 0 |
| *NAPE* | | | | | | | |
| Softmax | 100.0 | 100.0 | 99.9 | 99.9 | 99.4 | 96.1 | 85.5 |
| SSMax | 100.0 | 100.0 | 100.0 | 99.9 | 99.6 | 99.3 | 95.8 |
| ASSMax | 99.9 | 99.8 | 99.7 | 99.3 | 97.5 | 91.1 | 72.8 |
| Top-K, $K = 16$ | 100.0 | 99.9 | 86.3 | 0.6 | 0.0 | 0.0 | 0.0 |
| Top-K, $K = 32$ | 100.0 | 99.7 | 96.8 | 26.7 | 0.0 | 0.0 | 0.0 |
| Entmax | 100.0 | 99.0 | 86.0 | 28.5 | 0.2 | 0 | 0 |
| SEntmax | 100.0 | 100.0 | 99.9 | 99.0 | 96.2 | 69.7 | 6.5 |
| ASEntmax | 100.0 | 100.0 | 99.9 | 99.7 | 99.4 | 96.3 | 86.6 |
| SB Attention | 100.0 | 100.0 | 100.0 | 99.6 | 98.8 | 48.6 | 0.0 |

experiments, however, we observe that RoPE models poorly generalize at sequence lengths $8\times$ the in-distribution length.

While Flip-Flop is considered a challenging task for testing length extrapolation (Liu et al., 2023), we found that ALiBi and NAPE strategy almost perfectly solves both the sparse and dense variants. Surprisingly, RoPE models generalize better with the sparse variants.

## H.7 REVERSE

From Table 14, we can see that ASEntmax with NAPE achieved impressive $8\times$ length generalization which to our knowledge, represents the largest extrapolation reported. Moreover, RoPE models fail even at a sequence length of 96. Although NAPE improves Softmax and SSMax, it does not enable

Table 11: Exact match accuracy (%) for ablation of LLaMA 3 RoPE scaling on Flip-Flop (sparse)

| Model | Factor | ID 64 | Out-of-Distribution 128 | 256 | 512 | 1024 | 2048 | 4096 |
|-------|--------|-------|------|-----|-----|------|------|------|
| Softmax | - | 100.0 | 79.9 | 54.4 | 51.5 | 48.8 | 50.8 | 50.8 |
| Softmax | 4 | 100.0 | 99.6 | 33.8 | 11.2 | 3.3 | 0.5 | 0.0 |
| Softmax | 8 | 100.0 | 100.0 | 72.6 | 0.2 | 0 | 0 | 0 |
| Softmax | 16 | 100.0 | 99.9 | 97.3 | 36.7 | 0 | 0 | 0 |
| Softmax | 32 | 100.0 | 99.2 | 71.6 | 51.3 | 51.1 | 49.3 | 49.2 |

Table 12: Accuracy (%) on Flip-Flop (sparse).

| Model | ID 64 | Out-of-Distribution 128 | 256 | 512 | 1024 | 2048 | 4096 | 8192 | 16K | 32K |
|-------|-------|------|-----|-----|------|------|------|------|-----|-----|
| *RoPE* | | | | | | | | | | |
| Softmax | 100.0 | 99.9 | 97.2 | 36.6 | 0.0 | 0.0 | 0.0 | - | - | - |
| SSMax | 100.0 | 99.8 | 91.8 | 77.8 | 52.2 | 22.0 | 39.2 | - | - | - |
| Entmax | 100.0 | 99.9 | 89.0 | 64.0 | 50.6 | 50.6 | 55.1 | - | - | - |
| ASEntmax | 100.0 | 99.8 | 98.9 | 51.4 | 51.4 | 50.2 | 49.2 | - | - | - |
| *ALiBi* | | | | | | | | | | |
| Softmax | 100.0 | 99.9 | 99.8 | 99.9 | 99.9 | 100.0 | 99.7 | 99.7 | 99.9 | 99.7 |
| Entmax | 100.0 | 99.9 | 99.8 | 99.8 | 99.8 | 99.9 | 99.7 | 99.7 | 99.7 | 99.7 |
| *NAPE* | | | | | | | | | | |
| Softmax | 100.0 | 99.8 | 99.6 | 99.3 | 99.7 | 99.6 | 99.6 | 99.6 | 99.3 | 99.4 |
| SSMax | 100.0 | 99.8 | 99.9 | 99.8 | 99.8 | 99.7 | 99.8 | 100.0 | 99.9 | 99.6 |
| Entmax | 100.0 | 99.9 | 99.8 | 99.9 | 99.9 | 100.0 | 99.7 | 99.7 | 99.8 | 99.7 |
| ASEntmax | 100.0 | 99.8 | 99.6 | 99.3 | 99.5 | 99.3 | 99.5 | 99.7 | 99.6 | 99.5 |

Table 13: Accuracy (%) on Flip-Flop (dense).

| Model | ID 64 | Out-of-Distribution 128 | 256 | 512 | 1024 | 2048 | 4096 |
|-------|-------|------|-----|-----|------|------|------|
| *RoPE* | | | | | | | |
| Softmax | 100.0 | 70.2 | 62.2 | 53.2 | 49.2 | 50.3 | 53.1 |
| SSMax | 100.0 | 69.1 | 60.4 | 53.2 | 48.5 | 51.0 | 53.1 |
| Entmax | 100.0 | 80.4 | 73.6 | 60.3 | 49.3 | 51.2 | 53.1 |
| ASEntmax | 100.0 | 100.0 | 100.0 | 49.6 | 48.9 | 51.1 | 53.1 |
| *NAPE* | | | | | | | |
| Softmax | 100.0 | 100.0 | 100.0 | 100.0 | 100.0 | 100.0 | 100.0 |
| SSMAx | 100.0 | 100.0 | 100.0 | 100.0 | 100.0 | 100.0 | 100.0 |
| Entmax | 100.0 | 100.0 | 100.0 | 100.0 | 100.0 | 100.0 | 100.0 |
| ASEntmax | 100.0 | 100.0 | 100.0 | 100.0 | 100.0 | 100.0 | 100.0 |

generalization beyond $1.5\times$ the in-distribution length; however, applying adaptive scaling to Softmax (ASSMax) enables performance at $2\times$ the in-distribution length.

## H.8    SORT

Table 15 demonstrates superiority of $\alpha$-entmax on Sort, with two-layer models generalizing almost perfectly to $2\times$ under both NoPE and NAPE configurations. Furthermore, Softmax models with NAPE experience a performance decline relative to their NoPE counterparts, and adaptive scaling degrades performance for both Softmax and $\alpha$-entmax (we also report results for NoPE + SEntmax

Table 14: Exact match accuracy (%) on Reverse.

| Model | ID 64 | Out-of-Distribution 96 | 128 | 256 | 512 |
|---|---|---|---|---|---|
| *RoPE* | | | | | |
| Softmax | 100.0 | 0 | 0 | 0 | 0 |
| SSMax | 100.0 | 0 | 0 | 0 | 0 |
| ASSMax | 100.0 | 0 | 0 | 0 | 0 |
| Entmax | 100.0 | 0 | 0 | 0 | 0 |
| ASEntmax | 100.0 | 0 | 0 | 0 | 0 |
| *NoPE* | | | | | |
| Softmax | 100.0 | 0 | 0 | 0 | 0 |
| SSMax | 100.0 | 0 | 0 | 0 | 0 |
| Entmax | 100.0 | 77.1 | 0 | 0 | 0 |
| ASEtmax | 100.0 | 74.4 | 0 | 0 | 0 |
| *ALiBi* | | | | | |
| Softmax | 100.0 | 0 | 0 | 0 | 0 |
| Entmax | 100.0 | 96.1 | 78.5 | 0 | 0 |
| *NAPE* | | | | | |
| Softmax | 100.0 | 36.0 | 0 | 0 | 0 |
| SSMax | 100.0 | 54.6 | 0 | 0 | 0 |
| Top-K, $K = 16$ | 100.0 | 99.4 | 83.3 | 0.0 | 0.0 |
| Top-K, $K = 32$ | 100.0 | 100.0 | 98.7 | 57.0 | 0 |
| Top-K, $K = \frac{n}{2}$ | 100.0 | 100.0 | 98.8 | 68.9 | 0 |
| ASSMax | 100.0 | 98.7 | 62.4 | 0 | 0 |
| Entmax | 100.0 | 99.0 | 86.0 | 28.5 | 0.2 |
| SEntmax | 100.0 | 100.0 | 98.1 | 51.4 | 0.0 |
| ASEntmax | 100.0 | 100.0 | 99.8 | 96.4 | 56.7 |
| SB Attention | 100.0 | 100.0 | 99.1 | 0.0 | 0.0 |

to be convinced). However, combining NAPE with adaptive scaling enhances $\alpha$-entmax. This pattern suggest that sparsity, adaptive scaling, and NAPE can act complementarily.

## H.9 LANGUAGE MODELING

We present a comparison between RoPE and NAPE models on standard benchmarks in Table 16, and on length generalization in Table 17. NAPE alone improves short-context results on LAMBADA, HellaSwag, and PIQA. Furthermore, length generalization capabilities only emerge when using either RoPE with ABF or NAPE. While SSMax, Entmax, and ASEntmax with RoPE + ABF achieve near-perfect generalization on the S-NIAH-1 task, NAPE combined with scalable models demonstrates more consistent performance across all sequence lengths in both S-NIAH-1 and S-NIAH-2. Perhaps more importantly, we find that RoPE with ABF leads to a decline in in-distribution performance. For instance, Lambda perplexity/accuracy drops, as does accuracy on HellaSwag, PIQA, ARC-C, and the in-distribution S-NIAH tasks. Moreover, both Softmax + ABF and SSMax + ABF fail dramatically on S-NIAH-2. Given that NAPE shows more consistent performance across different tasks and sequence lengths, we believe it stands out as a more robust positional encoding approach than RoPE.

## I   AI ASSISTANTS

We used Cursor during development, and ChatGPT during paper writing for correcting grammar and polishing sentences.

Table 15: Exact match accuracy (%) on Sort.

| Model | ID 64 | Out-of-Distribution 128 | 256 | 512 |
|---|---|---|---|---|
| *RoPE* | | | | |
| Softmax | 100.0 | 0 | 0 | 0 |
| SSMax | 100.0 | 0 | 0 | 0 |
| Entmax | 100.0 | 0 | 0 | 0 |
| ASentmax | 100.0 | 0 | 0 | 0 |
| *NoPE* | | | | |
| Softmax | 100.0 | 97.6 | 46.6 | 0 |
| SSMax | 100.0 | 96.3 | 29.8 | 0 |
| Top-K, $K = 16$ | 99.7 | 48.0 | 0 | 0 |
| Top-K, $K = 32$ | 100.0 | 92.5 | 0 | 0 |
| Entmax | 100.0 | 99.9 | 66.2 | 0 |
| SEntmax | 100.0 | 99.4 | 47.7 | 0 |
| ASEnmtax | 100.0 | 97.5 | 20.8 | 0 |
| *ALiBi* | | | | |
| Softmax | 99.9 | 0 | 0 | 0 |
| Entmax | 99.2 | 0 | 0 | 0 |
| *NAPE* | | | | |
| Softmax | 100.0 | 0 | 0 | 0 |
| SSMax | 100.0 | 0 | 0 | 0 |
| ASSMax | 100.0 | 99.5 | 9.4 | 0 |
| Entmax | 100.0 | 99.3 | 57.8 | 0 |
| ASEntmax | 100.0 | 100.0 | 79.7 | 0 |
| SB Attention | 100.0 | 36.5 | 0 | 0 |

Table 16: Downstream-task results on short-context datasets encompassing a comparison between RoPE and NAPE.

| | Method | Lambada (PPL) | Lambada | Hellaswag | PIQA | Arc-C | WinoGrande | OpenbookQA |
|---|---|---|---|---|---|---|---|---|
| RoPE | Softmax | 59.7 | 30.3 | 32.5 | 64.3 | 26.3 | 50.8 | 28.6 |
| | SSMax | 54.5 | 30.9 | 32.3 | 63.4 | 26.0 | 51.7 | 31.0 |
| | Entmax | 53.0 | 31.1 | 32.9 | 62.9 | 25.8 | 49.9 | 27.8 |
| | ASEntmax | 56.5 | 29.9 | 32.2 | 62.4 | 25.5 | 52.3 | 27.8 |
| | + ABF, Softmax | 126.3 | 19.0 | 31.5 | 63.8 | 24.6 | 51.8 | 28.8 |
| | + ABF, SSMax | 96.8 | 21.6 | 31.9 | 63.2 | 24.7 | 51.1 | 30.2 |
| | + ABF, Entmax | 101.9 | 20.3 | 32.3 | 62.2 | 25.5 | 50.6 | 27.2 |
| | + ABF, ASEntmax | 99.0 | 20.0 | 29.4 | 62.7 | 24.1 | 52.0 | 29.2 |
| NAPE | Softmax | 52.3 | 30.9 | 33.1 | 65.1 | 25.6 | 49.5 | 28.2 |
| | SSMax | 48.9 | 31.6 | 32.9 | 65.1 | 25.0 | 51.5 | 30.4 |
| | Entmax | 47.9 | 32.1 | 32.8 | 63.6 | 24.6 | 50.9 | 29.0 |
| | ASEntmax | 41.6 | 34.3 | 33.4 | 63.8 | 26.0 | 50.0 | 28.6 |

Table 17: Retrieval performance on RULER benchmark

| | Model | S-NIAH-1 | | | | | S-NIAH-2 | | | |
|---|---|---|---|---|---|---|---|---|---|---|
| | | ID | | OOD | | | ID | | OOD | |
| | | 1K | 2K | 4K | 8K | 16K | 1K | 2K | 4K | 8K |
| RoPE | Softmax | 99.8 | 79.0 | 0.0 | 0.0 | 0.0 | 99.6 | 11.4 | 0.0 | 0.0 |
| | SSMax | 99.0 | 83.0 | 0.0 | 0.0 | 0.0 | 99.6 | 53.6 | 0.0 | 0.0 |
| | Entmax | 100.0 | 79.6 | 0.0 | 0.0 | 0.0 | 99.6 | 53.0 | 0.0 | 0.0 |
| | ASEntmax | 99.8 | 87.2 | 0.0 | 0.0 | 0.0 | 99.0 | 83.6 | 0.0 | 0.0 |
| | + ABF, Softmax | 99.2 | 97.2 | 93.4 | 75.4 | 75.6 | 0.0 | 0.0 | 0.0 | 0.0 |
| | + ABF, SSMax | 98.2 | 98.0 | 97.6 | 97.4 | 98.0 | 30.8 | 37.4 | 4.4 | 0.2 |
| | + ABF, Entmax | 98.4 | 98.2 | 99.4 | 100.0 | 100.0 | 98.8 | 89.0 | 64.8 | 32.4 |
| | + ABF, ASEntmax | 100.0 | 99.6 | 99.6 | 99.6 | 94.0 | 98.6 | 83.6 | 30.8 | 7.2 |
| NAPE | Softmax | 100.0 | 99.4 | 94.2 | 11.4 | 0.8 | 100.0 | 100.0 | 4.8 | 0.0 |
| | SSMax | 100.0 | 99.8 | 99.2 | 92.0 | 75.2 | 99.4 | 99.2 | 64.4 | 14.8 |
| | Entmax | 99.8 | 99.8 | 89.0 | 21.6 | 1.2 | 99.6 | 99.4 | 64.8 | 7.2 |
| | ASEntmax | 99.6 | 100.0 | 100.0 | 99.8 | 97.4 | 99.4 | 99.4 | 83.2 | 25.4 |

