# OpenReview forum: "Long-Context Generalization with Sparse Attention"
_ICLR.cc/2026/Conference — ICLR 2026 Poster_

### Official Review · Reviewer_7xk8 · 2025-11-02

**Soundness:** 3
**Presentation:** 4
**Contribution:** 3
**Rating:** 6
**Confidence:** 4

**Summary:**

This paper addresses the long-context generalization limitations of transformer architectures by replacing the traditional softmax attention with ASENTMAX, a differentiable sparse transformation that assigns exact zero attention to irrelevant tokens, achieving up to 1000× length extrapolation and superior long-context perplexity.

**Strengths:**

1. The paper provides theoretical analyses to address core limitations of softmax attention. It formally proves that alpha-entmax maintains non-vanishing attention probabilities.
2. ASEntmax introduces head-specific, learnable scaling parameters to adjust logits dynamically.
3. The empirical evaluation is thorough and diverse, covering synthetic tasks (retrieval-focused, memory-dependent, ordering) and real-world language modeling.

**Weaknesses:**

1. It is unclear whether ASEntmax can be integrated with FlashAttention-2, a widely adopted library for efficient softmax attention. Given FlashAttention’s popularity in industrial and academic settings, clarifying ASEntmax’s compatibility with FlashAttention (e.g., modifications needed for sparse logit scaling, runtime overhead comparisons) is critical for its practical adoption.
2. The paper does not explicitly confirm whether ASEntmax inherits the theoretical properties of alpha-entmax in Section 3. For instance, does the learnable temperature parameter disrupt the thresholding effect of ω-entmax that ensures zero attention to irrelevant tokens?
3. The theoretical motivation for ASEntmax’s scaling mechanism (Section 4.1) relies on offsetting the range growth of IID Gaussian logits. However, real-world attention logits are unlikely to follow IID Gaussian distributions—they are influenced by semantic relevance, positional biases, and layer-wise transformations.
4. Proposition 2 claims that alpha-entmax alleviates over-squashing “via stronger gradient signals,” but the reasoning is incomplete.
5. In the MQMTAR task, ASEntmax achieves 1024× length extrapolation, but the paper relies on standard positional encodings (e.g., NAPE, ALiBi). Existing literature shows that many positional encodings struggle with extreme extrapolation due to degraded positional awareness. The paper does not analyze whether NAPE alone can support 1024× extrapolation or if its performance is entirely dependent on ASEntmax.
6. Table 4 shows ASEntmax achieves near-perfect performance (97.4%) on S-NIAH-1 (OOD, 16K tokens) but only 25.4% on S-NIAH-2 (OOD, 8K tokens). The paper does not explain this stark disparity. Is S-NIAH-2 is presumably a harder task？
7. The related works should include a discussion about the broader landscape of sparse attention mechanisms designed for long-context modeling (e.g., [1-7]).
---
[1] Efficient streaming language models with attention sinks. ICLR 2024.

[2] MInference 1.0: Accelerating pre-filling for long-context LLMs via dynamic sparse attention. NeurIPS 2024.

[3] Core Context Aware Transformers for Long Context Language Modeling. ICML 2025.

[4] XAttention: Block Sparse Attention with Antidiagonal Scoring. ICML 2025.

[5] FlexPrefill: A Context-Aware Sparse Attention Mechanism for Efficient Long-Sequence Inference. ICLR 2025.

[6] Curse of High Dimensionality Issue in Transformer for Long Context Modeling. ICML 2025.

[7] MMInference: Accelerating Pre-filling for Long-Context VLMs via Modality-Aware Permutation Sparse Attention. ICML 2025

**Questions:**

NA

---

> ### Author Response · Authors · 2025-11-25
>
> Thank you for your positive comments. We address your main questions below.
>
> **> It is unclear whether ASEntmax can be integrated with FlashAttention-2, a widely adopted library for efficient softmax attention. Given FlashAttention’s popularity in industrial and academic settings, clarifying ASEntmax’s compatibility with FlashAttention (e.g., modifications needed for sparse logit scaling, runtime overhead comparisons) is critical for its practical adoption.**
>
> Yes, ASEntmax can be integrated with FlashAttention-like implementations. In our work, we use AdaSplash [1], which is an efficient implementation of entmax attention in Triton---akin to FlashAttention for softmax. Importantly, ASEntmax is an extension that does not require modifying the Triton kernel itself, since we operate only on the logits. In practice, in order to use AdaSplash, we only modify the query vectors by multiplying them by a query-dependent scale. We will provide our code with detailed instructions about how to do so.
>
> [1] AdaSplash: Adaptive Sparse Flash Attention. Nuno Gonçalves, Marcos Treviso, André F. T. Martins. ICML 2025
>
> **> The paper does not explicitly confirm whether ASEntmax inherits the theoretical properties of alpha-entmax in Section 3. For instance, does the learnable temperature parameter disrupt the thresholding effect of ω-entmax that ensures zero attention to irrelevant tokens?**
>
> Figure 2 shows how sparsity changes with different temperature values. As the inverse temperature approaches zero, entmax becomes denser. Therefore, entmax can return both dense and sparse outputs. This is an inherent property of entmax, and ASEntmax does not change it. Thus, the properties we show for entmax in §3 directly carry over to ASEntmax.
>
> **> The theoretical motivation for ASEntmax’s scaling mechanism (Section 4.1) relies on offsetting the range growth of IID Gaussian logits. However, real-world attention logits are unlikely to follow IID Gaussian distributions—they are influenced by semantic relevance, positional biases, and layer-wise transformations.**
>
> We fully agree that real attention logits might not be IID Gaussian. However, note that our IID Gaussian model only appears in §4.1 as a way to motivate the $(\log n)^\gamma$ factor  in a more principled manner (mirroring the classical derivation of the $1/\sqrt{d}$ ​ factor in transformers, which also assumes IID Gaussians).
>
>
> **> Proposition 2 claims that alpha-entmax alleviates over-squashing “via stronger gradient signals,” but the reasoning is incomplete.**
>
> Proposition 2 shows that, when $\alpha$-entmax has attention on a support of size at most $s$, the effective path counts scale as $O(s^L)$ rather than $O(n^L)$ as in softmax.
> For simplicity, if we assume the output distribution is uniform on its support, each non-zero edge has probability $1/s$ (versus $1/n$ with softmax), so a path in a L-layered model is attenuated by $(1/s)^L$ instead of $(1/n)^L$.
> This relative factor of $(n/s)^L$ is relevant because we usually have $s \ll n$ in practice. This is what we mean by "stronger gradient signals": fewer paths but with larger per-edge probabilities. We will make this clearer in the final version.
>
> **> In the MQMTAR task, ASEntmax achieves 1024× length extrapolation, but the paper relies on standard positional encodings (e.g., NAPE, ALiBi). Existing literature shows that many positional encodings struggle with extreme extrapolation due to degraded positional awareness. The paper does not analyze whether NAPE alone can support 1024× extrapolation or if its performance is entirely dependent on ASEntmax.**
>
> Please note that we compare the performance of various positional encodings in Table 9 (Appendix H.4), which includes RoPE, NoPE, AliBi, and NAPE. We also report results with Stick-Breaking Attention. The results show that the high extrapolation performance is achieved likely due to the complementary benefits of NAPE and ASEntmax.
>
> **> The related works should include a discussion about the broader landscape of sparse attention mechanisms designed for long-context modeling (e.g., [1-7]).**
>
> Thank you for this suggestion, which we will follow. Note that, in contrast to inference-only strategies, we study a more general modification: the choice of the attention transformation (softmax vs α-entmax/ASEntmax) and how it affects length generalization. We will make this distinction explicit while citing the most relevant papers.

---

> > ### Author Response · Authors · 2025-11-25
> >
> > **> Table 4 shows ASEntmax achieves near-perfect performance (97.4%) on S-NIAH-1 (OOD, 16K tokens) but only 25.4% on S-NIAH-2 (OOD, 8K tokens). The paper does not explain this stark disparity. Is S-NIAH-2 is presumably a harder task**？
> >
> > S-NIAH-2 is indeed a more difficult task. This can be seen in the original RULER paper as well as in recent SSM papers, where most models obtain good extrapolation scores on S-NIAH-1 but degrade sharply on S-NIAH-2. Nevertheless, the 25.4% is still quite a good result.
> >
> > To put our numbers in context, below we report the S-NIAH results from the recent [Mamba-3 paper](https://openreview.net/pdf?id=HwCvaJOiCj) (1.5B-parameter models, trained on more tokens) alongside our 420M ASEntmax model:
> >
> >
> > **S-NIAH-1**:
> >
> > | Model    | 1K   | 2K    | 4K    | 8K   | 16K  |
> > | -------- | ---- | ----- | ----- | ---- | ---- |
> > | DeltaNet (1.5B) | 100  | 100   | 99.8  |      |      |
> > | Mamba-2 (1.5B) | 100  | 99.6  | 62.0  |      |      |
> > | Mamba-3 (1.5B)  | 100  | 100   | 88.2  |      |      |
> > | ASEntmax (420M) | 99.6 | 100.0 | 100.0 | 99.8 | 97.4 |
> >
> > **S-NIAH-2**:
> >
> > | Model    | 1K   | 2K   | 4K   | 8K   |
> > | -------- | ---- | ---- | ---- | ---- |
> > | DeltaNet  (1.5B) | 100  | 93.8 | 49.8 |      |
> > | Mamba-2  (1.5B)  | 100  | 53.8 | 11.8 |      |
> > | Mamba-3  (1.5B)  | 100  | 95.4 | 50.6 |      |
> > | ASEntmax (420M) | 99.4 | 99.4 | 83.2 | 25.4 |
> >
> >
> > Even strong SSMs such as DeltaNet and Mamba-3, trained with larger capacity and more data, drop to around 50 on S-NIAH-2 at 2x the pretraining length (4K), while performing much better on S-NIAH-1 in the same regime.
> > Our ASEntmax model follows this general pattern but still compares favorably, especially given its smaller size.

---

> > > ### Comment · Reviewer_7xk8 · 2025-11-28
> > > **Thanks for the responses.**
> > >
> > > The rebuttal has addressed all of my concerns. I keep my scoring.

---

### Official Review · Reviewer_5G4R · 2025-11-02

**Soundness:** 3
**Presentation:** 3
**Contribution:** 3
**Rating:** 6
**Confidence:** 2

**Summary:**

The paper makes attention genuinely sparse with α-entmax and then auto-tunes temperature by context length via ASEntmax so the model still locks onto key tokens in very long sequences. The theory shows softmax spreads probability mass as sequences grow while entmax keeps entropy low and prevents weights from vanishing on important positions. The experiments mix controlled synthetic tasks with language modeling and point to steadier long-context behavior without hurting short-context quality.

**Strengths:**

1. The paper cleanly connects attention dispersion to representation collapse and over-compression with precise definitions and propositions.

2. The method plays nicely with NAPE and existing fast-attention kernels like FlashAttention/AdaSplash, which makes it engineer-friendly.

Overall,I think it is solid, but I'm not fully sure my evaluation is correct.

**Weaknesses:**

1. Entropy bounds rely on bounded-logit or near-Gaussian assumptions that may break on heavy-tailed or high-contrast distributions.
2. Deployment-critical profiles for memory, throughput, and tail latency are thin, especially for large models and ultra-long inputs.
3. Interplay with RoPE extensions, retrieval-style sparsity, or SSM hybrids is discussed but not systematically tested.
4. Robustness beyond NAPE and specific hyperparameters is unclear, so portability to other position encodings or task families needs work.

**Questions:**

1. What happens when the logit distribution departs from IID Gaussian or becomes extremely aligned, and can you show counterexamples and diagnostics.
2. How should this combine with Top-k, chunk routing, or RAG pruning, and are there simple ordering/budgeting rules to follow.
3. Can you provide an end-to-end compute profile and an early-exit strategy so systems can auto-tune ASEntmax intensity by sequence length.

---

> ### Author Response · Authors · 2025-11-25
>
> Thank you for your positive comments! We address your main points below.
>
> **> Deployment-critical profiles for memory, throughput, and tail latency are thin, especially for large models and ultra-long inputs.**
>
> Thanks for raising this point! While speed is not the main focus of our paper, it is an important point that we will make sure to discuss in the final version. In this work we do not focus on runtime and memory benchmarks since we use an efficient implementation called AdaSplash [1], which for high sparsity regimes (commonly the case in long-context scenarios) can lead to faster performance than softmax attention using FlashAttention, especially in the backward pass. We refer to the Adasplah paper (e.g. their Figure 1) for more information on runtime, memory and latency, and we will add these details to the related work section in the final version.
>
> [1] AdaSplash: Adaptive Sparse Flash Attention. Nuno Gonçalves, Marcos Treviso, André F. T. Martins. ICML 2025
>
> **> Interplay with RoPE extensions, retrieval-style sparsity, or SSM hybrids is discussed but not systematically tested.**
>
> We agree these combinations are interesting, but here we intentionally stick to base positional encodings (RoPE, ALiBi, NoPE, NAPE) without applying “extension tricks” to any of them, including NAPE, where such extensions remain underdeveloped. Since ASEntmax operates purely at the logit level, it's orthogonal to RoPE extensions, retrieval-style sparsity patterns, and SSM layers, and could in principle be combined with any of them. We think that this alone deserves a more thorough study and we leave it as future work. Nevertheless, we will discuss some of those extensions in the related work section as they can inspire future work.
>
>
> **> Robustness beyond NAPE and specific hyperparameters is unclear, so portability to other position encodings or task families needs work.**
>
> Please note that we provide additional results with other positional encoding methods in Appendix H, including RoPE, ALiBi, and NoPE. We also have a dedicated section in appendix E that analyzes the interplay between sparse attention and positional encodings, such as RoPE and ALiBi. We will make sure to highlight this more clearly in the main text.
>
>
> **> What happens when the logit distribution departs from IID Gaussian or becomes extremely aligned, and can you show counterexamples and diagnostics.**
>
> In the extreme “aligned” case where all logits are equal (or almost equal), α-entmax behaves like softmax, where it returns (or approaches) the uniform distribution, and thus there is no "scaling" that will prevent obtaining a fully dense output. For non-Gaussian but bounded logits (such as those arising in our experiments), we empirically observe that ASEntmax works well.
>
>
> **> How should this combine with Top-k, chunk routing, or RAG pruning, and are there simple ordering/budgeting rules to follow.**
>
> These mechanisms are indeed fully orthogonal to ASEntmax. In principle, one could use the ASEntmax probabilities to guide top-k thresholds or chunk/routing budgets.
>
> **> Can you provide an end-to-end compute profile and an early-exit strategy so systems can auto-tune ASEntmax intensity by sequence length.**
>
> Since our goal is to isolate the effect of the attention transformation itself, we do not investigate early-exit schemes or auto-tuning. Since ASEntmax is built on AdaSplash, it inherits its efficiency properties. Indeed, designing system-level early-exit strategies on top is an interesting but orthogonal direction.
>
> [1] AdaSplash: Adaptive Sparse Flash Attention. Nuno Gonçalves, Marcos Treviso, André F. T. Martins. ICML 2025

---

### Official Review · Reviewer_HXz2 · 2025-11-02

**Soundness:** 3
**Presentation:** 3
**Contribution:** 3
**Rating:** 6
**Confidence:** 3

**Summary:**

The paper introduces Adaptive-Scalable Entmax (ASEntmax), a sparse attention mechanism that improves long-context generalization by "zeroing out" attention (in an exact manner) on irrelevant tokens.
On multiple datasets and benchmarks, ASEntmax outperforms all baselines.

**Strengths:**

- a very simple change to existing attention mechanism
- works well across different datasets and tasks, showing consistent length generalization compared to baselines

**Weaknesses:**

- there are some understanding problems that i'm facing, refer to the questions

**Questions:**

- how is adaptive scalable entmax different from both scalable softmax and vanilla entmax ? is there a version of entmax that is only "adaptive" and not "scalable"? i understand entmax enables to turn of attention on some tokens below a certain threshold; scalable softmax amplifies attention if logits exceed by some amount compared to minimum logit; why can't vanilla entmax work by itself? why is the scaling solution proposed by SSMax required in this paper?

- i didn't understand the motivation behind figure 3. why should scalers learned in figure 3 help in long-context generalization?

- is QK-norm used before attention? can this fix some of the problems faced in long-context due to bounded logits?

- is there any finetuning used before testing on RULER ?

- why do the results vary a lot when using RoPE vs when using NAPE (e.g. in Appendix H) ?

I am definitely willing to improve the score if these are addressed :)

---

> ### Author Response · Authors · 2025-11-25
>
> Thank you for your positive review! We are happy to address your questions below.
>
> **> How is adaptive scalable entmax different from both scalable softmax and vanilla entmax? is there a version of entmax that is only "adaptive" and not "scalable"? i understand entmax enables to turn of attention on some tokens below a certain threshold; scalable softmax amplifies attention if logits exceed by some amount compared to minimum logit; why can't vanilla entmax work by itself? why is the scaling solution proposed by SSMax required in this paper?**
>
> SSMax [1] rescales logits as a function of sequence length but still uses softmax, so the attention remains dense even at extreme scales. Vanilla α-entmax [2] yields exact zeros but, with fixed $\alpha$, it can become too dense or too sparse as the input grows. ASEntmax achieves a good balance by combining these two ideas: it benefits from the sparsity of the α-entmax mapping, but it makes it more adaptive (and scalable to longer contexts) by introducing head- and query-dependent scalers, allowing each head to adapt the sparsity with sequence length.
>
> [1] Ken M. Nakanishi. 2025. Scalable-Softmax Is Superior for Attention.
> [2] Ben Peters, Vlad Niculae, and André F. T. Martins. 2019. Sparse Sequence-to-Sequence Models. In Proceedings of the 57th Annual Meeting of the Association for Computational Linguistics, pages 1504–1519, Florence, Italy. Association for Computational Linguistics.
>
>
> **> I didn't understand the motivation behind figure 3. why should scalers learned in figure 3 help in long-context generalization?**
>
> In Figure 3, we show that when trying to learn scalers for each position (as done in Figure 2 of the SSMax paper), the resulting pattern can be highly different for each head. For example, some heads learn monotonically increasing scalers, some learn monotonically decreasing ones, and some learn irregular structures at the beginning or end of the sequence. This empirically motivates ASEntmax’s design: instead of a single global scaling rule as in SSMax, we learn head- and query-specific scalers. We also show how fitted curves parametrized by $[\delta + \beta \log n]$ (as in SSMax) and $[\delta + \beta (\log n)^\gamma]$ compare, which further reinforces our idea that having a $\gamma$ parameter is beneficial. We will make this motivation clearer in the final version.
>
>
> **> Is QK-norm used before attention? can this fix some of the problems faced in long-context due to bounded logits?**
>
> No, we didn’t use QK-norm in our experiments. QK-norm basically normalizes queries and keys to a bounded range (or sphere). As mentioned [1], although QK-norm may help stabilize training, geometrically it makes the softmax output distribution less peaked (because it "removes" magnitude information from the query-key dot product). Thus, the theoretical issues of softmax, such as dispersion, remain. Moreover, we also note that [1] also shows that QK-norm hurts performance on the long-context NIAH task.
>
> [1] Bowen Yang · Bharat Venkitesh · Dwaraknath Gnaneshwar Talupuru · Hangyu Lin · David Cairuz · Phil Blunsom · Acyr Locatelli. 2025. Rope to Nope and Back Again: A New Hybrid Attention Strategy. In Proceedings of the 39th Annual Conference on Neural Information Processing Systems.
>
> **> Is there any finetuning used before testing on RULER ?**
>
> No, we don’t fine-tune the models since our goal is to test their true length extrapolation capabilities.
>
> **> Why do the results vary a lot when using RoPE vs when using NAPE (e.g. in Appendix H)?**
>
> Thanks for pointing this out. As several works have shown [1, 2, 3], vanilla RoPE tends to extrapolate poorly once we go significantly beyond the pretraining context length. In our experiments we keep everything else fixed (same architecture, same attention mechanism) and only change the positional encoding. At in-distribution lengths the differences between RoPE and NAPE (our NoPE+ALiBi variant) are modest, but at out-of-distribution lengths RoPE’s known extrapolation issues become much more pronounced, while NAPE remains stable. We believe this reflects a good synergy in NAPE: ALiBi preserves useful relative-distance information via its sliding-window bias, while NoPE lets the model be more “semantic-guided” rather than “position-guided”.
>
> [1] Samy Jelassi, David Brandfonbrener, Sham M. Kakade, and Eran Malach. 2024. Repeat after me: transformers are better than state space models at copying. In Proceedings of the 41st International Conference on Machine Learning (ICML'24), Vol. 235. JMLR.org, Article 863, 21502–21521.
>
> [2] Amirhossein Kazemnejad, Inkit Padhi, Karthikeyan Natesan Ramamurthy, Payel Das, Siva Reddy. 2023. The Impact of Positional Encoding on Length Generalization in Transformers
>
> [3] Ofir Press, Noah A. Smith, Mike Lewis. 2022. Train Short, Test Long: Attention with Linear Biases Enables Input Length Extrapolation

---

### Official Review · Reviewer_xkyo · 2025-11-05

**Soundness:** 4
**Presentation:** 3
**Contribution:** 4
**Rating:** 8
**Confidence:** 2

**Summary:**

The attention scores, even transformed by a highly distorting function (such as softmax), exhibit one unfortunate trait – they tend to have high entropy and get closer to uniform distribution, if the sequence length grows significantly. This work aims to analyze and offer a solution for  this problem by building on previous literature on a specific family of functions – Entmax – which have favorable properties, most notably sparseness. The authors propose a novel variation of this function, ASEntmax with learnable weights which allow regulating sparseness level with the increase of sequence length.

The scope of this work is essentially theoretical, as it examines long-range properties of softmax and Entmax family of transforms, successfully proving several and expanding our understanding of softmax limitations and methods to circumvent them. However, the paper also for the first time (as I’m aware of) empirically validates (AS)Entmax activations in causal language modeling on a scale of 420M parameter models.

Overall, I believe this is a solid work and it should be accepted.

**Strengths:**

* The problem of length generalization of Transformers is well-motivated and relevant.

* The paper is thorough and rigorous in math, although I didn’t check all proofs and derivations.

* The conclusions derived from principled theoretical analysis, are also plausible intuitively and corroborated by evidence.

* The empirical validation is convincing, and the results are promising. Specifically, Entmax and ASEntmax show superior out-of-distribution generalization capabilities as compared to Softmax in causal language modeling on two long-context datasets.

**Weaknesses:**

* It would be helpful to also compare different attention activation functions for language modeling perplexity as in Table 3, but with industry-standard RoPE embeddings. Appendix H.9. does not provide such comparisons, but it seems that the RoPE models are already pre-trained, and you would only need to test their perplexity on these long-context benchmarks.

* The paper could be more self-contained. To fully comprehend the material, it is necessary to familiarize oneself with several previous works about Entmax. The exposition in Appendix A is scarce and for the most part repeats section 2.2. I would appreciate if definitions of Tsallis Entropy, q-expential functions, and a brief recap of sections 2.1-2.2 of Sparsemax paper (https://proceedings.mlr.press/v48/martins16.pdf) and sections 3.1-3.3 of original Entmax paper (https://aclanthology.org/P19-1146.pdf) could be provided.

* The paper doesn’t establish which values of $\alpha$ and the functional form of $\tau(\cdot)$ are used in the experiments.

* There are no comparisons of speed of calculation between the 4 considered algorithms. An earlier work (https://proceedings.mlr.press/v48/martins16.pdf) alludes that run-time complexity for at least some of the Entmax variants is $O(L \log L)$ instead of softmax’s $O(L)$, so it warrants a test for understanding how real-life throughput is impacted.

**Questions:**

- Can ASEntmax be implemented as an efficient Triton kernel, similarly to AdaSplash (https://openreview.net/forum?id=OWIPDWhUcO), without incurring numerical instabilities?

- Will you open-source the model weights for all variants of Llama pre-trained on DCLM data?

- Proposition 2.1 (Preserved representations) – can the opposite be proved for softmax? If not, the statement of this proposition should be altered to include softmax.

---

> ### Author Response · Authors · 2025-11-25
>
> Thank you for your positive review! We are happy that you found the paper relevant, well-motivated, thorough, and rigorous in math, and the empirical validation convincing. We address your main questions below.
>
> **> It would be helpful to also compare different attention activation functions for language modeling perplexity as in Table 3, but with industry-standard RoPE embeddings. Appendix H.9. does not provide such comparisons, but it seems that the RoPE models are already pre-trained, and you would only need to test their perplexity on these long-context benchmarks.**
>
> We did not compare with RoPE extensions since our models with NAPE provided better results across the board for the synthetic experiments. However, we do have some current results in Appendix H (Tables 16 and 17) showing that RoPE + ABF leads to performance degradation in short contexts, but, of course, improving results on RULER, especially when using entmax/ASentmax.
>
> **> The paper could be more self-contained. To fully comprehend the material, it is necessary to familiarize oneself with several previous works about Entmax. The exposition in Appendix A is scarce and for the most part repeats section 2.2. I would appreciate if definitions of Tsallis Entropy, q-expential functions, and a brief recap of sections 2.1-2.2 of Sparsemax paper (https://proceedings.mlr.press/v48/martins16.pdf) and sections 3.1-3.3 of original Entmax paper (https://aclanthology.org/P19-1146.pdf) could be provided.**
>
> This is a very good suggestion.  We agree and will expand Appendix A to make the paper more self-contained and accessible to a broader audience.
>
>
> **> The paper doesn’t establish which values of alpha and the functional form of tau(.) are used in the experiments.**
>
> We use alpha = 1.5 in the LM experiments, which leads to closed-form expressions for alpha-entmax, making AdaSplash (flash version of entmax) more efficient. We report the value of alpha and other hyperparameters for all experiments in Appendix G. We will make sure to state the values of alpha in the main paper.
> Regarding tau, we use the standard α-entmax threshold without any new parameterization. That is, $\tau$ has the same form as in the original α-entmax.
>
> **> There are no comparisons of speed of calculation between the 4 considered algorithms. An earlier work (https://proceedings.mlr.press/v48/martins16.pdf) alludes that run-time complexity for at least some of the Entmax variants is O(L log L) instead of softmax’s O(L), so it warrants a test for understanding how real-life throughput is impacted.**
>
> Thanks for raising this point! While speed is not the main focus of our paper, it is an important point that we will make sure to discuss in the final version. Briefly, the earlier works on entmax variants (such as the ICML 2016 paper you mention) do not provide efficient GPU-friendly algorithms to exploit the sparsity and do not scale well for long context sizes. However, recent work introduced faster algorithms and an efficient kernel implementation, AdaSplash [1], which for high sparsity regimes (commonly the case in long-context scenarios) can lead to faster performance than softmax attention using FlashAttention, especially in the backward pass. We refer to the Adasplah paper (e.g. their Figure 1) for more information on runtime and latency, and we will add these details to the related work section in the final version.
>
> [1] AdaSplash: Adaptive Sparse Flash Attention. Nuno Gonçalves, Marcos Treviso, André F. T. Martins. ICML 2025
>
>
> **> Can ASEntmax be implemented as an efficient Triton kernel, similarly to AdaSplash (https://openreview.net/forum?id=OWIPDWhUcO), without incurring numerical instabilities?**
>
> Yes! We actually use AdaSplash without any modifications to the kernel. This is because ASEntmax is an extension that only modifies the logits (the inputs to entmax). In practice, this means that we modify only the query vectors by multiplying them by a “scalar”, then obtaining the “scaled” logits after the QK^T dot product.
>
>
> **> Will you open-source the model weights for all variants of Llama pre-trained on DCLM data?**
>
> Yes.
>
>
> **> Proposition 2.1 (Preserved representations) – can the opposite be proved for softmax? If not, the statement of this proposition should be altered to include softmax.**
>
> Yes. The complementary negative result for softmax has already been shown by Barbero et al. (2024, Theorem B.3) [1]. We will make this connection more explicit before stating Proposition 2.1.
>
> [1] Federico Barbero, Andrea Banino, Steven Kapturowski, Dharshan Kumaran, João Madeira Araújo, Oleksandr Vitvitskyi, Razvan Pascanu, and Petar Velickovic. Transformers need glasses! information over-squashing in language tasks. NeurIPS 2024.

---

### Meta-Review · Area_Chair_iCS6 · 2025-12-30

**Summary:**

All four reviewers are positive overall (scores: 8, 6, 6, 6) and agree the paper tackles an important and timely issue, i.e., Transformer attention becomes increasingly high-entropy / diffuse as context length grows, harming long-context generalization. The paper’s core contribution—ASEntmax, which combines exact sparsity (α-entmax) with learned, head/query-dependent scaling to maintain useful sparsity across lengths—was viewed as sound, well-motivated, and empirically promising, with a strong theoretical component and credible large-model validation. The main issues raised were not about correctness, but about presentation completeness, reproducibility details, practical deployability, and assumption realism. Given the review and rebuttal, I recommend the paper with a clear accept.

**Reviewer Concerns:**

**Reviewer xkyo (8)**

Addressed by rebuttal

Paper self-containedness: authors agree and commit to expanding Appendix A with the requested background.

Missing hyperparameter details: authors specify α = 1.5 for LM experiments and clarify τ uses the standard α-entmax threshold (no new parameterization).

Kernel efficiency / Triton feasibility: authors state ASEntmax works with AdaSplash and does not require kernel modifications (logit scaling via query rescaling).

Proposition 2.1 wording: they cite an existing negative result for softmax and will clarify.

Still outstanding (minor)

RoPE-based perplexity comparisons: the response explains why they focused on NAPE and points to appendix results, but the reviewer’s specific request (“Table 3 style perplexity comparisons with industry-standard RoPE”) is only partially satisfied unless they add the exact evaluation the reviewer asked for.

Direct speed comparisons across the four algorithms: authors largely defer to AdaSplash and plan to “discuss” runtime.



**Reviewer HXz2 (6)**

Addressed by rebuttal

Differentiation vs scalable softmax & vanilla entmax: authors clearly explain the “why not vanilla entmax alone” and motivate combining scaling (SSMax-style) with exact sparsity (entmax) via head/query scalers.

Motivation behind Figure 3: clarified as empirical evidence that scaling patterns vary by head, motivating head/query-specific scalers.

QK-norm question: answered directly.

RULER finetuning: clarified no finetuning; testing true extrapolation.

RoPE vs NAPE disparity: explained using known RoPE extrapolation limitations and why NAPE stays stable.




**Reviewer 5G4R (6)**

Addressed by rebuttal

Runtime / memory / latency concern: authors point to AdaSplash performance and commit to adding runtime discussion (and related work context).

Assumption failure modes (non-Gaussian / aligned logits): authors provide an “aligned logits” explanation and claim empirical robustness for bounded logits.

Portability beyond NAPE / hyperparameter sensitivity: authors point to appendix results across RoPE/ALiBi/NoPE and an appendix section analyzing interplay.

Still outstanding

Deployment-grade profiling remains thin if they only cite AdaSplash rather than report end-to-end numbers (throughput, memory, tail latency) in their own setup.

System-level guidance (early-exit, auto-tuning intensity, budgeting rules with Top-k/RAG/chunking) is acknowledged but deferred as “orthogonal.” That’s reasonable, but it remains an open practical concern.




**Reviewer 7xk8 (6)**

Addressed by rebuttal

FlashAttention-2 integration / practicality: answered with AdaSplash + “no kernel modification” claim; code/instructions promised.

Does ASEntmax inherit α-entmax properties?: clarified that temperature affects sparsity but this is inherent to entmax; properties carry over.

IID Gaussian motivation realism: framed as a motivating model analogous to the classic √d scaling derivation.

Over-squashing / gradient signal reasoning: authors provide a more concrete path-count argument and promise to clarify.

Positional encoding dependency: authors point to appendix comparisons and argue synergy between NAPE and ASEntmax.

S-NIAH-1 vs S-NIAH-2 disparity: directly explained as task difficulty.

Related work breadth on sparse long-context methods: authors agree to expand.

**Reviewer Scores:**

I believe all reviewers will keep positive scores.

---

### Decision · Program_Chairs · 2026-01-26

Accept (Poster)